# AlphaBench: Benchmarking Large Language Models in Formulaic Alpha Factor Mining

**Haochen Luo**[1] , **Ho Tin Ko**[2] , **Jiandong Chen**[1] , **David Sun**[4] , **Yuan Zhang**[3],[*] **Chen Liu**[1],[*]
[1]City University of Hong Kong    [2]Yuen Long Merchants Association Secondary School
[3]Shanghai University of Finance and Economics    [4]Harrow International School Hong Kong
`chester.hc.luo@my.cityu.edu.hk, ltu46166@gmail.com`
`jiandchen3-c@my.cityu.edu.hk, 0220220405@harrowschool.hk`
`zhang.yuan@sufe.edu.cn, chen.liu@cityu.edu.hk,`

## Abstract

Formulaic alpha factor mining (FAFM) is a central problem in quantitative investment, where interpretable formulas are designed to extract predictive signals from historical financial series. With the emergence of large language models (LLMs), recent studies have begun to explore their roles in FAFM, yet their capabilities across different tasks and configurations remain unclear. In this work, we introduce **AlphaBench**, the first systematic benchmark for evaluating LLMs in FAFM. AlphaBench covers three core tasks, including *factor generation*, *factor evaluation*, and *factor searching*, which are all popular tasks integrated in the workflow of quantitative researchers. Beyond task-level evaluation, we further analyze how different LLM settings, including model type, prompting paradigm, and reasoning strategy, influence performance. Our experiments on a range of open-source and closed-source models reveal that LLMs hold strong potential in automating factor mining, while also facing persistent challenges in robustness, search efficiency, and practical usability. The project is available at: https://alphabench.cc/.

## 1 Introduction

In quantitative investment, an *alpha factor* is a mathematical expression that extracts predictive signals from financial data. By computing factor values for each asset at a given time, investors can *rank assets* and systematically construct portfolios: going long on high-ranked assets and short on low-ranked ones (Fama & French, 1992; 1993; Carhart, 1997). This ranking-based framework is widely used in equity long-short strategies, risk-parity models, and multi-factor portfolios, where alpha signals are combined with risk models and execution constraints to seek excess returns over benchmarks. Because of interpretability, formulaic factors can be easily backtested, stress-tested across markets, and deployed at scale in production trading systems. Typically, a formulaic alpha factor consists of two main components: (1) **operators**, which define mathematical or statistical transformations, and (2) **variables**, such as historical prices and volumes, from which signals are extracted (Harvey et al., 2016; Zhang et al., 2020; Yu et al., 2023). To evaluate factors, practitioners commonly compute the *Information Coefficient (IC)* or *Rank Information Coefficient (RankIC)* to measure the correlation between factor values and realistic future returns (Grinold & Kahn, 2000). We introduce the details of these metrics in Appendix A.3.

*Formulaic alpha factor mining* (FAFM) is the process of discovering new factor formulas that provide predictive power. Effective factors are expected to achieve high correlation with future returns, exhibit stability across time and markets, and remain interpretable for risk management and regulatory compliance. In this regard, expanding the library of alpha signals through FAFM is essential for maintaining portfolio performance, particularly as older factors decay due to market adaptation (McLean & Pontiff, 2016). Historically, factor discovery relied on human experts who designed formulas based on financial intuition and iterative backtesting. While this approach produced many classic factors (e.g., momentum, value, quality) such as factor pools like Alpha101 (Kakushadze,

---

[*]Indicates corresponding authors.

2016) or Alpha158 (introduced in Appendix A.2), these handcrafted factors are limited by humans' prior knowledge and cannot quickly adapt to the market situation after their deployment. More recent efforts have been devoted to using machine learning to search alpha factors in a larger formula space automatically, such as reinforcement learning (Yu et al., 2023), genetic programming (Zhang et al., 2020), and symbolic regression (Shi et al., 2025a). They have uncovered novel signals but often require substantial engineering and computational resources.

The emergence of large language models (LLMs) introduces a new paradigm for FAFM. LLMs excel at symbolic reasoning, code generation, and formula synthesis (Naveed et al., 2025), making them appropriate choices for automated factor design. Recent studies show that prompting LLMs can yield interpretable alpha formulas with meaningful economic intuition (Wang et al., 2023; Li et al., 2023; Yuan et al., 2024; Luo et al., 2026; Cao et al., 2025; Shi et al., 2025b; Tang et al., 2025). Figure 1 illustrates the general workflow of applying LLMs to alpha factor mining. In this pipeline, carefully designed prompts are used to instruct LLMs to generate new factors. These factors are then evaluated through backtesting. We repeat such process to refine the alpha factors obtained using different search algorithms. Finally, the most promising factors are selected and applied in actual investment strategies. In principle, LLMs provide a flexible, scalable, and low-cost alternative to traditional methods, indicating their broad potential in FAFM.

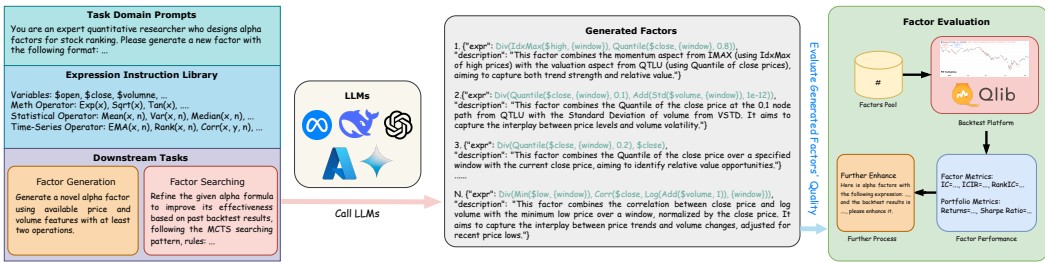

Figure 1: This workflow illustrates how large language models (LLMs) generate and refine alpha factors using task-specific prompts and an expression instruction library. Downstream tasks include zero-shot factor generation and guided factor searching. The generated factors are evaluated via a backtest engine (e.g., Qlib) using statistical metrics (IC, Rank IC, ICIR) and portfolio-level performance (e.g., Sharpe ratio), enabling further enhancement and selection of high-quality factors.

Despite the potential, significant challenges remain for employing LLMs to discover alpha factors. LLM-generated factors may embed biases, produce invalid or non-executable formulas, or lack robustness under market regime shifts. Furthermore, unlike established symbolic search frameworks, the behavior of LLMs in FAFM is still poorly understood, with no standardized benchmark to measure their performance. By contrast, the broader AI community has already developed rigorous benchmarks for domains such as code generation, mathematical reasoning, and scientific discovery. Yet no such benchmark currently exists for FAFM. In addition, in large-scale searching tasks under complex frameworks, it is crucial to understand how different LLM configurations influence performance and to identify more efficient strategies in prompt design and model selection.

To narrow down these gaps, we introduce **AlphaBench** in this work. To the best of our knowledge, it is the first benchmark dedicated to systematically evaluating LLMs in FAFM. Our contributions include: (1) we formally define and provide a unified perspective on the role of LLMs in FAFM; (2) we design and evaluate multiple tasks with diverse metrics under the FAFM setting, thereby uncovering the strengths and limitations of different LLMs across different dimensions; and (3) we further analyze how variations in LLM configurations, such as model type, reasoning paradigm, and prompting strategy—affect performance in FAFM; (4) we present empirical findings, highlight key limitations, and discuss open challenges for integrating LLMs into real-world investment workflows.

## 2 RELATED WORKS

**Formulaic Alpha Factor Mining.** Early research in formulaic alpha factor mining treated factors as symbolic expression trees evolved through genetic programming. For instance, Zhang et al. (2020) proposed *AutoAlpha*, which accelerates evolutionary search by introducing hierarchical mutation and

low-depth seeding, thereby improving both search speed and diversity. Subsequent works moved beyond purely random mutation to incorporate goal-directed exploration powered by reinforcement learning. Representative examples include *AlphaGen* (Yu et al., 2023), which directly optimises the portfolio-level Sharpe ratio across a set of factors, and *QuantFactor REINFORCE* (Zhao et al., 2025), which employs variance-bounded policy gradients to stabilise factor updates.

More recently, large language models (LLMs) have been introduced to enhance this field. Li et al. (2024b) developed *FAMA*, which addresses the issues of homogeneous outputs via cross-sample selection and a chain-of-experience memory. Wang et al. (2024b) proposed *QuantAgent*, a system that incorporates a self-improving generation–critique loop grounded in an internal knowledge base. Tang et al. (2025) introduced *AlphaAgent*, which regularises exploration to reduce alpha decay and enforce novelty. For algorithm-based searching, Luo et al. (2026) proposed a framework to apply LLMs in evolutionary algorithms to perform factor searching. In parallel, Shi et al. (2025b) formulated an LLM-powered Monte Carlo Tree Search (MCTS) framework, where partial formulas are treated as tree nodes, their expansions and rollouts are guided by back-test feedback. Complementary to these designs where we use LLM-guided automatic search algorithms, the Alpha-GPT series (Wang et al., 2023; Yuan et al., 2024) emphasise human-in-the-loop interaction. Beyond generation, Ding et al. (2025) introduced a novel framework for evaluating factors generated by LLMs.

**Generative and Financial Domain Benchmarks of LLM.** The rapid advancement of large language models (LLMs) has motivated a wide range of benchmark studies probing their capabilities in diverse domains such as logical reasoning, medicine, ethics, and education (Chang et al., 2023). While these efforts provide broad coverage, most benchmarks assess models primarily through free-form text generation, rather than evaluating their ability to produce *executable artefacts*. By contrast, formulaic alpha factor mining (FAFM) can be naturally framed as a *code-generation* problem, where an LLM must generate executable expressions that represent alpha factors. Several surveys and benchmarks have examined LLMs in unrestricted programming tasks (Liu et al., 2023a; Wang & Chen, 2023; Zhong & Wang, 2024), and more recent efforts target automated workflows in specialised domains, including data science (Zhang et al., 2025), website development (Tóth et al., 2024), and mobile agents (Deng et al., 2024). Parallel lines of research investigate benchmarks for natural language translation into domain-specific languages (DSLs). For example, Text-to-SQL tasks (Gao et al., 2023; Li et al., 2023; Lei et al., 2024) test query synthesis under semantic and execution constraints, while VerilogEval (Thakur et al., 2023; Liu et al., 2023b) evaluates LLMs in hardware-description generation with strict syntactic requirements. EvoCodeBench (Li et al., 2024a) further integrates multiple DSL tasks into a unified benchmark to enable systematic comparison.

Within the financial domain, benchmark design has primarily focused on numerical reasoning and question answering. Representative examples include FinQA (Chen et al., 2021), ConvFinQA (Chen et al., 2022), and FinTextQA (Chen et al., 2024). More comprehensive benchmarks such as FinBen (Xie et al., 2024) span 42 datasets covering information extraction, forecasting, and decision-making, while Pixiu (Xie et al., 2023) contributes instruction-tuning data, a domain-specific model (FinMA), and an eight-task benchmark mixing NLP and prediction. However, none of these efforts evaluate an LLM's ability to generate *executable alpha-factor DSL expressions*. To fill this gap, we introduce the first benchmark tailored for FAFM, bridging program-synthesis evaluation and financial AI.

## 3 OVERVIEW OF ALPHABENCH

### 3.1 TASKS DESIGN

AlphaBench is designed to evaluate LLMs in FAFM through three main tasks: **Factor Generation**, **Factor Evaluation**, and **Factor Searching**. Each task represents a critical stage in the alpha factor lifecycle, decomposing into subtasks that test the reasoning and editing capabilities of different LLMs and their settings. Overall, AlphaBench consists of 687 generation instructions and 1170 evaluation instructions. For factor searching, it contains 3 searching algorithms with 27 searching instructions. We defer more configuration details to Appendix B.

**Factor Generation**: In this task, LLMs transform natural language descriptions into candidate alpha factors. As shown in Figure 2, it includes the following subtasks; the instruction designs are provided in Appendix B.1.1:

- *Text2Alpha Generation*: The model receives a broad description, such as *momentum* or *mean reversion*, and translates it into a complete, formulaic expression using allowed variables and operators. This subtask tests whether the model can correctly interpret financial concepts and express them without relying on prior examples.

- *Directional Mining*: The model is provided with a specific theme, such as volatility-based or volume-driven signals, and produce a set of different factors within that theme. This evaluates both its ability to follow constraints and its creativity in generating diverse expressions for related market behaviors.

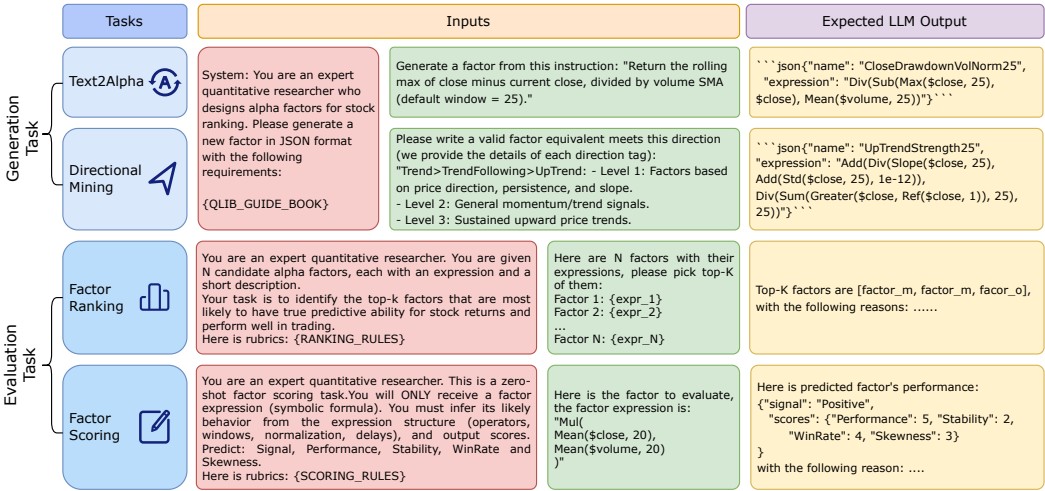

Figure 2: (1) *Generation Tasks*, where LLMs produce new factors from natural language or directional prompts (e.g., Text2Alpha, Directional Mining); and (2) *Evaluation Tasks*, where the generated factors are assessed through ranking or zero-shot scoring.

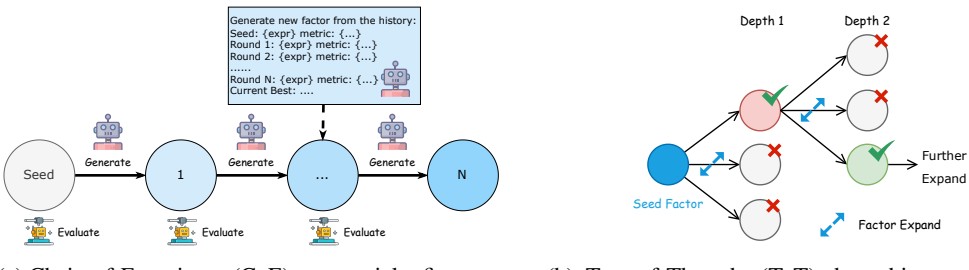

(a) Chain-of-Experience (CoE): sequential refinement from a seed factor.

(b) Tree-of-Thought (ToT): branching exploration with pruning.

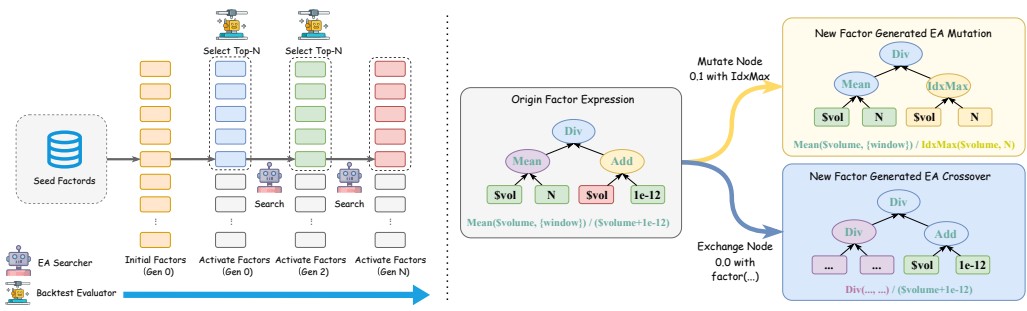

(c) Evolutionary Algorithm (EA): population-based iteration via mutation and crossover.

Figure 3: Illustrations of three LLM-driven search paradigms: (a) CoE, (b) ToT, and (c) EA.

**Factor Evaluation**: The backtesting step in factor mining is time and computation-intensive (Wang et al., 2024a), which motivates us to measure the ability of LLMs to act as intelligent evaluators

or "judges" in FAFM tasks, predicting the potential quality of a factor without full backtesting and thereby accelerating the process. In real-world applications, backtesting every candidate factor can be computationally expensive or infeasible, especially under short-term events or during early-stage idea screening. LLMs, therefore, offer a promising alternative by analyzing a factor's structure, operators, and economic intuition without full backtesting. Specifically, we evaluate two complementary subtasks below (see Figure 2 for an overview), with further details of the task design available in Appendix B.2.

- *Ranking*: The model is given a pool of candidate factors and asked to select the best-performed $k$ factors, simulating a filtering step in search-based pipelines. For this subtask, we report precision@$k$ and rank correlation with ground-truth results based on backtesting.

- *Scoring*: The model assigns an absolute score to a single factor, such as an estimated information coefficient (IC), Sharpe ratio, or a qualitative rating.

**Factor Searching**: This task evaluates the ability of LLMs to conduct iterative search over the combinatorial space of factors, rather than generating one-shot candidates. In practice, factor mining requires balancing wide exploration with efficient convergence to robust signals, so it often uses chain searching, tree-based expansion, or multi-turn refinement loops such as evolutionary algorithms. As shown in Figure 3, we benchmark performance under three representative search paradigms in AlphaBench, broadly covering the main approaches used in practice: (1) chain-based refinement (Li et al., 2023; Cao et al., 2025); (2) tree-structured exploration (Shi et al., 2025b); and (3) population-based evolutionary algorithms (Luo et al., 2026). The detailed designs of each algorithm are provided in Appendix B.3, while case studies of the search processes are presented in Appendix D.3.

For the searching task, we focus on two axes: (1) Search Cost. Since LLMs cannot reliably generate all required valid factors in a single step, we examine the token usage associated with successful generations. Specifically, we measure the cost in terms of the number of rounds needed for one valid generation, as well as the reliability of the outputs (i.e., the valid rate of generated factors). (2) Search Quality. This axis evaluates the effectiveness of the search. For individual search methods such as Chain-of-Experience (CoE) and Tree-of-Thought (ToT), we compare the final performance of discovered factors against the initial seed factor. For evolutionary algorithms (EA), we assess the overall performance of the factor population. Details of these metrics are provided in Appendix C.3.

## 3.2 LLMs Settings and Costs

To understand how FAFM's performance changes with different LLMs and settings, we conducted ablation studies on two key variables: the LLM and the prompting method. For our main experiments, we evaluate two groups of models: **commercial models**, including the Gemini family (Team et al., 2023; Reid et al., 2024) (Gemini-2.5-Pro, Gemini-2.5-Flash, Gemini-1.5-Flash-8B) and the GPT family (OpenAI, 2023) (GPT-4.1-Mini, GPT-5); and **open-source models**, including **DeepSeek-R1-Distill-Qwen-32B** (DeepSeek-AI et al., 2025a), **DeepSeek-V3** (DeepSeek-AI et al., 2025b), **LLaMA3.1-70B-Instruct** and **LLaMA3.1-8B-Instruct** (Meta AI, 2024), and **Qwen-2.5-14B-Instruct** (Qwen et al., 2025). This selection covers a broad spectrum of parameter scales and model families, ensuring that our benchmark reflects both state-of-the-art proprietary systems and the strongest available open-source alternatives.

Table 1: Token statistics combining generation/evaluation (top) and searching algorithms (bottom).

| Task / Algorithm | Prompt (K) | Completion (K) | Total (K) |
|---|---|---|---|
| Generate Vanilla | 944.5 | 75.5 | 1020.0 |
| Generate CoT | 1187.1 | 165.2 | 1352.3 |
| Evaluate Vanilla | 731.0 | 25.1 | 756.1 |
| Evaluate CoT | 1283.8 | 293.1 | 1576.9 |
| **Algorithm Results** | | | |
| CoE | 378 | 22 | 401 |
| ToT | 164 | 26 | 190 |
| EA | 167 | 25 | 193 |

For both generation and evaluation tasks, we compared a vanilla prompting approach against Chain-of-Thought (CoT), which is known to improve complex reasoning by forcing the model to articulate its steps. This helps us assess if FAFM's performance gains are due to the LLM's inherent capabilities or are enhanced by a more structured, step-by-step reasoning process.

For the searching task, we also evaluated the impact of different temperature and search count settings. This was done to determine how output randomness and the number of generated candidates, respectively, influence the efficiency and quality of the search results.

Finally, to estimate the token consumption and the corresponding costs, we provide a case study on **DeepSeek-V3** as a representative commercial LLM. Across all tasks in AlphaBench, the total token usage is about *5.5M*, including *4.2M* tokens in different prompts. The detailed breakdown for different tasks, and the comparison between whether or not to use chain of thought (CoT), are shown in Table 1. Notably, for models supporting prompt caching, we observed a cache hit ratio of **85%**, which significantly reduces repeated costs.

## 4 EVALUATION AND RESULTS

In AlphaBench, we control LLMs to generate formulaic alpha factor expressions in Qlib-compatible format, enabling direct execution and evaluation within the Qlib backtesting framework (Yang et al., 2020). In search task, we use **Alpha158** in Qlib as the initial factor pool and daily stock data for the CSI300 index constituents from 2020 to 2025 as our real market datasets, covering a diverse set of market conditions including bull, bear, and sideways regimes. The dataset includes standard features such as open, high, low, close prices, and volumes, from which factors are computed and evaluated.

### 4.1 METRICS

AlphaBench evaluates LLM performance across three dimensions, using a tailored set of metrics for each task. The core goal is to determine if an LLM can function as a reliable and efficient research partner, automating key parts of the factor discovery process. Metrics for each task are as follows: Generation task is measured by Reliability (produces valid code), Stability (delivers consistent output), and Accuracy (matches the user's intent); Evaluation task is assessed by how well the model acts as a "judge." We measure its ability to Rank (picking the best factors from a group) and Score (assigning a quality rating to a single factor); Searching task is evaluated on Cost (how many tokens and steps it takes to find a factor) and Quality (how much the new factor improves performance compared to the original).

For further details of the mentioned metrics, please refer to Appendix C. As Figure 4 shows, the overall performance of representative models varies across tasks. In the left chart, models are evaluated on **Generation** (Reliability, Stability, Accuracy) and **Evaluation** (Scoring, Ranking). In the right chart, their **Searching** performance is assessed based on metrics including Diversity, Cost, Reliability, ImprovedRate, and ImprovedRatio. While most models achieve strong reliability and competitive search performance, the results reveal larger gaps in accuracy and scoring. This highlights that even the better-performing models face significant challenges in the factor evaluation task.

### 4.2 FACTOR GENERATION

Table 3 summarizes the overall performance of all evaluated LLMs, in both vanilla and CoT settings, across three dimensions: reliability, stability, and accuracy. It is clear that larger commercial models outperform smaller open-source ones. Under the vanilla setting, GPT-5 attains the best overall score (0.72), while Gemini-2.5-Flash

Table 2: Generation task performance for coder-specific LLMs: Reliability, Stability, Accuracy, and Overall (average of the three) scores, under vanilla prompting.

| Model | Reliability | Stability | Accuracy | Overall |
|---|---|---|---|---|
| Qwen2.5-Coder-7B-Instruct | 0.76 | 0.49 | 0.30 | 0.52 |
| CodeLlama-70B-Instruct | 0.95 | 0.47 | 0.41 | 0.61 |
| Qwen3-Coder-480B-A35B-Turbo | 0.96 | 0.56 | 0.43 | 0.65 |

is competitive; CoT brings only marginal gains (e.g., a slight accuracy uptick for Flash) and sometimes reduces stability. Lighter open-source models (e.g., DeepSeek-R1-Distill-Qwen-32B, LLaMA3.1-8B) achieve substantially lower overall scores.

Detailed results in the subtasks of *Text2Alpha* and *Directional Mining* are provided in Table 23 and 25 in the appendix. In each subtasks, we divide the test cases into three categories named easy, medium and hard, indicating different levels of difficulties. The results for both sub-tasks indicate that the LLMs can achieve high reliability for all test instances, but their accuracy decreases significantly for

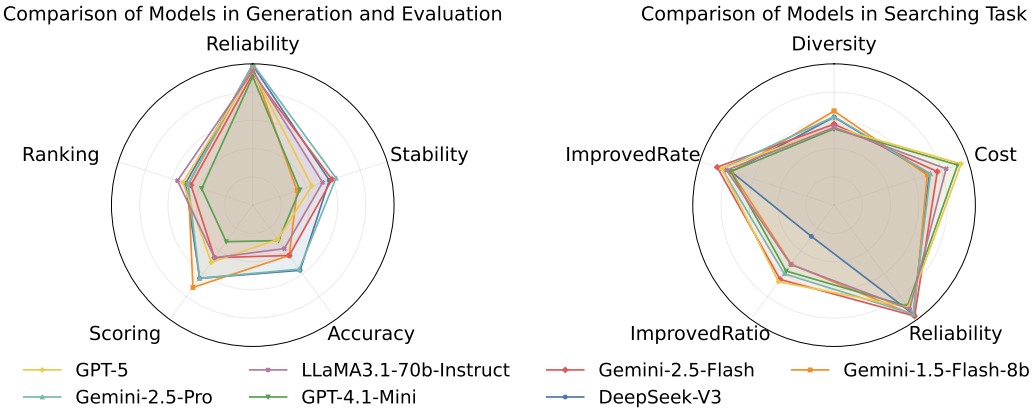

Figure 4: Radar chart comparison of model performance in AlphaBench. (Left) Models were assessed on the *generation task* across reliability, stability, and accuracy dimensions; The *evaluation task* was assessed based on scoring and ranking dimensions. (Right) The *searching task* was assessed based on diversity, cost, reliability, improvement rate, and improvement ratio dimensions.

hard instances. That is to say, models can reliably produce syntactically valid factors, but aligning the generated expressions with intended semantics under challenging instructions remains the core bottleneck, even for the best commercial LLMs.

Table 3: Generation task performance for general-purpose LLMs: Reliability, Stability, Accuracy, and Overall (average of the three) scores, under vanilla and CoT prompting.

| Model | Reliability | | Stability | | Accuracy | | Overall | |
|---|---|---|---|---|---|---|---|---|
| | Vanilla | CoT | Vanilla | CoT | Vanilla | CoT | Vanilla | CoT |
| GPT-5 | **1.00** | - | **0.62** | - | 0.56 | - | **0.72** | - |
| Gemini-2.5-Pro | 0.98 | - | 0.33 | - | 0.44 | - | 0.58 | - |
| Gemini-2.5-Flash | 0.99 | **0.99** | 0.57 | 0.49 | **0.57** | **0.58** | 0.71 | **0.69** |
| GPT-4.1-Mini | 0.93 | 0.93 | 0.59 | 0.41 | 0.44 | 0.44 | 0.65 | 0.59 |
| DeepSeek-V3 | 0.91 | 0.97 | 0.35 | 0.32 | 0.31 | 0.32 | 0.52 | 0.53 |
| LLaMA3.1-70b-Instruct | 0.95 | 0.94 | 0.52 | 0.44 | 0.38 | 0.47 | 0.62 | 0.61 |
| DeepSeek-R1-Distill-Qwen-32B | 0.35 | 0.58 | 0.19 | 0.24 | 0.14 | 0.14 | 0.23 | 0.32 |
| Qwen2.5-14B-Instruct | 0.79 | 0.58 | 0.50 | **0.51** | 0.34 | 0.46 | 0.54 | 0.52 |
| LLaMA3.1-8b-Instruct | 0.94 | 0.84 | 0.32 | 0.44 | 0.18 | 0.24 | 0.48 | 0.51 |
| Gemini-1.5-Flash-8b | 0.95 | 0.94 | 0.44 | 0.47 | 0.30 | 0.32 | 0.56 | 0.58 |

The overall results in Table 2, together with the details in Table 24 and Table 26, show that larger coder-specific LLMs consistently outperform smaller ones. Among them, Qwen3-Coder-480B-A35B-Turbo leads the group, achieving performance comparable to GPT-4.1-Mini. While these models show high reliability, they have noticeable gaps in stability and accuracy when compared to general-purpose LLMs of similar size. This indicates that despite scaling, coder-oriented LLMs struggle to generate consistently correct factors.

## 4.3 FACTOR EVALUATION

As shown in Table 4, the performance of almost all LLMs on the evaluation task is surprisingly poor. Unlike generation tasks, this task requires models to judge the quality of alpha factors by ranking and scoring them in a zero-shot manner, i.e., without utilizing other information like backtesting. The results suggest that LLMs struggle to serve as reliable evaluators of alpha performance. Specifically, no model achieves consistently strong performance in both ranking and scoring metrics. GPT-5 and Gemini-2.5-Pro show moderate results, but their overall scores remain below 0.40. In addition, CoT does not boost the performance in general in this task. Some performance metrics like precision even drops with CoT. Finally, open-source models such as DeepSeek-V3 and Qwen2.5-14B-Instruct

perform notably worse, highlighting their limited ability to capture the nuanced semantics of alpha factor evaluation. By contrast, Gemini-2.5-Flash, despite its moderate size, achieves competitive Signal Accuracy and maintains balanced performance across metrics. This highlights the diminishing returns of increasing model size and knowledge capacity for complex evaluation tasks.

In order to analyze LLM behaviors in evaluation more precisely, we decompose the problem into two atomic tasks, and construction details are provided in Appendix B.2.4:

- *Signal Classification*: Given a factor expression, the model analyzes its components and determines whether the factor represents a meaningful signal or noise.

- *Pairwise Selection*: The model is presented with two candidate factors and must compare them, selecting the one expected to perform better.

Table 4: Evaluation of factor Ranking (Precision) and Scoring (Signal ACC, MAE). Each cell shows *vanilla / CoT*, averaged over Neutral, Bear, and Bull scenarios. Bold marks the best per metric (higher is better for Precision/ND-CG/ACC/Overall; lower is better for MAE). Overall is the mean of all metrics, with MAE normalized by $1 - \text{MAE}/5$.

| Model | Ranking | Scoring | | Overall |
|---|---|---|---|---|
| | Precision | Signal ACC | MAE | |
| GPT-5 | 0.24 / - | 0.32 / - | 1.67 / - | 0.47 / - |
| Gemini-2.5-Pro | 0.24 / - | **0.36** / - | 1.66 / - | **0.48** / - |
| Gemini-2.5-Flash | 0.25 / 0.14 | 0.32 / **0.33** | 1.67 / 1.64 | 0.47 / 0.44 |
| GPT-4.1-Mini | 0.23 / 0.24 | 0.23 / 0.20 | 1.57 / 1.59 | 0.44 / 0.43 |
| DeepSeek-V3 | 0.19 / 0.17 | 0.16 / 0.17 | 1.60 / 1.57 | 0.40 / 0.40 |
| LLaMA3.1-70b-Instruct | **0.28** / 0.26 | 0.23 / 0.24 | 1.62 / 1.61 | 0.45 / 0.45 |
| DeepSeek-R1-Distill-Qwen-32B | 0.20 / 0.20 | 0.24 / 0.23 | 1.59 / 1.56 | 0.43 / 0.43 |
| Qwen2.5-14B-Instruct | 0.25 / 0.24 | 0.28 / 0.26 | 1.65 / 1.62 | 0.46 / 0.45 |
| Gemini-1.5-Flash-8b | 0.26 / **0.27** | 0.25 / 0.26 | **1.54 / 1.54** | 0.46 / **0.47** |
| LLaMA3.1-8b-Instruct | 0.26 / 0.26 | 0.26 / 0.26 | 1.67 / 1.58 | 0.45 / 0.46 |

Table 5: Vanilla vs. CoT accuracy across CSI300 and SP500 on atomic task. **Up**: *Signal Classification*; **Down**: *Pairwise Selection*.

| Model | CSI300 | | SP500 | |
|---|---|---|---|---|
| | Vanilla | CoT | Vanilla | CoT |
| DeepSeek-V3 | 0.46 | 0.40 | 0.51 | 0.51 |
| Gemini-2.5 Flash | 0.39 | 0.37 | 0.55 | 0.52 |
| Gemini-2.5 Pro | 0.44 | 0.42 | 0.47 | 0.50 |
| GPT-4.1 Mini | 0.41 | 0.39 | 0.49 | 0.50 |
| GPT-5 | 0.41 | 0.42 | 0.57 | 0.57 |
| DeepSeek-V3 | 0.48 | 0.40 | 0.53 | 0.48 |
| Gemini-2.5 Flash | 0.40 | 0.46 | 0.48 | 0.48 |
| Gemini-2.5 Pro | 0.40 | 0.52 | 0.46 | 0.48 |
| GPT-4.1 Mini | 0.44 | 0.48 | 0.44 | 0.44 |
| GPT-5 | 0.64 | 0.66 | 0.52 | 0.55 |

We report the accuracy in Table 5, and details metrics are provided in Table 30 and Table 31, our new experiments confirm that LLMs remain weak performers on both atomic tasks *Signal Classification* and *Pairwise Selection*, with accuracy generally close to random guessing and no consistent benefit from CoT reasoning under zero-shot testing.

The results of supervised fine-tuning (SFT) on GPT-4.1-Mini are provided in Table 6 and Table 32. With only a small amount of labeled data, in-market fine-tuning substantially boosts performance, particularly on the *Pairwise Selection* task, while offering limited gains for *Signal Classification*. The latter degradation is likely caused by strong overfitting, as the model tends to collapse toward predicting a single label of noise. We also find that experience learned on CSI300 generalizes better: a model fine-tuned on CSI300 still improves performance when evaluated on SP500 in *Pairwise Selection*. This may due to differences in market structure. The Chinese market

Table 6: Accuracy across markets and fine-tuning settings for atomic tasks.

| Market | Model | Classification | | Selection | |
|---|---|---|---|---|---|
| | | Vanilla | CoT | Vanilla | CoT |
| CSI300 | Origin | 0.39 | 0.41 | 0.48 | 0.44 |
| | FT on CSI300 | 0.48 | 0.50 | 0.83 | 0.86 |
| | FT on SP500 | 0.51 | 0.47 | 0.50 | 0.50 |
| SP500 | Origin | 0.50 | 0.49 | 0.44 | 0.44 |
| | FT on CSI300 | 0.53 | 0.52 | 0.64 | 0.64 |
| | FT on SP500 | 0.49 | 0.51 | 0.78 | 0.78 |

has stricter trading rules, more frictions, daily price limits, and stronger retail-driven volatility. These constraints create clearer and more frequent "noise-like" patterns in factor behaviors, giving the model a wider variety of structural cases to learn from.

## 4.4 SEARCHING TASK

We summarize the results in two dimensions in Table 7, which reveals a clear trade-off between search quality and cost-efficiency. The detailed performance of each algorithm is reported in Tables 33, 34, and 35. While models such as Gemini-2.5-Pro delivers the strongest raw performance, they are less economical in token usage, whereas GPT-5 strikes a more balanced outcome by combining high search effectiveness with superior cost-efficiency. Mid-sized models, including Gemini-2.5-Flash,

LLaMA-3.1-70B, and GPT-4.1-Mini, occupy a middle ground, providing usable performance at moderate costs but without leading in either dimension.

In contrast, smaller or less capable open-source models, such as DeepSeek-V3, lag notably behind, highlighting the gap between frontier commercial systems and lighter alternatives in real-world factor searching. We further analyze the parameter settings in searching, focusing on the temperature of LLMs (Figure 14) and the capacity of EA (Figure 15). The results indicate two main findings: (1) temperature exhibits a clear exploration–exploitation trade-off, where a

Table 7: Comparison of LLMs on factor searching, showing the trade-off between search quality and cost. Each score is the mean of detailed metrics, scaled to [0,1]

| Model | Search Quality | Search Cost |
|---|---|---|
| DeepSeek-V3 | 0.494 | 0.800 |
| Gemini-1.5-Flash-8b | 0.622 | 0.802 |
| Gemini-2.5-Flash | 0.646 | 0.850 |
| Gemini-2.5-Pro | 0.632 | 0.808 |
| GPT-4.1-Mini | 0.608 | 0.904 |
| GPT-5 | 0.656 | 0.940 |
| LLaMA3.1-70b-Instruct | 0.624 | 0.850 |

higher temperature (1.5) improves diversity and sometimes performance but with higher cost and lower stability, while a lower temperature (0.75) is more efficient and reliable, though at the expense of slightly reduced diversity; (2) EA capacity improves performance only up to a point and then saturates, with generating 20 candidates per round (EA-20) providing the best balance between cost, diversity, and improvement, whereas increasing the capacity to 30 candidates (EA-30) often increases cost without proportional gains.

## 5 ANALYSIS AND DISCUSSION

This section analyzes the performance of Large Language Models (LLMs) in Formulaic Alpha Factor Mining (FAFM), summarizing the key findings from our benchmark. We first assess the core capabilities of LLMs in generation and evaluation, followed by an analysis of prompting strategies and practical recommendations for factor searching.

**Factor Generation**. In alpha generation, LLMs generally perform well as translators, converting natural-language descriptions of factors into formulaic expressions or generating factors aligned with directional guidance. Most models achieve reliability above 80% in all difficulty levels. Across parameter sizes, LLMs consistently function as competent generators; however, the accuracy of the produced factors varies depending on model family, parameter size, and instruction complexity. Larger commercial models tend to generate more accurate results, and this performance gap becomes more pronounced as instructions grow more complex. Nevertheless, when instructions are extremely complicated, even large models fail to handle them, revealing limits in current generative capacity. For directional mining, performance differences between models are more pronounced. We observe that the distilled model exhibits the weakest performance, suggesting that the distillation process may have compromised its effectiveness for this specialized task. Detailed analysis suggests that these gaps stem partly from differences in the knowledge domains captured by each LLM, which affect their ability to produce meaningful directional signals.

**Factor Evaluation**. We observe consistently weak zero-shot performance on ranking and scoring, with two main causes: (1) Alpha mining is highly niche and lacks paired training data (factor expression, performance label). Existing labels from backtests (IC/RankIC, Sharpe) are rarely public and strongly regime-dependent, leaving general-purpose LLMs without supervision to map formulas to predictive value. (2) Our setting under-specifies context: plain-text expressions omit execution details (rebalance, lag, normalization, regime) and obscure operator precedence, windowing, and composition, making performance inference ill-posed. These issues explain the brittleness of CoT and point to possible remedies: (i) structured representations (e.g., ASTs/operator graphs), (ii) minimal execution metadata, and (iii) weak-label alignment via automated backtests and pairwise ranking. However, we provide a potential path forward to mitigate this challenge, we show that supervised fine-tuning (SFT) can help in specific tasks. Typically, the factor-searching process (e.g., our searching suite) naturally generates large numbers of paired factor expressions and their corresponding performance metrics. This provides a rich source of supervised data that can be used to construct larger datasets for future SFT efforts.

**Effect of Chain-of-Thought (CoT).** The effect of CoT prompting is overall limited. It provides modest improvements for smaller models (e.g., Qwen2.5-15B, LLaMA3.1-8B), but often reduces performance for larger ones, likely because longer reasoning chains increase variance or introduce unnecessary complexity. In evaluation tasks, CoT shows almost no benefit, as models already perform poorly at factor assessment. This pattern suggests that while LLMs may not need CoT for simple tasks, overly long reasoning can harm harder tasks, pointing to a trade-off between reasoning depth and reliability as well as the potential for more refined CoT designs.

**Suggestion in Searching Task**. Based on our findings, we propose several directions for future work in this field. (1) In terms of cost–performance trade-offs, lightweight commercial models such as the Gemini-Flash series or GPT-Mini series can serve as practical executors for factor generation or mining. These models demonstrate strong economic efficiency in large-scale searches while achieving performance comparable to their larger counterparts. (2) For the choice of search algorithm, population-based approaches such as evolutionary algorithms (EA) appear more advantageous than individual search methods, as they better exploit the exploration capacity of LLMs. (3) We find that vanilla prompt designs are generally sufficient for searching tasks, without requiring more complex CoT-style formats. (4) A complete mechanism is still needed to verify the validity of LLM-generated factors. Since current evaluation ability is limited, most existing work continues to rely heavily on backtesting results. To improve search efficiency, future studies should explore new evaluation metrics or apply task-specific fine-tuning to enhance evaluation capability.

**Beyond Quant: A Portable Template for Executable-DSL Reasoning.** AlphaBench offers an evaluation pipeline built on an executable DSL and task-aligned objective metrics, not limited to equity factors. Many domains rely on formula-like expressions similar to alpha factors. By swapping the factor language for another DSL, AlphaBench immediately targets: (i) programmatic feature design and symbolic regression task (Udrescu & Tegmark, 2020; de Franca et al., 2024); (ii) constraint-solving and planning DSLs (MiniZinc (Nethercote et al., 2007), PDDL (Fox & Long, 2003), Temporal Logics (Chen et al., 2023)), evaluated by satisfiability, objective values, and regret; (iii) experiment-metric design in A/B testing, where "metric formulas" run on log data and are scored for stability, variance, and bias sensitivity (Deng & Shi, 2016). The evaluation suite and metrics are able to transfer directly, while the searching modules (CoE, ToT, and EA) serve as general-purpose search controllers for any executable-DSL space.

## 6 CONCLUSION

In this work, we introduced AlphaBench, the first benchmark to comprehensively evaluate LLMs in formulaic alpha factor mining (FAFM). Our study shows that LLMs are reliable generators, with most models achieving high validity in translating descriptions into factor expressions and handling directional prompts, though accuracy depends heavily on model size and complexity of instructions. In contrast, factor evaluation remains the weakest link: models consistently fail to assess factor quality due to the lack of supervised training data and missing execution context, underscoring the need for structured representations and weak-label alignment. We provided insights of the effect of supervised fine-tuning in different evaluation task. We also find that Chain-of-Thought (CoT) provides only limited gains for smaller models but sometimes harmful for larger ones, suggesting a trade-off between reasoning depth and reliability. For searching, LLMs can explore and refine factors effectively, but the efficiency, cost, and validation mechanisms remain open challenges. Overall, AlphaBench highlights both the promise and the current limitations of LLMs in FAFM, offering concrete directions for future work on robustness, evaluation alignment, and knowledge integration.

## 7 LIMITATIONS AND FUTURE WORK

This benchmark is the first systematic evaluation of LLMs in formulaic alpha factor mining, but has specific boundaries. We focus on a limited set of models, prompting methods, and daily equity factors, leaving out intraday, cross-asset, and more advanced techniques like fine-tuning hybrid search methods. Future work should extend benchmark to these broader areas. We also see opportunities to improve LLM-driven factor discovery by integrating more sophisticated methods, such as retrieval-augmented generation (RAG), to enhance output quality and directly connect the process to portfolio optimization.

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

## REPRODUCIBILITY STATEMENT

To facilitate the reproducibility of our work, we provide the following materials as supplementary resources: sample code, searching checkpoints, and datasets. These materials are made available for reference, enabling other researchers to replicate the key components of our study.

## USAGE OF LLM

In our workflow, large language models (LLMs) are primarily employed in two ways. First, LLMs are used to assist with **data visualization**. By generating clear descriptions, titles, and captions, the models help in producing figures and tables that are both accurate and accessible.

Second, LLMs support **writing refinement and formatting**. They are leveraged to polish drafts, improve sentence flow, and enforce consistency in style and structure. This includes aligning with academic writing conventions, standardizing terminology, and adapting the text to required formatting guidelines.

# A  PRELIMINARY

In this section, we provide the background necessary for understanding our benchmark. First, we introduce the basic settings of alpha factors in Alpha158, including the set of operators and the pool of seed factors. Next, we describe the fundamental properties of factors and the evaluation metrics used in AlphaBench.

## A.1  ALPHA FACTOR OPERATORS

The construction of formulaic alpha factors in Qlib relies on a well-defined library of mathematical, statistical, and logical operators. These operators provide the building blocks for composing factor expressions, ranging from elementary arithmetic to advanced rolling statistics and regression-based descriptors. Table 8 summarizes the supported operators in our framework.

In addition to the standard operators included in Qlib, we further extend the operator set to support enhanced flexibility in factor design. Notably, we implement additional mathematical transformations (**Sqrt**, **Exp**, **Tanh**, **Reciprocal**), advanced statistical controls (**Clip** for bounding values), and other specialized utilities for factor stabilization and normalization. These extensions are highlighted in bold in Table 8. By enriching the operator library, we enable a broader search space of factor formulations, thereby supporting more expressive signal construction and improving the robustness of subsequent factor mining experiments.

Table 8: Summary of supported operators for Alpha Mining. Extended operators are highlighted in **bold**.

| Type | Operator | Description / Definition |
|---|---|---|
| Arithmetic & Logical | Add(x, y) | Addition, return $x + y$ |
| | Sub(x, y) | Subtraction, return $x - y$ |
| | Mul(x, y) | Multiplication, return $x \times y$ |
| | Div(x, y) | Division, return $x \div y$ |
| | Power(x, y) | Power, return $x^y$ |
| | Log(x) | Natural logarithm, $\log(x)$ |
| | Abs(x) | Absolute value, $|x|$ |
| | Sign(x) | Sign function, $\{-1, 0, 1\}$ |
| | Delta(x, n) | Change, $x - Ref(x, n)$ |
| | And(x, y) | Logical AND |
| | Or(x, y) | Logical OR |
| | Not(x) | Logical NOT |
| Special Math | **Sqrt(x)** | **Square root,** $\sqrt{x}$ |
| | **Exp(x)** | **Exponential,** $e^x$ |
| | **Tanh(x)** | **Hyperbolic tangent,** $\tanh(x)$ |
| | **Reciprocal(x)** | **Reciprocal,** $1/x$ |
| Comparison | Greater(x, y) | Return larger of $(x, y)$ |
| | Less(x, y) | Return smaller of $(x, y)$ |
| | Gt(x, y) | Return True if $x > y$ |
| | Ge(x, y) | Return True if $x \geq y$ |
| | Lt(x, y) | Return True if $x < y$ |
| | Le(x, y) | Return True if $x \leq y$ |
| | Eq(x, y) | Return True if $x = y$ |
| | Ne(x, y) | Return True if $x \neq y$ |
| Rolling Statistics (window $n$) | Mean(x, n) | Rolling mean |
| | Std(x, n) | Rolling standard deviation |
| | Var(x, n) | Rolling variance |
| | Max(x, n) | Rolling maximum |

*(Continued on next page)*

*(Continued from previous page)*

| Type | Operator | Description / Definition |
|------|----------|--------------------------|
| | Min(x, n) | Rolling minimum |
| | Skew(x, n) | Rolling skewness |
| | Kurt(x, n) | Rolling kurtosis |
| | Sum(x, n) | Rolling sum |
| | Med(x, n) | Rolling median |
| | Mad(x, n) | Median absolute deviation |
| | Count(x, n) | Count of non-missing values |
| | EMA(x, n) | Exponential moving average |
| | WMA(x, n) | Weighted moving average |
| | Corr(x, y, n) | Rolling correlation |
| | Cov(x, y, n) | Rolling covariance |
| | **Clip(x, a, b)** | **Clip values into $[a, b]$; one-sided bounds supported** |
| Regression / Decomposition | Slope(x, n) | Rolling regression slope |
| | Rsquare(x, n) | Regression $R^2$ |
| | Resi(x, n) | Regression residual |
| Ranking / Quantile | Rank(x, n) | Rank within window $n$ |
| | Quantile(x, n) | Quantile position |
| Indexing / Conditional Logic | Ref(x, n) | Value $n$ steps ago |
| | IdxMax(x, n) | Index of maximum in window $n$ |
| | IdxMin(x, n) | Index of minimum in window $n$ |
| | If(cond, x, y) | Conditional: return $x$ if True else $y$ |
| | Mask(cond, x) | Keep $x$ where cond is True, else NaN |
| | Delay(x, n) | Lagged value, alias of Ref |

## A.2 ALPHA FACTOR LIBRARIES

We begin with the **Alpha158** library in Qlib, which is a widely adopted benchmark for quantitative factor research. The construction of Alpha158 follows a modular design, starting from a collection of *base factors* that are categorized into three groups:

- **KBar factors** (9 features): candlestick-based measures derived from daily OHLC data, such as relative mid-price (KMID), price spread ratios (KLEN, KSFT), and normalized high/low differences (KUP2, KLOW2).

- **Price-related factors** (5 features): lagged references of daily prices and trading volume, which encode short-term memory effects and relative deviations from the most recent observations.

- **Rolling time-series factors** (29 features): operators applied to a moving window of historical observations, including momentum-type measures (ROC), moving averages and volatility estimators (MA, STD, VSTD), regression-based descriptors (BETA, RSQR, RESI), cross-sectional correlation terms (CORR, CORD), and event counters (CNTP, CNTN, SUMP).

Altogether, these three categories yield **42 base factors**. The full Alpha158 library is obtained by expanding the rolling operators across multiple temporal horizons (e.g., 5, 10, 20, 30, and 60 days), which systematically increases the feature set from 42 base factors to **158 factors**. In this manner, Alpha158 captures a broad range of dynamics across both short- and long-term windows.

In our experimental setting, we adopt the **smallest rolling window size of 5 days** as the default configuration to construct the initial pool of *seed factors*. This choice ensures comparability across categories while providing sufficient granularity for subsequent factor search. It should also be noted that although Alpha158 originally includes several features based on the volume-weighted average price (VWAP), we exclude these in our study due to limitations in the available data. The definitions of the base factors are summarized in Table 9.

Table 9: Alpha158 factors (KBar, Price, and Rolling). Default expressions shown; Type visually merged by leaving subsequent rows blank.

| Type | Name | Default Expression |
|------|------|--------------------|
| KBar | KMID | (close - open) / open |
| | KLEN | (high - low) / open |
| | KMID2 | (close - open) / (high - low + 1e-12) |
| | KUP | (high - Greater(open, close)) / open |
| | KUP2 | (high - Greater(open, close)) / (high - low + 1e-12) |
| | KLOW | (Less(open, close) - low) / open |
| | KLOW2 | (Less(open, close) - low) / (high - low + 1e-12) |
| | KSFT | (2 * close - high - low) / open |
| | KSFT2 | (2 * close - high - low) / (high - low + 1e-12) |
| Price | OPEN_REF | Ref(open, 1) / close |
| | HIGH_REF | Ref(high, 1) / close |
| | LOW_REF | Ref(low, 1) / close |
| | VWAP_REF | Ref(vwap, 1) / close |
| | VOLUME_REF | Ref(volume, 1) / (volume + 1e-12) |
| Rolling | ROC | Ref(close, 5) / close |
| | MA | Mean(close, 5) / close |
| | STD | Std(close, 5) / close |
| | BETA | Slope(close, 5) / close |
| | RSQR | Rsquare(close, 5) |
| | RESI | Resi(close, 5) / close |
| | MAX | Max(high, 5) / close |
| | LOW | Min(low, 5) / close |
| | QTLU | Quantile(close, 5, 0.8) / close |
| | QTLD | Quantile(close, 5, 0.2) / close |
| | RANK_CLOSE | Rank(close, 5) |
| | RSV | (close - Min(low, 5)) / (Max(high, 5) - Min(low, 5) + 1e-12) |
| | IMAX | IdxMax(high, 5) / 5 |
| | IMIN | IdxMin(low, 5) / 5 |
| | IMXD | (IdxMax(high, 5) - IdxMin(low, 5)) / 5 |
| | CORR | Corr(close, Log(volume + 1), 5) |
| | CORD | Corr(close / Ref(close, 1), Log(volume / Ref(volume, 1) + 1), 5) |
| | CNTP | Mean(close > Ref(close, 1), 5) |
| | CNTN | Mean(close < Ref(close, 1), 5) |
| | CNTD | Mean(close > Ref(close, 1), 5) - Mean(close < Ref(close, 1), 5) |
| | SUMP | Sum(Greater(close - Ref(close, 1), 0), 5) / (Sum(Abs(close - Ref(close, 1)), 5) + 1e-12) |
| | SUMN | Sum(Greater(Ref(close, 1) - close, 0), 5) / (Sum(Abs(close - Ref(close, 1)), 5) + 1e-12) |
| | SUMD | (Sum(Greater(close - Ref(close, 1), 0), 5) - Sum(Greater(Ref(close, 1) - close, 0), 5)) / (Sum(Abs(close - Ref(close, 1)), 5) + 1e-12) |
| | VMA | Mean(volume, 5) / (volume + 1e-12) |
| | VSTD | Std(volume, 5) / (volume + 1e-12) |
| | WVMA | Std(Abs(close / Ref(close, 1) - 1) * volume, 5) / (Mean(Abs(close / Ref(close, 1) - 1) * volume, 5) + 1e-12) |
| | VSUMP | Sum(Greater(volume - Ref(volume, 1), 0), 5) / (Sum(Abs(volume - Ref(volume, 1)), 5) + 1e-12) |

*(Continued on next page)*

*(Continued from previous page)*

| Type | Name | Default Expression |
|---|---|---|
| | VSUMN | Sum(Greater(Ref(volume, 1) - volume, 0), 5) / (Sum(Abs(volume - Ref(volume, 1)), 5) + 1e-12) |
| | VSUMD | (Sum(Greater(volume - Ref(volume, 1), 0), 5) - Sum(Greater(Ref(volume, 1) - volume, 0), 5)) / (Sum(Abs(volume - Ref(volume, 1)), 5) + 1e-12) |

### A.3 METRICS OF FACTORS

To evaluate the predictive ability of an alpha factor, we adopt several well-established statistical metrics.

**(1) Information Coefficient (IC).** The IC measures the linear correlation between the factor values and the subsequent asset returns. For factor $f$ at time $t$, let $f_{i,t}$ be the factor score of asset $i$, and $r_{i,t+1}$ its realized return at the next period. The cross-sectional IC is

$$\text{IC}_t = \text{corr}(\{f_{i,t}\}_i, \{r_{i,t+1}\}_i),$$

where $\text{corr}(\cdot)$ denotes the Pearson correlation. The mean IC over time provides a direct measure of the factor's predictive power.

**(2) Rank Information Coefficient (RankIC).** The RankIC is the Spearman rank correlation between factor scores and future returns:

$$\text{RankIC}_t = \text{Spearman}(\{f_{i,t}\}_i, \{r_{i,t+1}\}_i).$$

RankIC is more robust to outliers and is often used in portfolio construction where only the ranking of assets matters. However, it has limitations in practice: - In the Chinese stock market, daily up/down limits truncate returns, making rank ordering unreliable. - In strong bull or bear markets, when most assets move in the same direction, rank differences become weak. Therefore, in AlphaBench, we primarily use IC rather than RankIC to measure factor performance.

**(3) Information Ratio (ICIR / RankICIR).** To evaluate the consistency of a factor's predictive ability, we normalize the mean IC by its volatility across time:

$$\text{ICIR} = \frac{\mathbb{E}[\text{IC}_t]}{\text{Std}[\text{IC}_t]}, \qquad \text{RankICIR} = \frac{\mathbb{E}[\text{RankIC}_t]}{\text{Std}[\text{RankIC}_t]}.$$

A higher ICIR indicates more stable and reliable signals.

**(4) Directional Properties.** Beyond the average correlation, we also analyze directional characteristics of factor signals:

- Win Rate. The fraction of periods in which the IC has the same sign as the expected direction of the factor:

$$\text{WinRate} = \frac{1}{T} \sum_{t=1}^{T} \mathbf{1}\{\text{IC}_t \cdot s > 0\},$$

  where $s \in \{+1, -1\}$ denotes the assumed signal direction.

- Skewness. The asymmetry of the IC distribution, indicating whether extreme positive or negative outcomes dominate:

$$\text{Skew} = \frac{\mathbb{E}\left[(\text{IC}_t - \bar{\text{IC}})^3\right]}{(\text{Std}[\text{IC}_t])^3}.$$

  A positive skew suggests occasional strong positive predictive power, while a negative skew indicates occasional sharp failures.

## A.4 DIVERSITY OF FACTORS

To quantify the diversity of generated factors, we measure similarity from two complementary perspectives: structural distance of expressions and output correlation of signals.

**(1) Structural Distance.** Each factor expression is parsed into an Abstract Syntax Tree (AST), where constants are discarded so that only variables and operators are preserved. For two factors $f_i$ and $f_j$, we compute their tree-edit distance using the Zhang–Shasha algorithm (Zhang & Shasha, 1989):

$$d_{\text{AST}}(f_i, f_j) = \text{TED}(\text{AST}(f_i), \text{AST}(f_j)),$$

where $\text{TED}(\cdot, \cdot)$ denotes the minimum number of edit operations (insert, delete, substitute) to transform one tree into the other.

We summarize the structural diversity within a population $\mathcal{F} = \{f_1, \ldots, f_n\}$ by

$$\bar{d}_{\text{AST}} = \frac{2}{n(n-1)} \sum_{1 \le i < j \le n} d_{\text{AST}}(f_i, f_j), \quad d_{\max} = \max_{i<j} d_{\text{AST}}(f_i, f_j),$$

and define the normalized diversity score as

$$D_{\text{AST}} = \begin{cases} \bar{d}_{\text{AST}}/d_{\max}, & d_{\max} > 0, \\ 0, & \text{otherwise.} \end{cases}$$

**(2) Output Correlation.** Structural distance does not always reflect behavioral difference: two seemingly different formulas may generate highly similar signals. Hence, we also measure similarity via the correlation of factor outputs. For two factors $f_i, f_j$, let $X_{f_i}, X_{f_j} \in \mathbb{R}^T$ be their time series outputs over a given horizon of length $T$. We compute the Pearson correlation:

$$\rho(f_i, f_j) = \text{corr}(X_{f_i}, X_{f_j}) \in [-1, 1].$$

The overall similarity of a factor set is quantified by applying the absolute value to the correlation, treating negative correlations as indicative of differences in signal direction. When two factors exhibit a high negative correlation, this is typically due to opposing signal directions, but we still regard them as similar in the context of factor diversity. In our approach, we focus exclusively on the diversity metric, which accounts for both positive and negative correlations, to measure the diversity of the factor set:

$$\bar{\rho}_{\text{abs}} = \frac{2}{n(n-1)} \sum_{i<j} |\rho(f_i, f_j)|.$$

We define the correlation-based diversity as,

$$D_{\text{corr}} = 1 - \bar{\rho}_{\text{abs}}.$$

where $\bar{\rho}_{\text{abs}}$ represents the average absolute value of the correlation between all pairs of factors in the set. Since the absolute correlation values $\rho_{\text{abs}}$ range from 0 to 1, a lower value of $\rho_{\text{abs}}$ indicates greater dissimilarity between factors, which corresponds to higher diversity. To convert this into a diversity score where higher values indicate more diversity, we apply $1 - \bar{\rho}_{\text{abs}}$, ensuring that a higher diversity score reflects lower correlation and thus greater diversity among the factors.

**(3) Diversity in Population.** Given a factor population, we compute all pairwise distances and correlations. The metrics $(\bar{d}_{\text{AST}}, d_{\max}, D_{\text{AST}})$ and $(\bar{\rho}, \bar{\rho}_{\text{abs}}, D_{\text{corr}})$ jointly describe how structurally and behaviorally diverse the population is. Intuitively:

- $D_{\text{AST}}$ captures expression-level variety (different formulas/operators).
- $D_{\text{corr}}$ captures signal-level variety (different outputs over time).

Together, they provide a balanced assessment of factor diversity during the search process.

A.5 FACTOR QUALITY CONTROL

For factors generated by LLMs, we design a filtering stage to ensure that only valid and practical candidates are kept for further evaluation. Since LLM outputs may occasionally deviate from the required format, we first enforce a structured JSON output schema (details and examples are provided in Appendix D). After passing this structural check, each factor expression undergoes a series of validation steps to guarantee correctness, efficiency, and usability.

Specifically, our factor filter contains the following components:

- **Expression validation.** We check whether parentheses are properly matched, whether any illegal characters appear, and whether all variables are valid. To avoid over-engineered or fragile signals, we reject expressions whose operator nesting depth exceeds five layers.

- **Operator validation.** We verify that only supported operators are used, and that each is called with the correct arguments. Factors using undefined or incorrectly-argued operators are discarded.

- **Efficiency validation.** To ensure factors can be practically applied to large universes, we measure their computational cost against the CSI300 index constituents. Factors requiring more than 30 seconds to compute over a two-week period are filtered out.

- **Output validation.** We examine the ratio of missing values (NaN) in the computed series. If a factor produces more than 1% NaNs across the CSI300 index universe over a two-week rolling window, it is rejected.

These rigorous checks ensure we maintain a curated pool of factors that are not only syntactically correct but also efficient and robust, making them suitable for large-scale backtesting.

# B ALPHABENCH CONSTRUCTION

This section describes how we build the evaluation scenarios used in AlphaBench. We first define the factor generation task, including the Text2Alpha setting with instruction variations and the directional factor mining workflow that couples tree-based edits and evolutionary operators with LLMs. We then present the factor evaluation task on CSI 300 data, where factors are assessed by ranking and scoring protocols under different market regimes. Finally, we detail the factor searching task, covering three paradigms—Chain-of-Experience, Tree-of-Thought, and Evolutionary Algorithm—and the associated metrics for efficiency, diversity, quality, and stability. Throughout, we emphasize reproducibility, clear prompts, validity checks, and traceable histories for every generated candidate.

## B.1 CONSTRUCTION OF FACTOR GENERATION TASK

### B.1.1 TEXT2ALPHA

The Text2Alpha task evaluates the zero-shot factor generation ability of large language models. In this setting, the model is given a natural language instruction that describes a desired signal, and it must output a valid factor expression in the Qlib-style DSL. To reflect different levels of difficulty, we organize the instructions into three groups: easy, medium, and hard. The details and representative examples are shown in Table 10.

Table 10: Difficulty levels for zero-shot factor generation (Text2Alpha) instructions.

| Level | Design Objective | Instruction Example |
|-------|------------------|---------------------|
| **Easy** | Assess the model's ability to follow clear and direct instructions based on common financial patterns using basic operators and fixed time windows. No signal combination or abstraction is required. | *"Compute a simple moving average of closing prices (default window = 10)."* |
| **Medium** | Evaluate compositional reasoning with moderate abstraction, requiring the model to combine multiple signals (e.g., price and volume) and optionally apply statistical smoothing or normalization. | *"Compute cumulative closing-price return divided by mean volume over the same period (default window = 10)."* |
| **Hard** | Challenge the model to generate complex and creative factor structures involving multi-step logic, standardization, ranking, or conditional rules. | *"Design a factor that ranks stocks by z-scored 10-day cumulative return, then multiplies the rank by the inverse of rolling volume volatility (default return window = 10, volume vol window = 20)."* |

We design 84 easy cases, 106 medium cases, and 121 hard cases. The test cases are created by human experts with the assistance of LLMs, followed by careful filtering to ensure coverage and uniqueness. Easy instructions mainly involve simple operations such as moving averages, ratios, or differences of single price series. Medium instructions combine two or three components, often linking price with volume information, applying normalization, or including simple ranking rules. Hard instructions require multi-step compositions and more advanced logic, such as conditional rules, cross-sectional ranking, winsorization, or interactions across multiple windows.

Each instruction is written in short and implementation-ready form, with explicit default parameters (for example, "default window = 20"). This design makes the instructions clear and unambiguous, while still resembling how financial analysts naturally describe factor ideas.

### B.1.2 DIRECTIONAL FACTOR MINING

Different from the Text2Alpha task, directional factor mining is more flexible and open-ended.

Table 11: Directional factor taxonomy (full).

| Category | Subcategory | Example Signals |
|---|---|---|
| **Trend** | Trend Following | UpTrend; DownTrend; TrendStrength; TrendAcceleration; TrendSlope; RelativePricePosition; CloseOpenGapBias; HighLowRatio |
| | Momentum | MomentumPositive; MomentumNegative; ShortTermMomentum; LongTermMomentum; MomentumAcceleration; MomentumDeceleration; VolatilityAdjustedMomentum; VolumeWeightedMomentum |
| | Mean Reversion | MeanReversionShort; MeanReversionLong; OversoldBounce; OverboughtDrop; Reversal |
| **Volume** | Volume Activity | VolumeSpike; VolumeSurge; VolumeDryUp; RelativeVolumeStrength |
| | Volume Flow | OBV; AccumDist; ChaikinOscillator; MoneyFlowIndex; PriceVolumeTrend |
| | Volume Divergence | PriceVolumeDivergence; PriceVolumeCorrelation |
| **Volatility** | Dispersion | RealizedVol; HistoricalVol; TrueRange; ATR |
| | Band-Based | BollingerWidth |
| | Volatility Dynamics | Expansion; Contraction; Spike; Crush; VolatilityBreakout |
| | Statistical | RollingMean; RollingStd; Skewness; Kurtosis; ZScoreNormalized |
| **Pattern** | Breakout | BreakoutHigh; BreakoutLow; ChannelBreakout; RangeBreakout |
| | Gap | GapUp; GapDown; IslandGap |
| | Candlestick | Doji; ReversalHammer; ReversalShootingStar; EngulfingBull; EngulfingBear; HangingMan; PiercingLine; DarkCloudCover; Marubozu; ThreeWhiteSoldiers; ThreeBlackCrows; MorningStar; EveningStar; Harami; HaramiCross; PinBar |
| | Classical Patterns | DoubleTop; DoubleBottom; HeadAndShoulders; TrianglePattern |
| **Relative Performance** | Core Set | RelativeStrength; Weakness; RSI; StochasticOscillator; MACD |
| **Seasonality** | Calendar Effects | DayOfWeekEffect; MonthEndEffect; HolidayEffect |
| **Risk** | Risk States | CrashRisk; DrawdownAlert; Stability; StableTrend; UnstablePrice |
| **Microstructure Proxy** | Intra-bar Biases | IntradayRangeBias; OpenCloseBias; RangeSkew |

Instead of requiring the model to reproduce a specific formula, we only ask it to generate a factor that belongs to a designated type of signal. This reflects a common workflow in practice, where analysts explore factors within a broader conceptual family (for example, momentum or volatility) rather than replicating a single canonical expression.

To organize the space of directional signals, we design a taxonomy of tags covering the major categories of technical analysis. The taxonomy spans trend, volume, volatility, pattern, relative performance, seasonality, risk, and microstructure, and is further divided into 90 subcategories in total. Representative examples are shown in Table 11, which illustrates how each category is mapped into more fine-grained signal families.

Based on this taxonomy, we construct three difficulty levels for the directional mining task.

- Easy: the instruction specifies a single signal type (90 cases).
- Medium: the instruction requires two different signal types to be combined (65 cases).
- Hard: the instruction requires between three and five signal types to be combined into one factor (60 cases).

For the medium and hard cases, we ensure that the requested signal types are compatible and do not contradict each other. For example, momentum and volatility can be combined naturally, while logically inconsistent requirements are excluded. This design ensures that all tasks remain feasible while still testing the model's ability to integrate multiple signal dimensions into coherent factor expressions. Overall, the taxonomy and difficulty design provide broad coverage of the major classes of directional factors studied in quantitative finance. By requiring models to generate expressions consistent with different signal families, the benchmark evaluates both breadth of knowledge across categories and compositional ability when multiple tags must be satisfied simultaneously.

## B.2 Construction of Factor Evaluation Task

To construct the benchmark for factor evaluation, we use the CSI 300 index constituents as our test universe. The dataset spans the period from 2021 to 2025, covering both a bear market (2023–2024) and a rebound (late 2024 to mid-2025). Through large-scale random searches, we generated 1,762 alpha factors. Their overall performance distribution is shown in Figure 5.

### B.2.1 Assumption

The design of our evaluation tasks is based on the following assumptions:

- Factor comprehension by LLMs. We assume that large language models (LLMs) possess the ability to interpret and judge alpha factors purely from their symbolic expressions. Specifically, LLMs are expected to recognize the construction, components, and signal type of a factor, and to form an internal representation of its potential predictive meaning.

- Comparative reasoning ability. We assume that LLMs are capable of comparing multiple factor expressions and making internal judgments about their relative quality. This includes the ability to weigh structural differences, identify signal overlaps or complementarities, and infer which expressions may yield more reliable or novel signals.

### B.2.2 Ranking

We formally define the ranking task as follows. In a searching step, when an LLM produces $N$ new candidate factors, it is necessary to select the top-$K$ factors as "good" ones to carry forward into the next iteration. A factor is regarded as good if it exhibits sufficiently strong predictive ability, regardless of being a positive or negative signal. Concretely, we apply absolute thresholds on the backtest metrics: $|IC| > 0.025$ or $|RankIC| > 0.03$.

In practice, backtest information may not always be available, so the ranking task challenges the LLM to identify the most promising factors from a given group. Performance is evaluated by ranking-based metrics such as **Precision@K** and **NDCG@K**. To construct the dataset, we design three scenarios that represent increasing levels of difficulty: (1) Small scale: 10 factors with 3 to be identified; (2) Medium scale: 20 factors with 5 to be identified; (3) Large scale: 40 factors with 10 to be identified.

For each test instance, we sample $K$ factors from the good factor pool as ground-truth positives, while the remaining $N - K$ factors are drawn to match the empirical IC distribution of the larger pool. The distribution of factor performance in pool are shown in Figure 5.

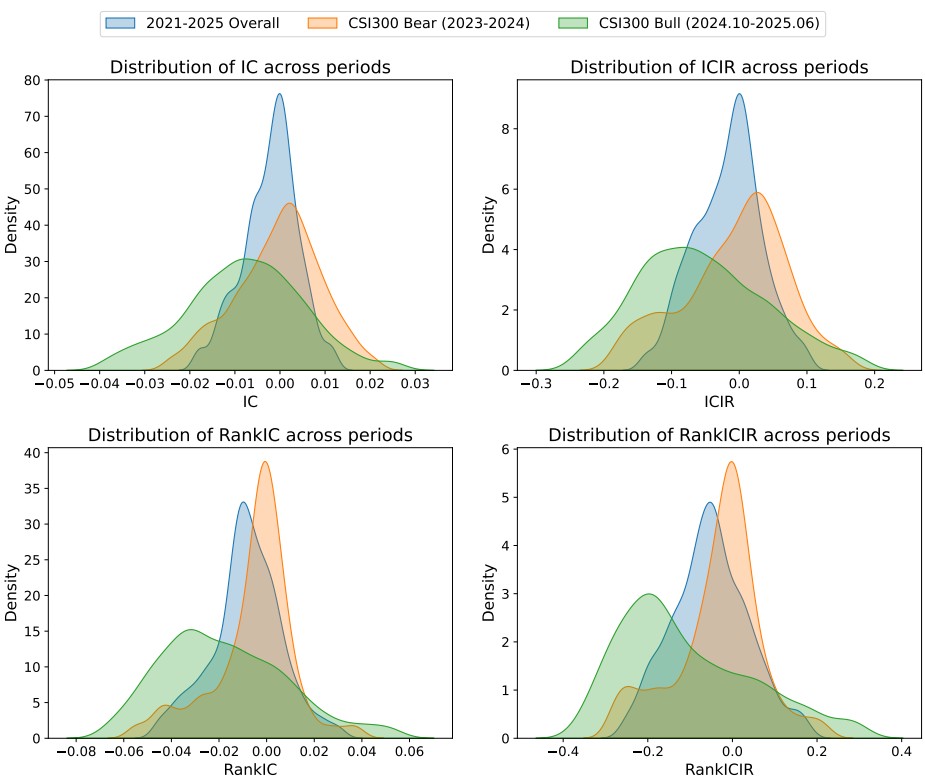

Figure 5: Distributions of predictive performance metrics across different market regimes. We show the density plots of four key metrics: IC, ICIR, RankIC, and RankICIR. These metrics were calculated for factors tested across three distinct periods: the full backtest period (2021–2025), a bear market (CSI300, 2023–2024), and a subsequent bull market (CSI300, Oct 2024–Jun 2025). The analysis highlights how factor performance varies with different market conditions.

### B.2.3 SCORING

The scoring task evaluates an LLM's ability to assess the quality of a single alpha factor based solely on its mathematical expression. Each factor is categorized as **Positive**, **Negative**, or **Noise** based on the sign and magnitude of its average Information Coefficient (IC) or RankIC. For example, a factor with an average $|IC| < 0.01$ is classified as **Noise**. To evaluate the LLM's ability at categorizing the signal, we construct testing datasets from all market regimes of the CSI300 index. Each regime includes 100 **Positive**, 100 **Negative**, and 50 **Noise** factors to ensure a balanced and diverse evaluation set. The LLM's performance is also evaluated by its ability to predict the labels for four numerical dimensions — **Performance**, **Robustness**, **Win Rate**, and **Skewness** (see appendix A.3 for the definitions) — for each factor within its specific category and market regime. The ground truth labels are derived from backtest statistics of the full set of factors. We assign integer scores from 1 to 5 for each dimension based on its ranking within each market regime and category with the exception that all numerical dimension scores for **Noise** factors are fixed to 1.

### B.2.4 ATOMIC TASKS

We posit that evaluating LLMs on formulaic alpha factors can be reduced to two *atomic* decision tasks that capture the most fundamental abilities needed for factor vetting:

- **Signal Classification**: assess whether a given factor exhibits meaningful predictive structure or merely reflects noisy, low-information patterns.

- **Pairwise Selection**: compare two factors under the same backtesting setup and choose the one with higher expected predictive strength.

Absolute IC prediction from expressions is under-determined, since factor performance depends on universe, lag structure, rebalance frequency, market regime, and volatility conditions. In contrast, (i) **Signal Classification** captures whether the factor has usable structure (e.g., normalization, smoothing, stability) and (ii) **Pairwise Selection** captures the basic *relative* decision practitioners routinely make when choosing between candidate factors. Together, they target the most primitive, context-aware evaluative capabilities.

We construct datasets for both the CSI300 (CN) and S&P500 (US) universes. For each market, we sample **500** examples per task, with a fixed **300/200** train/test split. Sampling is balanced by design (noise/signal for FNC, A/B wins for CAPR). All samples use the **latest one-year** window between 2024–2025, with strict time segmentation to avoid leakage. Let $IC_{f,t}$ denote the daily cross-sectional information coefficient for factor $f$ on date $t$. Define

$$\overline{IC}_f = \frac{1}{T} \sum_{t=1}^{T} IC_{f,t}, \qquad ICIR_f = \frac{\overline{IC}_f}{\text{std}(IC_{f,.})},$$

and the *sign-flip ratio*

$$\text{SFR}_f = \frac{\#\{\, t : \text{sign}(IC_{f,t}) \neq \text{sign}(IC_{f,t-1}) \,\}}{T - 1}.$$

A factor is labeled as **noise** if it satisfies at least two of:

$$|\overline{IC}_f| < 0.005, \qquad |ICIR_f| < 0.05, \qquad \text{SFR}_f > 0.60.$$

A factor is labeled as **non-noise (signal)** if it satisfies at least two of:

$$\overline{IC}_f \geq 0.015, \qquad \overline{RankIC}_f \geq 0.025, \qquad ICIR_f \geq 0.10, \qquad \text{SFR}_f \leq 0.40.$$

For the **Pairwise Selection** task, instead of assigning winners based on a fixed IC margin, we directly construct pairs by *randomly sampling one Noise factor and one Non-Noise factor* from the sets defined above. Each pair $(f_A, f_B)$ is thus composed of factors with clearly distinct statistical profiles, and the target label is assigned deterministically: the Non-Noise factor is treated as the stronger candidate in the pair. This sampling scheme further increases contrast between the two classes, avoids ambiguous comparisons, and yields a balanced binary decision problem with a natural 50% random baseline. Each record contains the factor expression and a *structure-only* AST, no realized returns or factor performance metrics are exposed to the model. For **Signal Classification** and **Pairwise Selection**, we report Accuracy, Precision, Recall, and F1, treating "Noise Signal" as the positive class in classification.

### B.3 CONSTRUCTION OF FACTOR SEARCHING TASK

#### B.3.1 CHAIN-OF-EXPERIENCE (CoE)

CoE searching is an extended application of Chain-of-Thought reasoning (Wei et al., 2022). Building on the framework proposed in prior work (Li et al., 2023; Cao et al., 2025), CoE iteratively refines candidate factors by leveraging both the current best solution and the historical trajectory of past explorations. At each round, the LLM is prompted with the seed factor, evaluation results of previous candidates, and the accumulated search history to generate improved variants. The generated candidates are then evaluated, compared against the best-so-far factor, and either accepted as new improvements or discarded. This process naturally forms a trajectory of "experience" that guides the search toward more robust and effective factors. The full procedure is summarized in Algorithm 1.

#### B.3.2 TREE-OF-THOUGHT (ToT)

Prior tree-based alpha search often adopts MCTS, where the LLM mainly expands nodes (Shi et al., 2025b). In our ToT variant (Yao et al., 2023), the search starts from the seed factor, which is evaluated to initialize the best record and history. At each node, the LLM generates $N$ new candidates, duplicates are removed using a global seen set, and all candidates are batch-evaluated. Survivors are

---

**Algorithm 1** Chain-of-Experience (CoE) Searcher

---

**Require:** Seed factor $(name, expr)$; evaluation function $\mathcal{E}(\cdot)$; LLM generator $\mathcal{G}(\cdot)$; number of rounds $R$
**Ensure:** Final best factor and chain history
 1: Evaluate seed expression via API $\rightarrow m_0$ (IC, RankIC, etc.)
 2: Initialize chain history $\mathcal{C} \leftarrow [(expr, m_0)]$
 3: Set current best $(f^*, m^*) \leftarrow (expr, m_0)$
 4: **for** $r = 1$ to $R$ **do**
 5:     Compose instruction $\mathcal{I}_r$ with seed, current best $(f^*, m^*)$, and history $\mathcal{C}$
 6:     Query LLM: $f_r \leftarrow \mathcal{G}(\mathcal{I}_r)$
 7:     **if** $f_r$ valid **then**
 8:         Evaluate $m_r \leftarrow \mathcal{E}(f_r)$
 9:         **if** $m_r$ better than $m^*$ **then**
10:             Update best: $(f^*, m^*) \leftarrow (f_r, m_r)$
11:         **end if**
12:         Append $(f^*, m^*)$ to $\mathcal{C}$
13:     **else**
14:         Append "no candidate" record to $\mathcal{C}$
15:     **end if**
16: **end for**
17: **return** Best factor $(f^*, m^*)$ and chain $\mathcal{C}$

---

then selected by a pruning rule that keeps up to $k$ candidates whose IC exceeds the seed (or otherwise the best-of-round). These survivors are appended to the history, used to update the global best, and recursively expanded until the depth limit $R$. The algorithm finally returns the best factor discovered and the full search trace (Algorithm 3).

---

**Algorithm 2** LLM-Guided Evolutionary Factor Searcher (EA Searcher)

---

**Require:** Initial pool $P = \{(name, expr)\}$; mutation rate $\mu$; crossover rate $\gamma$; candidates per round $N$; rounds $R$; pool size $K$; seed budget $k_{\text{seed}}$
**Ensure:** Final pool $P^{(R)}$ (size $K$), baseline $\overline{\text{IC}}_0$, history $\mathcal{H}$
 1: $P \leftarrow \text{Eval}(P)$ {assign metrics}
 2: $current\_pool \leftarrow \text{TopK}_{\text{IC}}(P, K)$
 3: $\overline{\text{IC}}_0 \leftarrow \mathbb{E}[\text{IC}(current\_pool)]$
 4: $\mathcal{H} \leftarrow [\,]$
 5: **for** $r = 1$ to $R$ **do**
 6:     $S \leftarrow \text{TopK}_{\text{IC}}(current\_pool, k_{\text{seed}})$
 7:     $n_{\text{mut}} \leftarrow \lfloor N\mu \rfloor$
 8:     $n_{\text{cross}} \leftarrow \lfloor N\gamma \rfloor$
 9:     $n_{\text{mut}} \leftarrow n_{\text{mut}} + \max(0, N - n_{\text{mut}} - n_{\text{cross}})$
10:     $M \leftarrow \mathcal{G}_{\text{mut}}(S, n_{\text{mut}})$
11:     $X \leftarrow \mathcal{G}_{\text{cross}}(S, n_{\text{cross}})$
12:     $C \leftarrow \text{DeDup}(M \cup X)$
13:     $C \leftarrow \text{Eval}(C)$
14:     $current\_pool \leftarrow \text{TopK}_{\text{IC}}(current\_pool \cup C, K)$
15:     $\mathcal{H} \leftarrow \mathcal{H} \cup \big[\, (r, S, M, X, C,$
16:             $\max \text{IC}(current\_pool),\ \mathbb{E}[\text{IC}(current\_pool)]) \,\big]$
17: **end for**
18: **return** $current\_pool$, $\overline{\text{IC}}_0$, $\mathcal{H}$

---

### B.3.3 EVOLUTIONARY ALGORITHM (EA)

Building on prior LLM-assisted EA designs (Luo et al., 2026), we adopt a steady-state evolutionary search over a fixed-size factor pool. Each round $r$ selects the top-$k_{\text{seed}}$ seeds by IC, then allocates a candidate budget $N$ to two specialized operators: an LLM mutation agent $\mathcal{G}_{\text{mut}}$ (rate $\mu$) that

perturbs a single parent's expression, and an LLM crossover agent $\mathcal{G}_{\mathrm{cross}}$ (rate $\gamma$) that recombines two parents' sub-expressions. We de-duplicate the union of generated candidates, batch-evaluate them to obtain metrics (IC, RankIC, etc.), and merge them with the current pool, retaining the top-$K$ by IC (steady-state replacement). The baseline $\overline{\mathrm{IC}}_0$ is recorded from the initial pool, and the history $\mathcal{H}$ logs seeds, generated sets, and pool statistics per round. This design decouples mutation and crossover into dedicated LLM agents, enables vectorized evaluation, and enforces global de-duplication and pool size control. The full procedure is given in Algorithm 2.

---

**Algorithm 3** Tree-of-Thought (ToT) Searcher

---

**Require:** Seed factor $(name, expr)$; evaluation $\mathcal{E}$; batch eval $\mathcal{E}_B$; generator $\mathcal{G}$; depth limit $R$; candidates per node $N$; survivor top-$k$
**Ensure:** Best factor $(f^*, m^*)$ and full history $\mathcal{H}$
1: $m_0 \leftarrow \mathcal{E}(expr)$
2: $(f^*, m^*) \leftarrow (expr, m_0)$
3: $\mathcal{H} \leftarrow [(expr, m_0)]$
4: $\mathcal{S} \leftarrow \{ expr \}$ {global seen set}
5:
6: **procedure** SEARCHBRANCH$(parent, depth)$
7: **if** $depth > R$ **then**
8:     **return** $(\{parent\}, parent)$ {(history, best-of-subtree)}
9: **end if**
10: Build instruction $\mathcal{I}$ from seed and $parent$
11: $C \leftarrow \mathcal{G}(\mathcal{I}, N)$ {generate $N$ candidates}
12: Remove duplicates / seen in $C$ using $\mathcal{S}$
13: **if** $C = \emptyset$ **then**
14:     Append "no candidate" to $\mathcal{H}$
15:     **return** $(\emptyset, parent)$
16: **end if**
17: $M \leftarrow \mathcal{E}_B(C)$ {evaluate all candidates}
18: $R_C \leftarrow$ candidates in $C$ sorted by $\mathrm{IC}(M)$ (desc)
19: $Q \leftarrow \{ c \in R_C \mid \mathrm{IC}(c) > \mathrm{IC}(seed) \}$
20: **if** $|Q| \geq k$ **then**
21:     Survivors $\leftarrow$ top-$k$ of $Q$
22: **else if** $|Q| > 0$ **then**
23:     Survivors $\leftarrow Q$
24: **else**
25:     Survivors $\leftarrow$ first of $R_C$ {keep best-of-round}
26: **end if**
27: Update $(f^*, m^*)$ if any survivor beats it
28: child_best $\leftarrow parent$
29: hist $\leftarrow [\,]$
30: **for all** $s \in$ Survivors **do**
31:     $(h, b) \leftarrow$ SEARCHBRANCH$(s, depth + 1)$
32:     hist $\leftarrow$ hist $\cup h$
33:     **if** $\mathrm{IC}(b) > \mathrm{IC}(\text{child\_best})$ **then**
34:         child_best $\leftarrow b$
35:     **end if**
36: **end for**
37: **return** (hist, child_best)
38: **end procedure**
39:
40: $(h_{\mathrm{root}}, b_{\mathrm{root}}) \leftarrow$ SEARCHBRANCH$((expr, m_0), 1)$
41: $\mathcal{H} \leftarrow \mathcal{H} \cup h_{\mathrm{root}}$
42: **return** $b_{\mathrm{root}}$ and $\mathcal{H}$

---

## C   EVALUATION METRICS OF ALPHABENCH

In this section, we provide detailed descriptions and definitions of the evaluation metrics used in AlphaBench. These metrics are designed to assess the performance of large language models (LLMs) across different tasks, including factor generation, factor evaluation, and factor searching, they are summarized in Table 12

Table 12: Summary of Evaluation Metrics in AlphaBench

| | | |
|---|---|---|
| **Generation** | **Reliability** | Measures the consistency of LLMs in producing valid factor expressions across retries. |
| | **Stability** | Assesses how consistently LLMs generate factors with similar predictive behavior under different prompts. |
| | **Accuracy** | Evaluates whether the generated factor faithfully meets the requirements of the instruction. |
| **Evaluation** | **Precision@K** | The fraction of truly good factors among the top-$K$ selected by the model in ranking tasks. |
| | **Signal Type Accuracy** | The accuracy of the model's predicted signal type (Positive, Negative, Noise). |
| | **Mean Absolute Error (MAE)** | Measures the difference between predicted and true values for numerical dimensions (Performance, Stability, etc.). |
| **Searching** | **Search Cost** | The number of retries per step in the factor search task, indicating the cost of generating valid factors. |
| | **Success Rate** | The ratio of valid factors to the total number of candidates generated in a search task. |
| | **IC Improvement** | Measures the improvement in the Information Coefficient (IC) of a factor. |
| | **Best Update Rate (EA)** | The frequency of successful updates in the evolutionary algorithm (EA) search, indicating the effectiveness of factor generation over iterations. |
| | **Diversity** | Evaluates the structural and behavioral diversity of the generated factors in the search process. |

### C.1   METRICS FOR EVALUATING FACTOR GENERATION

#### C.1.1   RELIABILITY

Reliability measures how consistently an LLM can produce a valid factor expression in response to a natural language instruction, without requiring manual correction. For each test instruction, the model is allowed up to $N = 5$ retries if the generated output fails to satisfy basic syntactic or semantic validity checks. We define the reliability score of an individual instruction as:

$$R_i = \max\left(0,\, 1 - 0.2 \cdot f_i\right),$$

where $f_i$ is the number of failed retries before a valid factor is obtained. If no valid response is generated within $N$ attempts, then $R_i = 0$.

The overall reliability is the average across all instructions at a given difficulty level:

$$\text{Reliability} = \frac{1}{M} \sum_{i=1}^{M} R_i,$$

where $M$ denotes the total number of test instructions. This metric captures both the success rate of producing valid factors and penalizes frequent failures by reducing the score proportionally to the number of retries used. Finally, the overall reliability score is a weighted average across the three levels:

$$\text{Reliability}_{\text{overall}} = 0.3 \cdot \text{Reliability}_{\text{easy}} + 0.3 \cdot \text{Reliability}_{\text{medium}} + 0.4 \cdot \text{Reliability}_{\text{hard}}.$$

This weighting emphasizes performance on harder instructions, which better reflect the model's robustness in more challenging scenarios.

### C.1.2 STABILITY

To evaluate the stability of LLMs under instruction variations, we select **17** factors from the Alpha158 library with medium or higher difficulty. For each factor, we use LLMs to generate 10 semantically equivalent instructions describing the same logic in different ways (see Table 14 for an illustration). This approach simulates real-world use cases to account for the variation in how different users might express the same analytical idea.

Table 14: Instruction variants for stability evaluation

| | |
|---|---|
| **Expression** | `((Mean($close, 10) - Ref($close, 20)) / $close) * (Mean($volume, 5) > Ref(Mean($volume, 5), 10))` |
| **Origin Instruction** | Calculate the N-day (default 10) mean of the close price minus the close price N days (default 20) ago, divided by the current close price, multiplied by a boolean indicating whether the N-day (default 5) mean of volume is greater than the same mean N days (default 10) ago. |
| **New Instructions** | • Compute (N-day (default 10) average close less close N days (default 20) ago) / current close, times (N-day (default 5) average volume greater than the same average N days (default 10) prior).
• Find the product of ((N-day (default 10) mean close minus close N days (default 20) ago) / close) and (N-day (default 5) mean volume exceeding its value N days (default 10) ago).
• Derive the factor as ((N-day (default 10) mean of close minus close N days (default 20) ago) divided by close) times (whether N-day (default 5) mean volume is greater than N-day (default 5) mean volume N days (default 10) ago).
• Calculate ((N-day (default 10) mean of close - close N days (default 20) ago) / close) * (N-day (default 5) mean of volume > N-day (default 5) mean of volume N days (default 10) ago).
• Generate a factor by multiplying the ratio of (N-day (default 10) mean close less close N days (default 20) ago) to close, and a boolean for (N-day (default 5) mean volume exceeding its value N days (default 10) ago). |

A natural idea for measuring stability is to directly match the similarity between generated factor expressions. However, this approach is often difficult and overly strict, since factors with different surface forms may still capture essentially the same signal. To address this, we instead evaluate stability at the level of factor outputs. The intuition is that if an instruction is truly stable, then even with perturbations (e.g., random seeds, slight prompt variations), the generated factors should behave similarly when applied to market data.

Formally, given two generated factors $f^{(a)}$ and $f^{(b)}$, we compute their correlation over realized returns in the backtest window (January 2023 to March 2025). The stability metric is defined as the average pairwise correlation among all generated factors for the same instruction:

$$\text{Stability} = \frac{2}{K(K-1)} \sum_{1 \le a < b \le K} \text{Corr}\left( f_t^{(a)}, f_t^{(b)} \right),$$

where $K$ is the number of generated candidates per instruction, and Corr is measured on the time series of factor values within the given backtest period. A higher stability score indicates that the LLM produces factors with consistent predictive behavior, rather than erratic or divergent outputs.

### C.1.3 GENERATION ACCURACY

Generation accuracy evaluates whether the generated factor faithfully meets the requirements specified in the instruction. Directly performing an exact match on the expression or relying purely on human inspection is difficult to scale and often ambiguous. To address this, we adopt an *LLM-as-a-judge* paradigm: a separate evaluator LLM is prompted to check whether the generated factor expression satisfies the intended requirements (e.g., correct variables, operations, or structural constraints). The prompt used in this task are shown in Figure 6. This evaluation is formulated as a binary classification task, where each output is labeled as either *correct* or *incorrect*.

To ensure reliability, we also conduct a small-scale human audit on sampled outputs to validate the evaluator's consistency. The final accuracy score is then defined as:

$$\text{Accuracy} = \frac{1}{M} \sum_{i=1}^{M} \delta(\text{LLMJudge}(f_i)),$$

where $M$ is the total number of generated factors, $f_i$ denotes the $i$-th factor, and $\delta(\cdot)$ is an indicator function that returns $1$ if the judgment is *correct* and $0$ otherwise. A higher accuracy indicates that the model can more faithfully follow instructions to generate factors that align with task requirements.

Similar to reliability, we report accuracy separately at each difficulty level $d \in \{\text{easy}, \text{medium}, \text{hard}\}$ and then compute the overall weighted score:

$$\text{Accuracy}_{\text{overall}} = 0.3 \cdot \text{Accuracy}_{\text{easy}} + 0.3 \cdot \text{Accuracy}_{\text{medium}} + 0.4 \cdot \text{Accuracy}_{\text{hard}}.$$

This weighting ensures that performance on more challenging instructions contributes more strongly to the final evaluation.

---

**Generation Judgement Agent Prompt**

You are an expert quantitative researcher who evaluates alpha factor generation.

Your role is to judge whether the generated factor (in JSON format) faithfully follows the given instruction,
regardless of whether the task is Text2Alpha generation (natural language −> formula) or Directional Mining (generate diverse factors under a theme).
Think step by step before giving the final judgement.

### Input
− Task Type: {task}
− Instruction: {instruction}
− Generated Factor (JSON): {factor}

### Evaluation Criteria
1. **Variable Validity**
   Must only use allowed variables: \$close, \$open, \$high, \$low, \$volume.
   If any other variable appears, it is invalid.

2. **Operator Validity**
   Must only use allowed operators/functions.
   Must also satisfy any explicit requirements (e.g., Mean, Std, normalization, ratio, rolling window).
   If operator usage is missing, extra, or wrong, it is invalid.

3. **Instruction Faithfulness**
   Expression must reflect the intent of the instruction.
   Examples:
   − If instruction requires a 20−day moving average, check for `Mean(..., 20)`.
   − If instruction requires normalization/z−score, check scaling by Std.
   − If instruction requires ratio (e.g., high/low), check numerator and denominator match.

– If instruction asks for multiple factors under a theme (e.g., volatility), verify all generated factors belong to that theme.
Any deviation means invalid.

4. **JSON Format Consistency**
Output must be valid JSON with required keys (`expression.template`, `expression.parameters`).
If missing or malformed, it is invalid.

### Output
First, reason step by step to yourself (chain-of-thought). Then return the final decision in **strict JSON format**:

```json
{
  "reason": "<short explanation of why it is correct or incorrect>",
  "result": "correct" or "incorrect"
}
```

Figure 6: System prompt for the Generation Judgement Agent, which evaluates whether a generated alpha factor in JSON format faithfully follows the given instruction. The agent checks variable validity, operator correctness, instruction faithfulness, and JSON consistency before outputting a final binary judgement.

As an illustration, Table 15 shows a concrete example of a generated factor under the *Directional Mining* task and its evaluation outcome.

Table 15: Example of evaluation result for a generated factor under the Directional Mining task.

| Field | Content |
|---|---|
| Task Type | Directional Mining |
| Instruction | You are a financial quantitative researcher. Please write a valid factor equivalent to the following description: "Sum open-to-close differences over the short horizon (default window = 7)." |
| Generated Factor (JSON) | `{` `    "name":  "sum_open_to_close_diff_7_2abd49d9",` `    "expression":  "Sum(Sub($close, $open), 7)"` `}` |
| LLM Output | `{` `    "reason":  "The factor correctly computes the sum over a 7-day window of the difference between close and open prices, which matches the instruction.  It uses valid variables ($close, $open) and appropriate operators (Sum, Sub).  The factor aligns with the directional momentum theme.",` `    "result":  "correct"` `}` |

## C.2 METRICS FOR EVALUATING FACTOR EVALUATION

### C.2.1 RANKING

For the ranking task, we measure the ability of an LLM to correctly identify the most promising factors within a candidate set. Two standard metrics are used:

- **Precision@K**:

$$\text{Precision@}K = \frac{|\,\text{Top-}K \cap \text{Good}\,|}{K},$$

  which represents the fraction of truly good factors among the top-$K$ selected by the model.

### C.2.2 SCORING

For the scoring task, we evaluate the agreement between the LLM's predicted labels and the ground-truth scores derived from backtests. The evaluation includes both categorical and numerical dimensions:

- **Signal Type Accuracy**:

$$\text{Accuracy} = \frac{1}{N}\sum_{i=1}^{N}\mathbf{1}\{\hat{y}_i = y_i\},$$

  where $y_i \in \{\text{Positive}, \text{Negative}, \text{Noise}\}$ is the true signal type and $\hat{y}_i$ is the model's prediction.

- **Mean Absolute Error (MAE, single dimension)**:

$$\text{MAE}_d = \frac{1}{N}\sum_{i=1}^{N}\big|\hat{s}_i^{(d)} - s_i^{(d)}\big|,$$

  where $d \in \{\text{Performance}, \text{Stability}, \text{Win Rate}, \text{Skewness}\}$ denotes one numerical dimension, and $s_i^{(d)}$ (ground truth) and $\hat{s}_i^{(d)}$ (prediction) are the integer scores of factor $i$ on that dimension.

These metrics jointly assess whether an LLM can act as a reliable evaluator of factor quality, capturing both high-level signal classification and fine-grained quantitative scoring.

## C.3 METRICS FOR EVALUATING FACTOR SEARCHING

To evaluate performance in the factor searching task, we consider two complementary dimensions: *search efficiency* and *search performance*.

### C.3.1 SEARCH EFFICIENCY

- **Search Cost**:

$$\text{Cost} = \text{Retries per step}, \quad 1 \leq \text{Cost} \leq 5,$$

  where retries denote the number of attempts required for one valid search step. In the factor searching task, both single and multi-generation processes involve the possibility that the LLM may fail to generate a valid result in a single session. We set a maximum of 5 attempts for retries, with the cost being 1 when the task is completed in a single session and 5 when it requires the maximum number of retries. A higher cost indicates that the LLM struggled to complete the task within the allowed attempts, making the result less efficient.

- **Success Rate**:

$$\text{SuccessRate} = \frac{N_{\text{valid}}}{N_{\text{total}}},$$

  where $N_{\text{valid}}$ is the number of valid factors passing the filtering stage, and $N_{\text{total}}$ is the total number of candidates generated. This ratio records the success of the generation process, especially in population-based search tasks such as Evolutionary Algorithm (EA). It is computed for each search task and reflects the proportion of generated factors that meet the validation criteria in Appendix A.5, thereby indicating the effectiveness of the search process in producing useful factors according to our standards.

### C.3.2 SEARCH PERFORMANCE

- **IC Improvement**: A factor is considered successful if

$$\mathrm{IC} > 0.03,$$

  and its normalized gain is defined as

$$\mathrm{Gain} = \min\left(\max\left(\frac{\mathrm{IC}}{0.03}, 0\right), 1\right)$$

  In our task, we specifically require factors to have a positive IC value. Therefore, we set a threshold of 0.03, where factors exceeding this value are recognized as effective. The gain ratio measures the improvement of a factor based on its IC, quantifying how much it exceeds the threshold.

- **Fraction of Successful Runs (CoE / ToT)**:

$$\mathrm{FracSuccess} = \frac{N_{\text{runs with improvement}}}{N_{\text{total runs}}},$$

  where the numerator counts runs that discover at least one factor above the IC threshold.

- **Best Update Rate (EA)**:

$$\mathrm{UpdateRate} = \frac{1}{T}\sum_{t=1}^{T} U_t,$$

  where $T$ is the number of iterations, and $U_t$ is the number of successful updates among the top factors at iteration $t$. This metric is used to measure the update frequency in EA-based searching. At each round, the pool of top-$n$ factors is updated, and this helps track how often effective factors are generated. Typically, in the early rounds of the search, a higher update rate indicates that the search is generating factors more frequently and that the LLMs have high potential in generating effective factors.

## D  PROMPTS DESIGN AND OUTPUT SAMPLES

In both the *generation* and *searching* tasks, we design the system prompts with a shared Qlib reference section. This reference serves as a unified guideline that specifies the supported operators, function signatures, and safety rules for constructing valid Qlib factor expressions. By providing the same reference content across tasks, we ensure consistency in factor generation and avoid errors such as missing arguments, unbalanced parentheses, or unsafe divisions. The reference also encodes strict requirements (e.g., mandatory use of functional operators instead of symbols, rolling window constraints so that the LLMs can reliably follow Qlib's factor definition standards.

---

**Qlib Generate Instruction**

\*\*Arithmetic / Logic\*\*
– Add(x,y), Sub(x,y), Mul(x,y), Div(x,y)
– Power(x,y), Log(x), Sqrt(x), Abs(x), Sign(x), Delta(x,n)
– And(x,y), Or(x,y), Not(x)
– Sqrt(x), Tanh(x)
– Comparators: Greater(x,y), Less(x,y), Gt(x,y), Ge(x,y), Lt(x,y), Le(x,y), Eq(x,y), Ne(x,y)

\*\*Rolling (n is positive integer)\*\*
– Mean(x,n), Std(x,n), Var(x,n), Max(x,n), Min(x,n)
– Skew(x,n), Kurt(x,n), Sum(x,n), Med(x,n), Mad(x,n), Count(x,n)  | Med is for median and
 Mad for Mean Absolute Deviation
– EMA(x,n), WMA(x,n), Corr(x,y,n), Cov(x,y,n)
– Slope(x,n), Rsquare(x,n), Resi(x,n)

\*\*Ranking / Conditional\*\*
– Rank(x,n), Ref(x,n), IdxMax(x,n), IdxMin(x,n), Quantile(x,n,qscore (float number
between 0–1))
– If(cond,x,y), Mask(cond,x), Clip(x,a,b)

Note: function signatures must be complete.
– Corr(x,y,n) requires 3 arguments
– Quantile(x,n,qscore) requires 3 arguments
– Rank(x,n) requires 2 arguments
– Ref(x,n) requires 2 arguments

Important rules:
a. For arithmetic operations, do NOT use symbols. Instead, use: Add for +, Sub for –, Mul
for ∗, Div for /
b. Parentheses must balance.
c. Correct arity, no missing arguments.
d. Rolling windows (n) must be positive integers.
e. Division safety, always add epsilon:
   – Div(x, Add(den, 1e−12)) correct
   – Div(x, den) incorrect
  Sqrt safely, ensure no negative inputs.
f. No undefined / banned functions (e.g., SMA, RSI), and above operation is low/upper–
case sensitive.
g. Expressions must be plain strings, no comments or backticks.

---

Figure 7: Qlib operator reference used in our Generation task.

### D.1  GENERATION TASK

The prompts for the generation task mainly consist of two parts. First, we describe the role of the model and the overall task, asking it to act as a quantitative researcher and generate a new factor

in JSON format. Second, we provide detailed instructions about the available operators in QLib, including arithmetic, rolling statistics, comparisons, and conditional logic, together with rules such as using CamelCase and defining all parameters. These two parts ensure that the model produces both valid and executable factor expressions.

Here we provide example of generated task instruction and sample output in table

---

**Generation Prompt**

You are an expert quantitative researcher who designs alpha factors for stock ranking. Please generate a new factor in JSON format with the following requirements:

1. The factor must be defined using a mathematical expression in a string template, provided under "expression.template".
2. Allowed variables (must be prefixed with '\$') include: \$close, \$open, \$high, \$low, \$volume.
3. Use only CamelCase function names and operators (Qlib style).

More details about operators are given in the Qlib reference below:
QLIB_GENERATE_INSTRUCTION

4. The factor should be efficient: execution within 30 seconds, max depth 4.
5. Self−check: variables in \{close,open,high,low,volume\}; functions in allowed list; \(n\) positive integer; balanced parentheses; safe division (add 1e−12).
6. Output must be a single valid JSON object.

IMPORTANT: Keep JSON a **single** object with no trailing commas.

**Vanilla Output Example:**
```
{
  "name": "meanreversion_short_term",
  "expression": "Div(Sum(\$close, 5), Sum(\$volume, 10))"
}
```

**CoT Output Example:**
Please use Chain of Thought reasoning to generate the factor and write your thinking process in the 'CoT' field of the output JSON.
The CoT progress includes (1) What sub item you want to use (2) How to combine them step by step (3) Format the final expression in the 'expression.template' field. Make sure don't write too much in CoT.
```
{
  "name": "meanreversion_short_term",
  "CoT": "This is your chain of thought thinking process...",
  "expression": "Div(Sum(\$close, 5), Sum(\$volume, 10))"
}
```

---

Figure 8: Instruction prompt template for the factor generation task. The template instructs the model to output a factor in JSON format with strict variable and operator constraints. Two output modes are supported: (1) a **Vanilla mode**, where only the final expression is returned, and (2) a **CoT (Chain-of-Thought) mode**, where the model must also include a concise reasoning trace in the CoT field explaining the construction steps before producing the final expression.

Table 16: Example of evaluation result for a generated factor under the Text2Alpha task.

| Field | Content |
|---|---|
| Task Type | Text2Alpha |
| Instruction | Produce high-low range divided by rolling volume standard deviation (default window = 15). |
| CoT | To capture intraday volatility normalized against trading activity, I will use the daily high-low price range as the numerator. For the denominator, I will compute the standard deviation of volume over a rolling window. Step 1: Compute the difference between $high and $low for the daily price range. Step 2: Compute Std($volume, 15) for the rolling standard deviation of volume. Step 3: Normalize price range by dividing it by the rolling volume standard deviation. Step 4: Format this as Div(Sub($high, $low), Std($volume, 15)). |
| Output | `Div(Sub($high, $low), Std($volume, 15))` |

Table 17: Example of evaluation result for a generated factor under the Directional Mining task.

| Field | Content |
|---|---|
| Task Type | Directional Mining |
| Instruction | You are a financial quantitative researcher. Please write a valid factor equivalent meets this direction (we provide the details of each direction tag): MicrostructureProxy>IntradayRangeBias: 
 - Level 1: Pseudo-microstructure factors derived from OHLCV. 
 - Level 2: Close position within intraday high-low range." |
| CoT | To capture the close position within the intraday high-low range, I first calculate the range by Subtracting low from high. Then I find how far the close is from the low (Sub close, low). Dividing this difference by the range gives the relative position of close within the day's range, which ranges from 0 to 1. To ensure safety, I add 1e-12 in the denominator to avoid division by zero. This is a straightforward pseudo-microstructure factor that directly measures close position bias within the day's price range. |
| Output | `Div(Sub($close, $low), Add(Sub($high, $low), 1e-12))` |

## D.2 EVALUATION TASK

To systematically assess the ability of LLMs to evaluate candidate alpha factors, we introduce two complementary tasks: Factor Ranking and Factor Scoring. Both tasks are designed to move beyond pure formula generation, focusing instead on judgment, reasoning, and interpretability—key capabilities required for practical quant research.

### D.2.1 FACTOR RANKING

In this task, models are given $N$ candidate factors, each with its symbolic expression and a brief natural language description. The model is instructed to identify the top-$k$ factors most likely to exhibit true predictive ability for stock returns and generalize across market regimes. A "good factor" is defined by three principles: (i) stronger absolute IC/RankIC performance in backtests (sign irrelevant, as both positive and negative correlations are useful), (ii) economic interpretability (momentum, reversal, volatility, liquidity, etc.), and (iii) robustness and stability against overfitting or spurious noise. The prompt explicitly emphasizes these requirements and guides models to reason about both statistical validity and economic plausibility. As an example, in a 10-pick-3 scenario, the model must examine ten provided factors, interpret their construction (e.g., VWAP deviations, price–volume divergences, breakout rules), and output the top three candidates. This setting measures whether models can prioritize robust, interpretable signals over noisy or convoluted formulas.

**Factor Ranking Agent Prompt**

You are an expert quantitative researcher.
You are given N candidate alpha factors, each with an expression and a short description.
Your task is to identify the top−k factors that are most likely to have true predictive ability
for stock returns and perform well in trading.

Definition of a good factor:
− It should demonstrate strong predictive power, reflected by absolute higher Information
Coefficient (IC)
  and RankIC in backtests, notice in this case we care about absolute values, because no
matter positive or negative, the value is higher, the better performance.
− It should capture meaningful and economically intuitive patterns (e.g., momentum,
reversal, volatility, liquidity).
− It should be robust and generalizable, avoiding spurious correlations or overfitting to
noise.
− Its mathematical form should align with known signal categories and provide stable
signals.

The given example is ID with their expressions:
−1: expr
−2: expr
....

Figure 9: Instruction prompt used to guide LLMs in ranking candidate alpha factors by predictive
quality.

Table 18: Example of evaluation result for a factor ranking task (10-pick-3 scenario).

| Field | Content |
|---|---|
| Instructions | Rank the following 10 candidate factors by expected robustness across up, down, and neutral markets under the **10_pick_3** scenario; output the top 3. |
| Given Factors | ``` 1:  Div(Sub($close, Ref($close, 5)), Ref($close, 5)) 2:  Div(Sub(Slope(Sum(Mul(Div(Sub($close, Ref($close, 1)), Ref($close, 1)), $volume), 10), 7), Mean(Slope(Sum(Mul(Div(Sub($close, Ref($close, 1)), Ref($close, 1)), $volume), 10), 7), 20)), Std(Slope(Sum(Mul(Div(Sub($close, Ref($close, 1)), Ref($close, 1)), $volume), 10), 7), 20)) 3:  Sum(Div(Sub($close, $open), Std($close, 10)), 5) 4:  Mul(Sub($close, Div(Sum(Mul($close, $volume), 10), Sum($volume, 10))), Div($volume, Mean($volume, 10))) 5:  Mul(Sub($close, Mean($close, 120)), If(Lt(Div(Mean($volume, 5), Mean($volume, 30)), 0.18), 1, 0)) 6:  Sub(Rank(Div(Sub($close, Ref($close, 1)), Ref($close, 1)), 21), Rank(Div(Sub($volume, Ref($volume, 1)), Ref($volume, 1)), 21)) 7:  Std(Power(Mul(Power(Sub($close, Ref($close, 1)), 2), Power(Sub($volume, Ref($volume, 1)), 2)), 0.25), 60) 8:  Mul(Div(Std($close, 10), Max(Std($close, 10), 60)), If(Eq($close, Max($close, 14)), -1, 0)) 9:  Mul(Gt($close, Max(Ref($high, 1), 10)), Gt($volume, Mean($volume, 20))) 10: Mul(-1, Div(Sub(Skew(Div(Sub($close, Ref($close, 1)), Ref($close, 1)), 20), Kurt($volume, 20)), Sqrt(Add(Mul(Std(Div(Sub($close, Ref($close, 1)), Ref($close, 1)), 20), Std(Div(Sub($close, Ref($close, 1)), Ref($close, 1)), 20)), Mul(Std($volume, 20), Std($volume, 20)))))) ``` |
| Output CoT | First, assess each factor's robustness and interpretability. (1) 5-day return is classic but potentially crowded; moderate robustness. (2) Z-scored slope of 10-day volume-weighted returns is complex and likely noisy. (3) Summed intraday momentum normalized by 10-day std has reasonable intuition. (4) Deviation from 10-day VWAP scaled by relative volume captures value + liquidity; robust across regimes. (5) Long-term mean deviation gated by low volume is over-conditional and unstable. (6) Price–volume divergence via rank spread is a classic, robust motif. (7) Vol-of-vol concoction is hard to interpret and likely noisy. (8) Volatility-scaled 14-day peak flag is reversal-like but can be twitchy. (9) Breakout with volume confirmation is a proven pattern with regime-agnostic appeal. (10) Higher-moment blend is convoluted and unstable. Prioritize signals with clear economics and normalization that generalize across up/down/neutral markets—thus favor (4), (6), (9). |
| Output Ranking | [4, 6, 9] |

D.2.2 FACTOR SCORING

The second task evaluates whether models can directly score the predictive quality of a single factor expression in a *zero-shot* setting. Here, only the raw symbolic formula is provided, without natural language description or backtest results. The model must infer the factor's likely behavior based solely on operator composition, window choices, normalization schemes, and economic heuristics. It outputs a JSON object with:

- A qualitative analysis of the factor design
- A predicted signal direction (Positive, Negative, or Noise)
- Four numeric scores (1–5 scale) for *Performance*, *Stability*, *WinRate*, and *Skewness*.

Table 19: Example of evaluation result for a factor scoring task (Bull market).

| Field | Content |
|---|---|
| Instructions | Please evaluate the factor `negative_momentum_drawdown_alert` with the following expression in the **Bull** market environment:
`If(Gt(Max(Sub($high, $close), 10), Mul($close, 0.1)),`
`Sub(0, Slope($close, 20)), 0)` |
| Output (JSON) | ```json
{
  "analysis": "Expression checks if the 10-day max
drawdown from high to close exceeds 10% of close; if so,
outputs negative of 20-day close slope (i.e., reversal
after large drawdown), else zero. Operators: Max, Sub,
Slope (window 20), If, Gt, Mul. No leakage. Archetype:
mean-reversion trigger after large drop. Signal
direction: Negative (betting on reversal after sharp
fall). Performance: plausible event-driven reversal,
but conditional logic means less frequent activation.
Stability: moderate, as slope is smoothed but output
is sparse (mostly zero). WinRate: moderate, as signal
only fires on large events, so hit rate might be low.
Skewness: low, as reversal signals after drawdown can
have left-tail risk.",
  "signal": "Negative",
  "scores": {
    "Performance": 3,
    "Stability": 3,
    "WinRate": 2,
    "Skewness": 2
  }
}
``` |

The prompt encodes explicit semantic rules: larger windows imply smoother signals, rank/z-score normalization stabilizes outputs, while excessive nesting, division by small std, or use of lookahead leaks lead to penalties. Economic archetypes are also built in: trend-following motifs are marked Positive, overbought–oversold reversals as Negative, and trivial/noisy expressions as Noise. Hard penalties ensure that any factor with lookahead (Ref/Delay $k \leq 0$) or constant output is scored as Noise with minimal scores.

> **Factor Scoring Agent Prompt**
>
> You are an expert quantitative researcher. This is a **zero−shot factor scoring** task.
> You will ONLY receive a factor expression (symbolic formula). You must infer its likely behavior
> from the expression structure (operators, windows, normalization, delays), and output scores.
>
> ### Goal

For each factor expression, predict:
− "signal": "Positive" | "Negative" | "Noise"
− "scores": 1−−5 for each dimension:
  \# Performance  (higher expected predictive strength is better)
  \# Stability    (lower variability / more robustness is better)
  \# WinRate      (likelihood of correct directional calls is higher)
  \# Skewness     (right−tail / positive−outlier tendency is better)

Output ∗∗ONLY∗∗ a single JSON object per factor with the schema below. No extra text.

### Assumptions \& semantics
− Variables: \$close, \$open, \$high, \$low, \$volume (daily).
− Lag/Delay: Delay(x, k) or Ref(x, k) uses k−day lag (\(k > 0\)) means past). Any non−positive lag (\(k \leq 0\)) is data leakage $\rightarrow$ severe penalty.
− Ranking/Normalization: Rank(.), ZScore(.), Normalize(.), Division by Std/Vol are stabilizers.
− Windows: Larger windows imply smoother/stabler signals; overly small windows (\(<5\)) are noisy.
− Composition risk: Excessive nesting, unstable operations (Exp, Pow with large exponents, division by small Std) reduce Stability and WinRate.
− Economic archetypes (heuristics):
  \# Momentum/Trend (e.g., MA cross, ROC, breakout, positive weights on recent returns) $\rightarrow$ Positive
  \# Mean−reversion (e.g., raw ZScore(price vs MA), deviation−from−mean without a negative sign) $\rightarrow$ Negative
  \# Volatility−carry (stable realized vol or ATR changes with sensible smoothing) $\rightarrow$ usually Positive
  \# Pure price level, raw difference without horizon or smoothing $\rightarrow$ Noise (unless strongly structured)
  \# If the expression is nearly constant or trivially transforms a constant $\rightarrow$ Noise

### Direction (signal) inference rules
Decide sign based on how the factor value co−moves with expected future return:
− If factor increases with recent positive returns (trend), e.g. is −ZScore(deviation) for overbought/oversold buy−low logic $\rightarrow$ "Positive".
− If factor is raw overbought measure, e.g. ZScore(close − MA\_N) (high value = overbought $\rightarrow$ expect negative future return) $\rightarrow$ "Negative".
− If expression is ambiguous, level−only, or dominated by noise/unstable ops $\rightarrow$ "Noise".
− Any data leakage (Ref/Delay with \(k \leq 0\)) $\rightarrow$ set "signal"="Noise" and minimum scores.

### Dimension scoring (1−−5) from expression heuristics
Score each dimension independently using these qualitative cues:

1) Performance (higher is better)
  + Higher: classic motifs (trend, mean−reversion, carry), sensible windows (10−−60), use of ranking/normalization, combination with volume/vol for conditioning, simple and interpretable structure.
  −− Lower: level−only, arbitrary composites with no economic story, extreme nonlinearity (Exp, high Pow), tiny windows, algebraic cancelations, obvious leakage flags.
  Heuristic tiers:
    5: canonical well−formed signal (e.g., multi−window momentum or robust z−scored reversal) with stabilization
    4: solid single−idea factor with some smoothing/normalization
    3: plausible but simplistic or slightly noisy

2: weakly specified, likely low edge
1: meaningless/unstable/leaky

2) Stability (higher is better when smoother/robust)
 + Higher: Rank/ZScore/Normalize, long windows, moving averages, EWMA, median filters, winsorization/clipping.
 −− Lower: division by tiny Std, nested differences, tiny windows ($<5$), Exp/Pow explosions, many unbounded mult/div chains.
 Map: very smooth \& normalized = 5; moderate smoothing = 3−−4; noisy/unstable = 1−−2.

3) WinRate (higher is better)
 Proxy from construction:
 + Higher: stable signals with incremental edges (trend with smoothing, carry), rank−based thresholds, ensemble/averaging $\rightarrow$ more frequent small wins.
 −− Lower: tail−dependent or trigger−based signals (hard thresholds, breakout−only) $\rightarrow$ lower hit−rate but bigger payoffs.
 Map: smoothed trend/carry with rank/zscore = 4−−5; noisy thresholds = 2−−3; chaotic = 1.

4) Skewness (higher prefers right−tail outcomes)
 + Higher: breakout/regime/trend−follow signals that occasionally capture large moves; conservative normalization limiting downside.
 −− Lower: mean−reversion with crash risk, leverage via division/Exp causing left tails.
 Map: breakout/trend with controls = 4−−5; balanced smooth factors = 3; reversal with downside/tight division = 1−−2.

### Hard penalties
− Any lookahead/leakage (Ref/Delay $(k \leq 0)$, using future bars) $\rightarrow$ "signal":" Noise"; all scores=1.
− Expressions that are constant or nearly constant $\rightarrow$ "Noise"; all scores=1.

Figure 10: Instruction prompt used to guide LLMs in zero-shot scoring of alpha factor expressions.

## D.3 SEARCHING TASK

For the searching task, we design dedicated system prompts that differ from those used in the generation task. The key distinction is that searching requires multi-factor generation within a single session, along with explicit guidance for planning, pruning, and diversifying candidate factors. Therefore, the prompt specifies a universal search process—plan → generate → heuristic evaluation → refine/prune → diversify—that mirrors how human quantitative researchers typically explore a factor space. As shown in Figure 11, the prompt emphasizes several aspects:

• Output structure: factors must be returned in JSON list format with unique, descriptive CamelCase names and valid Qlib expressions.

• Constraints: only allowed market variables and operators can be used, with careful validation rules (balanced parentheses, safe denominators, positive windows).

• Search methodology: candidates are generated according to distinct blueprints (e.g., momentum, mean reversion, volatility scaling, volume conditioning), heuristically evaluated for normalization and stability, and refined by pruning redundancies.

• Diversity: the final set must cover multiple archetypes such as price momentum, volume-adjusted signals, volatility-normalized returns, and cross-variable relations.

**Searching Prompt**

You are an expert quantitative researcher who designs alpha factors for stock ranking. Your job is a **search task**: plan $\rightarrow$ generate $\rightarrow$ evaluate heuristically $\rightarrow$ refine/prune $\rightarrow$ diversify, then output candidates. Please generate N new factors in JSON list format.

### Requirements:
1. **Output Format**
   – Return a JSON list. Each element should be an object with:
     – key: factor name (string, CamelCase, short but descriptive)
     – value: full Qlib expression (string)

2. **Allowed Variables**
   – Only use variables prefixed with '\$': \$close, \$open, \$high, \$low, \$volume.

3. **Operators and Functions**
   – Use only CamelCase function names and operators (Qlib style).
   – Ensure all '(' and ')' are correctly paired.
More details about operators (Qlib reference):
QLIB_GENERATE_INSTRUCTION

4. **Expression Rules**
   – Always use function calls (e.g., Div(x, y)) instead of arithmetic symbols (+, −, ∗, /).
   – Parentheses must always be properly closed.
   – Do not use invalid or undefined functions.
   – Ensure all rolling window parameters are positive integers.
   – No missing or NaN parameters.

5. **Factor Naming**
   – Each factor must have a unique and descriptive CamelCase name (e.g., Momentum20, VolumeSpikeRatio).
   – Names should briefly describe the intuition of the factor.

6. **Diversity**
   – The N generated factors should cover different ideas: momentum, mean reversion, volatility, volume dynamics, cross−variable relations, etc.

### Universal Search Process (algorithm−agnostic)
A. **Plan**
   – Decompose candidates into roles: Core signal (trend/mean−revert/range), Normalizer (vol/range/level), Conditioner/Smoother (volume, Rank, Mean).
   – Sketch 3−−5 distinct blueprints before writing expressions.

B. **Generate**
   – For each blueprint, emit 1−−2 concise expressions (2−−4 ops main chain).
   – Align windows: prefer (short, long) pairs like (5−−10, 30−−60).
   – Encourage price $\oplus$ volume combinations in part of the set.

C. **Heuristic Evaluate (no backtest here)**
   – Prefer normalized forms: Div(core, Add(Std(..., L) or Mean(..., L), 1e−12)).
   – Smooth noisy cores: Mean(..., short L) or Rank(...).
   – Avoid brittle constructs (deep nesting, stacked Power, Rank−in−Rank).

D. **Refine / Prune**
   – If two candidates share the same skeleton, keep the cleaner one.

– Replace fragile pieces (e.g., raw Delta) with stabilized variants.

E. ∗∗Diversify∗∗
– Ensure at least one price+volume candidate and one range/volatility−scaled candidate.
– Vary operator families (Sub/Delta vs. Sub(\$high,\$low); Std vs. Mean; Rank vs. Mean smoothers).

F. ∗∗Validate (pre−output self−check)∗∗
– Count equals N.
– Only allowed variables/operators; windows in [2, 120].
– All denominators have epsilon.
– Depth $\leq$ 6; expressions are parsable Qlib style.
– Names unique; expressions materially different.

**Vanilla Output Example (no reasoning):**
```
{
  "generated": [
    {"name": "Momentum20", "expression": "Div(Sub(\$close, Ref(\$close, 20)), Add(Ref(\$close, 20), 1e−12))"},
    {"name": "VolumeVolatility10", "expression": "Std(\$volume, 10)"}
  ]
}
```

**With Reasoning Output Example:**
```
{
  "generated": [
    {"name": "Momentum20", "reason": "Measures 20−day price momentum as percentage change.", "expression": "Div(Sub(\$close, Ref(\$close, 20)), Add(Ref(\$close, 20), 1e−12))"},
    {"name": "VolumeVolatility10", "reason": "Captures short−term fluctuations in trading volume.", "expression": "Std(\$volume, 10)"}
  ]
}
```

Figure 11: Instruction system prompt used to guide LLMs in the factor searching task.

### D.3.1 CHAIN-OF-EXPERIENCE (CoE)

In the Chain-of-Experience (CoE) setup, we design prompts in Figure 12 to guide the model through iterative single-factor refinement. Each prompt presents the seed factor and its evaluation history, then clearly separates the improvement guidelines. This structure ensures the model can reason step by step, balance novelty with stability, and make small but targeted modifications. By constraining operations and output format, the CoE prompts enforce consistency while still allowing exploration, enabling the search process to steadily improve factor quality over successive rounds.

---

**Single-Factor Refinement Prompt (CoE Step)**

Role: You are an expert quantitative researcher refining one alpha factor. Your objective is to maximize IC (primary), then RankIC, then IR, while keeping the expression computable and not overly complex.

History (Seed → Current)
- r0 (seed): KMID => ($close - $open) / $open | IC=-0.009126, RankIC=-0.025256, ICIR=-0.052347
- Current best: KMID => ($close - $open) / $open | IC=-0.009126, RankIC=-0.025256, ICIR=-0.052347

Guidelines
- Allowed ops: Parameter jumps (e.g., 3/5/10/20/60/120/252), structural edits (add/remove ops; chain Rank, Corr, Ref, Delta, Power), stabilization (normalize by rolling Std/Mean or price range), regime gating with If(...) and MAs, volume conditioning, or exploration reset.
- Constraints: Only use variables \$close, \$open, \$high, \$low, \$volume. Keep novelty vs. history. Avoid overly long or redundant chains.
- Complexity: Small, incremental change per step—no huge rewrites.
- Output (STRICT): Return exactly one JSON object with keys name and expression.

Seed: KMID => (\$close − \$open) / \$open
Goal: Propose ONE improved variant of the current best OR a fresh alternative to increase IC (then RankIC, IR).

Figure 12: System prompt for the Single-Factor Refinement task (CoE step). It provides history of the seed factor and guidelines for incremental improvement.

Table 20: Example of a Chain-of-Experience (CoE) search trajectory. At each round, a new candidate factor is proposed. If its IC surpasses the best so far, it is promoted.

| Round | Candidate Expression | IC | Best Updated? |
|---|---|---|---|
| 1 | `Mean(Div(Sub($close, $open), Add(Sub($high, $open), 1e-12)), 10)` | 0.0122 | ✓ |
| 2 | `Rank(Mean(Div(Sub($close, Min($low, 5)), Add(Sub(Max($high, 5), Min($low, 5)), 1e-12)), 20), 120)` | 0.0048 | – |
| 3 | `Corr(Div(Sub($close, $open), Add($open, 1e-12)), $volume, 20)` | -0.0018 | – |
| 4 | `Mean(Div(Sub($close, $open), Add(Sub($high, $low), 1e-12)), 20)` | 0.0049 | – |
| 5 | `Rank(Div(Sub($close, Ref($close, 20)), Add(Std($close, 20), 1e-12)), 120)` | 0.0052 | – |
| 6 | `Rank(Mean(Div(Sub($close, $open), Add(Sub($high, $open), 1e-12)), 10), 120)` | -0.0013 | – |
| 7 | `Mean(Div(Sub($close, $open), Add(Std(Sub($close, $open), 20), 1e-12)), 5)` | 0.0165 | ✓ |
| 8 | `Mean(Mul(Div(Sub($close, $open), Add($open, 1e-12)), Div($volume, Add(Mean($volume, 20), 1e-12))), 10)` | 0.0089 | – |
| 9 | `Rank(Mean(Div(Sub($close, $open), Add(Std(Sub($close, $open), 20), 1e-12)), 5), 120)` | 0.0117 | – |

Table 20 presents an example trajectory of the Chain-of-Experience (CoE) search. Unlike Tree-of-Thought, which expands multiple candidates in parallel, CoE follows a sequential refinement process: at each round, a single new candidate factor is proposed and immediately evaluated. If its Information Coefficient (IC) surpasses the current best factor in the chain, it is promoted as the new reference point; otherwise, the trajectory continues without update.

In the example, the search begins in Round 1 with a factor based on the normalized difference between close and open prices, averaged over a 10-day window, which establishes the initial best IC (0.0122). Several subsequent candidates (Rounds 2–6) explore variations involving ranking, rolling correlations, or volatility adjustments, but none outperform the current best. In Round 7, however,

a new variant incorporating a volatility-scaled transformation achieves a higher IC (0.0165) and is promoted. The remaining rounds (8–9) generate additional modifications, yet they fall short of this new benchmark.

This example illustrates the step-by-step evolutionary nature of CoE search: only candidates that improve upon prior performance are retained as anchors for further refinement, producing a focused but narrower exploration compared to the branching strategy of ToT.

### D.3.2    TREE-OF-THOUGHT (ToT)

In the Tree-of-Thought (ToT) setup, we design prompts in Figure 13 to encourage branching exploration from a given seed factor. Instead of refining a single candidate step by step, the model expands one seed into six structurally diverse factors at once. The prompt explicitly enforces diversity constraints across factor families, such as momentum, mean reversion, range breakout, volatility, and volume scaling, while also ensuring syntactic validity through strict expression rules. This design allows the search process to cover a broader space of possible signals in parallel.

---

**Multi-Factor Expansion Prompt (ToT Step)**

Role: You are an expert quantitative researcher generating formulaic alpha factors.
Your task is to propose \emph{exactly six} candidate factors in JSON list format.

Context (Seed)
- Seed: KSFT2 => (2 * $close - $high - $low) / ($high - $low + 1e-12)
- You will perform a Tree-of-Thought expand step from the seed. Generate exactly 6 diverse candidates.

Strict Expression Rules
1) Allowed variables: $close, $open, $high, $low, $volume.
2) Allowed ops (Qlib style): Mean, Std, Corr, Rank, Ref, Sum, Sub, Add, Mul, Div, Max, Min, Abs, Power, Delta, Slope, Rsquare, Quantile, If, Greater, Gt.
3) Use function calls only (e.g., Div(x,y)); never raw arithmetic symbols.
4) Windows must be positive integers; operator depth within 2–4.
5) Every denominator in Div must be Add(<den>, 1e-12).
6) Use at most one Rank(...) per expression; avoid nested Rank.
7) Keep expressions compact; no triple Power.
ToT-Expand Diversity Constraints
• Branch across families: momentum, mean-reversion, range/breakout, volatility-scaled, volume-conditioned.
• No two candidates may share the same operator skeleton (ignoring constants/windows).
• Ensure >=30% token difference between any pair (operators + window values).
• Prefer (short, long) pairs: short in 5,10, long in 20,30,60.
• Optional gating: If(Gt($close, Mean($close,L)), A, B) with L in 20,60; keep A/B simple.

Output Format (STRICT)
Return only a JSON list of 6 objects, each with:
− "name": short CamelCase identifier
− "expression": valid Qlib−style expression string
Example:
[{"name":"Momentum20Adj","expression":"Div(Delta(\$close,20), Add(Std(\$close,20), 1e−12))"}]

---

Figure 13: System prompt for the Tree-of-Thought expansion task (ToT step). It specifies the seed factor, strict expression rules, and diversity constraints for generating six candidate factors.

In Table 21, we provide a concrete example of the Tree-of-Thought (ToT) search process applied to alpha factor exploration. The search begins with a given parent factor at Depth 1 (`KLOW2`), which is evaluated for its Information Coefficient (IC). From this parent, six candidate factors are generated through

different transformations, such as moving-average mean reversion (`MeanRevert5_30`), breakout signals (`RangeBreakout10`), or volume-conditioned variants (`VolumeTrendCross20_60`). Each candidate is scored by IC, and the top-3 candidates are selected for further expansion. At Depth 2, the selected parents from Depth 1 (`MeanRevert5_30` and `RangeBreakout10`) are each expanded into a new set of six candidates.

For instance, under the `MeanRevert5_30` branch, we observe refinements such as volatility-normalized mean reversion (`RangeNormalizedMR5_30`) and conditional volume adjustments (`VolumeCondMRIf60`). Similarly, the `RangeBreakout10` branch produces variations like volatility-scaled breakouts (`VolatilityScaledRange`) and momentum-driven breakouts (`MomentumBreak10_30`). Again, each is evaluated and top-performing candidates are retained.

This example illustrates the branching and pruning nature of ToT search: multiple candidates are generated in parallel, but only the most promising ones (based on IC) are expanded at the next depth. Compared with sequential refinement (CoE), ToT provides broader coverage of the search space while maintaining selectivity, enabling the discovery of more diverse and high-performing factors.

Table 21: Example Tree-of-Thought search path (depth up to 2). Each round generates 6 candidate factors and selects the top-3 by IC to expand in the next round.

| Name | Expression | IC | Selected |
|------|-----------|----|----------|
| **Depth 1: Parent = KLOW2** | | | |
| Parent: KLOW2 | `(Less($open,$close) - $low) / ($high - $low + 1e-12)` | -0.0223 | Parent |
| MeanRevert5_30 | `Div(Sub(Mean($close,5), Mean($close,30)), Add(Std($close,30), 1e-12))` | 0.0076 | Yes |
| RangeBreakout10 | `Div(Sub($close, Max($high,10)), Add(Sub(Max($high,10), Min($low,10)), 1e-12))` | 0.0084 | Yes |
| VolAwareSlope30 | `Div(Slope($close,30), Add(Mean(Abs(Sub($close, Ref($close,1))),30), 1e-12))` | 0.0060 | |
| MomentumRelVolume | `Div(Delta($close,5), Add(Mean($volume,30), 1e-12))` | 0.0062 | |
| VolumeTrendCross20_60 | `If(Gt(Mean($volume,20), Mean($volume,60)), Div(Delta($close,5), Add(Mean($close,5), 1e-12)), Div(Sub($close, Ref($close,5)), Add(Std($close,5), 1e-12)))` | 0.0075 | Yes |
| MomentumCorrel10withVolume30 | `Corr(Delta($close,10), $volume,30)` | -0.0031 | |
| **Depth 2: Parent = MeanRevert5_30 (selected from Depth 1)** | | | |
| Parent: MeanRevert5_30 | `Div(Sub(Mean($close,5), Mean($close,30)), Add(Std($close,30), 1e-12))` | 0.0076 | Parent |

*Continued on next page*

Table 21: Example Tree-of-Thought search path (depth up to 2). Each round generates 6 candidate factors and selects the top-3 by IC to expand in the next round. (Continued)

| Name | Expression | IC | Selected |
|------|-----------|-----|----------|
| RangeNormalizedMR5_30 | `Div(Sub(Mean($close,5), Mean($close,30)), Add(Mean(Sub($high, $low), 30), 1e-12))` | 0.0083 | Yes |
| VolumeCondMRIf60 | `If(Gt($close, Mean($close, 60)), Div(Sub(Mean($close,5), Mean($close,30)), Add(Mean($volume,30), 1e-12)), Div(Sub(Mean($close,30), Mean($close,5)), Add(Mean($volume,30), 1e-12)))` | -0.0026 | |
| MeanRevert10_60 | `Div(Sub(Mean($close,10), Mean($close,60)), Add(Std($close,60), 1e-12))` | -0.0011 | |
| MomentumDeltaAbs10_30 | `Div(Delta($close,10), Add(Mean(Abs(Delta($close,1)),30), 1e-12))` | 0.0066 | Yes |
| SlopeMeanRevert5_30 | `Div(Slope(Mean($close,5), 30), Add(Std($close,30), 1e-12))` | 0.0027 | |
| VolatilityCorrMomentum | `Mul(Div(Delta($close,5), Add(Std($close,30),1e-12)), Corr($close, $volume, 30))` | 0.0110 | Yes |
| **Depth 2: Parent = RangeBreakout10 (selected from Depth 1)** | | | |
| Parent: RangeBreakout10 | `Div(Sub($close, Max($high,10)), Add(Sub(Max($high,10), Min($low,10)), 1e-12))` | 0.0084 | Parent |
| VolatilityScaledRange | `Div(Sub($close, Max($high,10)), Add(Std(Sub($high, $low), 20), 1e-12))` | 0.0115 | Yes |
| VolumeCorrMomentum | `Corr(Delta($close,5), Delta($volume,5), 10)` | 0.0049 | |
| IfCloseAboveMeanCloseVolMom | `If(Gt($close, Mean($close,20)), Div(Delta($close,5), Add(Mean($volume,30), 1e-12)), Div(Delta($close,5), Add(Std($close,10), 1e-12)))` | 0.0068 | Yes |
| MeanReversionRange30_60 | `Div(Sub(Mean($high,30), $close), Add(Sub(Max($high,60), Min($low,60)), 1e-12))` | -0.0108 | |

Continued on next page

Table 21: Example Tree-of-Thought search path (depth up to 2). Each round generates 6 candidate factors and selects the top-3 by IC to expand in the next round. (Continued)

| Name | Expression | IC | Selected |
|------|------------|-----|----------|
| RangeBreakoutVolAdj | `Div(Sub($close,`
`Max($high,10)),`
`Add(Mul(Sub(Max($high,10),`
`Min($low,10)),`
`Mean($volume,20)), 1e-12))` | -0.0020 | |
| MomentumBreak10_30 | `Div(Sub($close,`
`Max($close,10)),`
`Add(Std($close,30),`
`1e-12))` | 0.0105 | Yes |

### D.3.3 EVOLUTIONARY ALGORITHM (EA)

The EA (Evolutionary Algorithm) search results in Table 22 highlight how factors evolve across rounds through selection, mutation, and crossover. In the last round, factors like SUMD, SUMP, and VMA achieved the highest IC values around 0.013, forming the parent pool. From this pool, mutation produced improved variants such as SlopeQuality9 (IC = 0.021) and DirectionalBalanceNorm7 (IC = 0.017), showing clear performance gains. Meanwhile, crossover generated diverse candidates like UpEffTrend5 and MomentumVolDampen5, which also delivered competitive IC scores (0.011–0.012).

Table 22: Example EA search path showing last-round selected factors and next-round candidates generated via mutation and crossover.

| Name | Expression | IC |
|------|------------|-----|
| **Last Round Factors** | | |
| SUMD | `(Sum(Greater($close-Ref($close,1),0),`
`5)-Sum(Greater(Ref($close,1)-$close,0),`
`5))/(Sum(Abs($close-Ref($close,1)),`
`5)+1e-12)` | 0.013 |
| SUMP | `Sum(Greater($close-Ref($close,1),0),`
`5)/(Sum(Abs($close-Ref($close,1)),`
`5)+1e-12)` | 0.013 |
| VMA | `Mean($volume, 5)/($volume+1e-12)` | 0.012 |
| BETA | `Slope($close, 5)/$close` | 0.011 |
| CNTD | `Mean($close>Ref($close,1),`
`5)-Mean($close<Ref($close,1), 5)` | 0.010 |
| VSTD | `Std($volume, 5)/($volume+1e-12)` | 0.009 |
| IMXD | `(IdxMax($high, 5) - IdxMin($low, 5)) / 5` | 0.009 |
| CORR | `Corr($close, Log($volume+1), 5)` | 0.009 |
| LOW | `Min($low, 5)/$close` | 0.008 |
| CNTP | `Mean($close>Ref($close,1), 5)` | 0.007 |
| IMAX | `IdxMax($high, 5)/5` | 0.007 |
| RSQR | `Rsquare($close, 5)` | 0.007 |
| **Next Round (Mutation)** | | |
| **SlopeQuality9** | `Mul(Div(Slope($close, 9),`
`Add(Std($close, 60), 1e-12)),`
`Rsquare($close, 9))` | 0.021 |

Continued on next page

Table 22: Example EA search path showing last-round selected factors and next-round candidates generated via mutation and crossover. (Continued)

| Name | Expression | IC |
|---|---|---|
| **DirectionalBalanceNorm7** | `Div(Sub(Sum(Greater(Sub($close, Ref($close, 1)), 0), 7), Sum(Greater(Sub(Ref($close, 1), $close), 0), 7)), Add(Std(Sub($close, Ref($close, 1)), 30), 1e-12))` | 0.017 |
| SignedMomentumNorm9x30 | `Div(Mean(Sign(Sub($close, Ref($close, 1))), 9), Add(Std(Sub($close, Ref($close, 1)), 30), 1e-12))` | 0.005 |
| **LowToEmaRatio7** | `Div(Min($low, 7), Add(EMA($close, 7), 1e-12))` | 0.013 |
| **Next Round (Crossover)** | | |
| **UpEffTrend5** | `Mul(Div(Sum(Greater(Sub($close, Ref($close, 1)), 0), 5), Add(Sum(Abs(Sub($close, Ref($close, 1))), 5), 1e-12)), Rsquare($close, 5))` | 0.011 |
| TrendStrengthRsq5 | `Mul(Div(Slope($close, 5), Add(Mean(Abs(Sub($close, Ref($close, 1))), 5), 1e-12)), Rsquare($close, 5))` | 0.003 |
| **MomentumVolDampen5** | `Mul(Div(Mean(Sub($close, Ref($close, 1)), 5), Add(Mean(Abs(Sub($close, Ref($close, 1))), 5), 1e-12)), Div(Mean($volume, 5), Add($volume, 1e-12)))` | 0.012 |
| LowReboundVolWeighted5 | `Mul(Div(Sub($close, Min($low, 5)), Add(Sum(Abs(Sub($close, Ref($close, 1))), 5), 1e-12)), Div(Mean($volume, 5), Add($volume, 1e-12)))` | 0.005 |
| CorrCntd5 | `Mul(Corr($close, Log(Add($volume, 1)), 5), Sub(Mean(Greater(Sub($close, Ref($close, 1)), 0), 5), Mean(Greater(Sub(Ref($close, 1), $close), 0), 5)))` | -0.010 |
| DirectionVsVolumeStd5 | `Div(Sub(Mean(Greater(Sub($close, Ref($close, 1)), 0), 5), Mean(Greater(Sub(Ref($close, 1), $close), 0), 5)), Add(Div(Std($volume, 5), Add($volume, 1e-12)), 1e-12))` | 0.004 |

# E  ADDITIONAL EXPERIMENT RESULTS

In this section, we present detailed evaluation results and analyses from AlphaBench that were not included in the main text due to space constraints.

## E.1  ADDITIONAL RESULTS OF GENERATION TASK

Tables 23 and 25 provide a detailed breakdown of reliability and accuracy across difficulty levels for the Text2Alpha and Directional Mining tasks. The results reveal several notable trends.Table 22 presents the per-difficulty breakdown of reliability and accuracy for the Text2Alpha task. Values are shown as Vanilla/CoT pairs across Easy, Medium, and Hard difficulty levels. While table 24 shows the corresponding per-difficulty results for the Directional Mining task, following the same Vanilla/CoT evaluation framework."

Table 23: Text2Alpha: per-difficulty Reliability/Accuracy (values shown as Vanilla/CoT).

| Model | Easy | | Medium | | Hard | |
|---|---|---|---|---|---|---|
| | Reliability | Accuracy | Reliability | Accuracy | Reliability | Accuracy |
| GPT-5 | 1.00/- | 0.99/- | 1.00/- | 0.68/- | 0.99/- | 0.10/- |
| Gemini-2.5-Pro | 0.99/- | 0.78/- | 0.97/- | 0.50/- | 0.99/- | 0.09/- |
| Gemini-2.5-Flash | 1.00/1.00 | 0.99/0.98 | 0.99/0.98 | 0.73/0.74 | 0.98/0.96 | 0.14/0.20 |
| GPT-4.1-Mini | 1.00/1.00 | 0.98/0.99 | 0.97/0.95 | 0.63/0.57 | 0.96/0.87 | 0.06/0.06 |
| DeepSeek-V3 | 0.95/1.00 | 0.74/0.65 | 0.91/0.99 | 0.43/0.44 | 0.89/0.97 | 0.03/0.02 |
| LLaMA3.1-70b-Instruct | 1.00/1.00 | 0.93/0.95 | 0.99/0.98 | 0.57/0.59 | 0.97/0.96 | 0.03/0.05 |
| DeepSeek-R1-Distill-Qwen-32B | 0.46/0.55 | 0.46/0.33 | 0.34/0.65 | 0.14/0.23 | 0.37/0.48 | 0.02/0.05 |
| Qwen2.5-14B-Instruct | 0.99/0.92 | 0.93/0.95 | 0.94/0.65 | 0.46/0.61 | 0.85/0.36 | 0.08/0.18 |
| LLaMA3.1-8b-Instruct | 0.96/0.88 | 0.68/0.82 | 0.88/0.83 | 0.16/0.19 | 0.94/0.74 | 0.02/0.03 |
| Gemini-1.5-Flash-8b | 0.96/0.99 | 0.85/0.86 | 0.98/0.94 | 0.37/0.47 | 0.93/0.97 | 0.05/0.04 |

Table 24: Text2Alpha: per-difficulty Reliability/Accuracy for LLM coder

| Model | Easy | | Medium | | Hard | |
|---|---|---|---|---|---|---|
| | Reliability | Accuracy | Reliability | Accuracy | Reliability | Accuracy |
| Qwen2.5-Coder-7B-Instruct | 0.98 | 0.84 | 0.80 | 0.42 | 0.64 | 0.05 |
| Codellama-70B-Instruct | 1.00 | 0.94 | 0.99 | 0.57 | 0.98 | 0.04 |
| Qwen3-Coder-480B-A35B-Instruct-Turbo | 1.00 | 0.98 | 0.94 | 0.62 | 0.95 | 0.05 |

Table 25: Directional Mining: per-difficulty Reliability/Accuracy (values shown as Vanilla/CoT).

| Model | Easy | | Medium | | Hard | |
|---|---|---|---|---|---|---|
| | Reliability | Accuracy | Reliability | Accuracy | Reliability | Accuracy |
| GPT-5 | 1.00/- | 0.72/- | 1.00/- | 0.65/- | 1.00/- | 0.42/- |
| Gemini-2.5-Pro | 1.00/- | 0.62/- | 0.95/- | 0.45/- | 0.95/- | 0.32/- |
| Gemini-2.5-Flash | 0.99/0.98 | 0.66/0.73 | 1.00/1.00 | 0.62/0.65 | 0.98/1.00 | 0.46/0.40 |
| GPT-4.1-Mini | 0.91/0.91 | 0.56/0.54 | 0.88/0.95 | 0.46/0.42 | 0.88/0.93 | 0.15/0.23 |
| DeepSeek-V3 | 0.82/0.96 | 0.42/0.36 | 0.94/0.97 | 0.26/0.37 | 0.93/0.93 | 0.12/0.20 |
| LLaMA3.1-70b-Instruct | 0.91/0.93 | 0.41/0.43 | 0.88/0.89 | 0.26/0.33 | 0.95/0.90 | 0.23/0.56 |
| DeepSeek-R1-Distill-Qwen-32B | 0.30/0.71 | 0.15/0.12 | 0.26/0.54 | 0.06/0.06 | 0.35/0.57 | 0.10/0.09 |
| Qwen2.5-14B-Instruct | 0.76/0.74 | 0.41/0.61 | 0.60/0.60 | 0.21/0.31 | 0.62/0.35 | 0.11/0.29 |
| LLaMA3.1-8b-Instruct | 0.97/0.84 | 0.11/0.13 | 0.94/0.94 | 0.07/0.16 | 0.95/0.83 | 0.12/0.18 |
| Gemini-1.5-Flash-8b | 0.93/0.96 | 0.33/0.35 | 0.95/0.95 | 0.16/0.23 | 0.93/0.87 | 0.18/0.15 |

Models generally exhibit very high reliability in the easy setting, indicating that generating syntactically valid expressions is not a major challenge. However, accuracy declines substantially as difficulty

Table 26: Directional Mining: per-difficulty Reliability/Accuracy for LLM coder.

| Model | Easy | | Medium | | Hard | |
|---|---|---|---|---|---|---|
| | Reliability | Accuracy | Reliability | Accuracy | Reliability | Accuracy |
| Qwen2.5-Coder-7B-Instruct | 0.74 | 0.33 | 0.71 | 0.15 | 0.75 | 0.16 |
| Codellama-70B-Instruct | 0.88 | 0.37 | 0.89 | 0.31 | 0.97 | 0.36 |
| Qwen3-Coder-480B-A35B-Instruct-Turbo | 0.96 | 0.55 | 0.95 | 0.34 | 0.98 | 0.22 |

increases. GPT-5, for example, maintains reliability near 1.0 on all levels, yet its accuracy drops from 0.99 in easy Text2Alpha to only 0.10 in the hard split. This pattern illustrates that producing structurally correct formulas does not necessarily translate into capturing the intended semantics under more complex instructions.

Table 27: Pass rates by result model and judge model across tasks and difficulty (merged Result Model cells; rounded to 2 decimals).

| Result Model | Judge Model | Text2Alpha | | | Directional Mining | | |
|---|---|---|---|---|---|---|---|
| | | Easy | Medium | Hard | Easy | Medium | Hard |
| DeepSeek-V3 | DeepSeek-V3 | 0.74 | 0.45 | 0.03 | 0.34 | 0.25 | 0.07 |
| | Gemini-2.5-Flash | 0.74 | 0.43 | 0.03 | 0.42 | 0.26 | 0.12 |
| | GPT-4.1-Mini | 0.72 | 0.44 | 0.03 | 0.43 | 0.38 | 0.29 |
| Gemini-2.5-Flash | DeepSeek-V3 | 1.00 | 0.70 | 0.14 | 0.57 | 0.40 | 0.14 |
| | Gemini-2.5-Flash | 0.99 | 0.73 | 0.14 | 0.66 | 0.62 | 0.46 |
| | GPT-4.1-Mini | 0.98 | 0.74 | 0.12 | 0.75 | 0.72 | 0.66 |
| GPT-4.1-Mini | DeepSeek-V3 | 0.98 | 0.62 | 0.04 | 0.46 | 0.35 | 0.06 |
| | Gemini-2.5-Flash | 0.98 | 0.63 | 0.06 | 0.56 | 0.46 | 0.15 |
| | GPT-4.1-Mini | 0.95 | 0.65 | 0.04 | 0.68 | 0.54 | 0.36 |
| Gemini-2.5-Pro | DeepSeek-V3 | 0.78 | 0.52 | 0.08 | 0.53 | 0.31 | 0.09 |
| | Gemini-2.5-Flash | 0.78 | 0.50 | 0.09 | 0.62 | 0.45 | 0.32 |
| | GPT-4.1-Mini | 0.77 | 0.50 | 0.07 | 0.72 | 0.56 | 0.35 |
| GPT-5 | DeepSeek-V3 | 0.99 | 0.66 | 0.10 | 0.62 | 0.45 | 0.17 |
| | Gemini-2.5-Flash | 0.99 | 0.68 | 0.10 | 0.72 | 0.65 | 0.42 |
| | GPT-4.1-Mini | 0.95 | 0.67 | 0.07 | 0.82 | 0.78 | 0.62 |
| LLaMA3.1-70b-Instruct | DeepSeek-V3 | 0.98 | 0.61 | 0.06 | 0.37 | 0.19 | 0.09 |
| | Gemini-2.5-Flash | 0.93 | 0.57 | 0.03 | 0.41 | 0.26 | 0.23 |
| | GPT-4.1-Mini | 0.94 | 0.60 | 0.04 | 0.48 | 0.33 | 0.33 |

Performance differences across models are also evident. Gemini-2.5-Flash and GPT-4.1-Mini achieve comparatively stronger accuracy in medium and hard cases, whereas smaller open-weight models such as LLaMA3.1-8B and Qwen2.5-14B degrade more severely with increasing difficulty. The distilled DeepSeek-R1-Qwen-32B is consistently weaker in both reliability and accuracy, highlighting the trade-offs introduced by compression. The impact of Chain-of-Thought (CoT) prompting varies by model and task. In Text2Alpha, CoT offers moderate improvements for models like Gemini-2.5-Flash and Qwen2.5-14B, but has little effect for stronger baselines. In Directional Mining, the benefits are more pronounced, particularly in medium and hard settings. LLaMA3.1-70B, for instance, improves its hard-case accuracy from 0.23 to 0.56 when using CoT. These results suggest that CoT is more effective when reasoning about directional constraints, while its value is limited for relatively straightforward formula construction.

To examine potential biases introduced by different judge models, we conducted additional experiments in which the same set of generation outputs was evaluated by three distinct models, the results are shown in Table 27. For the *Text2Alpha* task, the pass rates obtained under different judges were broadly consistent, indicating that evaluation outcomes are relatively robust to the choice of judge. In contrast, the *Directional Mining* task exhibited much larger variation, especially on the hard split. This discrepancy likely reflects differences in the semantic knowledge that judge models possess regarding directional signals and label conventions.

Given these variations, the reported results in the main paper adopt Gemini-2.5-Flash as the judge, which produced outcomes generally situated between the stricter DeepSeek-V3 and the more lenient GPT-4.1-Mini. This choice provides a balanced perspective and reduces the risk of overstating or understating performance due to model-specific biases in judgment.

### E.2  ADDITIONAL RESULT OF EVALUATION TASK

For the ranking task, the expected Top-$K$ precision under random selection is determined by the positive ratio in each candidate set. In our splits, the expected values are approximately 0.33 for the $(10, 3)$ scenario and 0.25 for both $(20, 5)$ and $(40, 10)$. These values serve as practical reference lines for interpreting Table 28. Across market regimes (Overall, Bear, Bull), most models perform near these random baselines. In the Overall split, Top-$K$ precision typically ranges from 0.20 to 0.38 for $(10, 3)$, and from 0.17 to 0.34 for both $(20, 5)$ and $(40, 10)$, reflecting a limited ability to prioritize truly strong factors. A few models perform slightly better than random—such as Llama-3.1-70B-Instruct, which achieves precision between 0.28 and 0.30 in the $(10, 3)$ Overall scenario, and 0.28 in the $(20, 5)$—but the improvements are modest and inconsistent across regimes. NDCG@K values follow a similar trend: reranking quality is poor (with typical values ranging from 0.20 to 0.30), yet the improvements over random selection remain minor and unstable. In the Bull market regime, a few models perform marginally better (e.g., GPT-4.1-Mini reaches 0.37 and 0.38 for $(10, 3)$), suggesting that trend-friendly conditions align better with the simple heuristics embedded in the models, though these advantages largely disappear in Bear market conditions.

Table 28: Performance comparison under three market environments (Neutral, Bear, Bull). Each cell reports *Origin / CoT*. Bold indicates the best per-scenario value in each column (Origin and CoT considered separately). (M,N) means select top-N from M candidates.

| Scenario | Model | (10,3) | | (20,5) | | (40,10) | |
| --- | --- | --- | --- | --- | --- | --- | --- |
| | | Prec@k | NDCG@k | Prec@k | NDCG@k | Prec@k | NDCG@k |
| Random | - | 0.33 | - | 0.25 | - | 0.25 | - |
| Neutral | GPT-5 | 0.27 / - | 0.22 / - | 0.18 / - | 0.09 / - | 0.17 / - | 0.06 / - |
| | Gemini-2.5-pro | 0.27 / - | 0.23 / - | 0.19 / - | 0.13 / - | 0.22 / - | 0.12 / - |
| | GPT-4.1-mini | 0.21 / 0.17 | 0.20 / 0.17 | 0.17 / 0.20 | 0.20 / 0.20 | 0.26 / 0.25 | **0.28** / 0.19 |
| | Gemini-2.5-Flash | 0.23 / 0.13 | 0.21 / 0.10 | 0.22 / 0.11 | 0.13 / 0.08 | 0.23 / 0.13 | 0.17 / 0.08 |
| | DeepSeek-V3 | 0.25 / 0.15 | 0.26 / 0.16 | 0.18 / 0.13 | 0.14 / 0.11 | 0.14 / 0.17 | 0.05 / 0.10 |
| | LLaMA-3.1-70b-Instruct | 0.28 / **0.30** | 0.25 / **0.33** | **0.28** / **0.27** | **0.26** / **0.21** | **0.27** / 0.25 | 0.20 / **0.26** |
| | DeepSeek-R1-Distill-Qwen-32B | 0.29 / 0.21 | 0.29 / 0.21 | 0.18 / 0.21 | 0.15 / 0.18 | 0.13 / 0.14 | 0.10 / 0.10 |
| | LLaMA-3.1-8b-Instruct | 0.27 / 0.26 | 0.25 / 0.28 | 0.22 / 0.25 | 0.17 / 0.16 | 0.20 / **0.28** | 0.06 / 0.12 |
| | Gemini-1.5-Flash-8b | 0.27 / **0.30** | 0.26 / 0.27 | 0.23 / 0.22 | 0.14 / 0.13 | 0.24 / 0.23 | 0.11 / 0.08 |
| | Qwen2.5-14B-Instruct | **0.32** / 0.27 | **0.34** / 0.24 | 0.21 / 0.20 | 0.17 / 0.13 | 0.23 / 0.21 | 0.15 / 0.16 |
| Bear | GPT-5 | 0.23 / - | 0.16 / - | 0.16 / - | 0.11 / - | 0.12 / - | 0.08 / - |
| | Gemini-2.5-pro | 0.23 / - | 0.21 / - | 0.21 / - | 0.15 / - | 0.16 / - | 0.15 / - |
| | GPT-4.1-mini | 0.18 / 0.22 | 0.15 / 0.18 | 0.16 / 0.19 | 0.12 / 0.15 | 0.16 / 0.16 | 0.12 / 0.17 |
| | Gemini-2.5-Flash | 0.26 / 0.12 | 0.21 / 0.10 | 0.16 / 0.09 | 0.11 / 0.04 | 0.18 / 0.11 | 0.16 / 0.06 |
| | DeepSeek-V3 | 0.22 / 0.17 | 0.18 / 0.13 | 0.12 / 0.13 | 0.08 / 0.10 | 0.11 / 0.19 | 0.06 / 0.09 |
| | LLaMA-3.1-70b-Instruct | 0.27 / 0.27 | **0.29** / **0.29** | 0.24 / 0.21 | **0.23** / **0.18** | **0.24** / 0.21 | **0.23** / **0.30** |
| | DeepSeek-R1-Distill-Qwen-32B | 0.20 / 0.22 | 0.19 / 0.21 | 0.19 / 0.19 | 0.14 / 0.14 | 0.16 / 0.17 | 0.13 / 0.17 |
| | LLaMA-3.1-8b-Instruct | **0.31** / 0.29 | 0.27 / 0.26 | **0.26** / 0.20 | 0.13 / 0.17 | 0.22 / 0.21 | 0.11 / 0.10 |
| | Gemini-1.5-Flash-8b | 0.30 / **0.37** | 0.25 / **0.29** | 0.24 / **0.26** | 0.14 / 0.16 | 0.23 / **0.28** | 0.21 / 0.23 |
| | Qwen2.5-14B-Instruct | 0.21 / 0.25 | 0.21 / 0.22 | 0.19 / 0.18 | 0.14 / 0.15 | 0.21 / 0.16 | 0.21 / 0.13 |
| Bull | GPT-5 | 0.35 / - | 0.34 / - | **0.34** / - | 0.26 / - | **0.34** / - | 0.14 / - |
| | Gemini-2.5-pro | 0.31 / - | 0.31 / - | 0.30 / - | 0.25 / - | 0.25 / - | 0.12 / - |
| | GPT-4.1-mini | **0.37** / **0.38** | 0.32 / 0.34 | 0.32 / **0.30** | 0.26 / 0.21 | 0.28 / **0.29** | 0.14 / 0.17 |
| | Gemini-2.5-Flash | **0.37** / 0.16 | 0.35 / 0.13 | 0.30 / 0.19 | **0.27** / 0.15 | **0.34** / 0.23 | 0.15 / 0.12 |
| | DeepSeek-V3 | 0.22 / 0.15 | 0.20 / 0.14 | 0.22 / 0.27 | 0.14 / 0.20 | 0.23 / 0.17 | 0.15 / 0.08 |
| | LLaMA-3.1-70b-Instruct | 0.36 / 0.28 | 0.35 / 0.31 | 0.28 / 0.28 | 0.26 / **0.27** | 0.30 / 0.28 | 0.18 / 0.18 |
| | DeepSeek-R1-Distill-Qwen-32B | 0.28 / 0.31 | 0.25 / 0.30 | 0.23 / 0.22 | 0.17 / 0.15 | 0.16 / 0.15 | 0.10 / 0.12 |
| | LLaMA-3.1-8b-Instruct | 0.28 / 0.34 | 0.27 / **0.34** | 0.28 / 0.27 | 0.19 / 0.20 | 0.26 / 0.23 | 0.10 / 0.12 |
| | Gemini-1.5-Flash-8b | 0.27 / 0.24 | 0.26 / 0.22 | 0.29 / 0.29 | 0.21 / 0.22 | 0.27 / 0.25 | 0.16 / 0.13 |
| | Qwen2.5-14B-Instruct | 0.36 / 0.33 | **0.38** / **0.34** | 0.27 / 0.28 | 0.25 / 0.20 | 0.27 / **0.29** | **0.19** / **0.22** |

The impact of Chain-of-Thought (CoT) prompting is highly variable. In some cases, it reduces ranking quality— for example, `Gemini-2.5-Flash` on Bull `10 pick 3` drops from 0.37 to 0.16—indicating that additional reasoning can sometimes detract from key cues. In other cases, particularly with smaller models or in Bear markets, CoT provides modest improvements. A notable example is `Gemini-1.5-Flash-8B`, which improves to 0.37 on `10 pick 3` in the Bear market. Overall, CoT does not consistently enhance ranking beyond random-adjacent baselines and often introduces additional variance.

Table 29: Scoring metrics under three market environments (Neutral, Bear, Bull). Each cell reports *Origin* and *CoT*. Bold indicates the per-scenario best value for each metric (higher is better for Accuracy of signal; lower is better for MAE metrics).

| Scenario | Model | Signal | | Performance | | Robustness | | WinRate | | Skewness | |
|---|---|---|---|---|---|---|---|---|---|---|---|
| | | Origin | CoT | Origin | CoT | Origin | CoT | Origin | CoT | Origin | CoT |
| Neutral | GPT-5 | 0.25 | - | 2.17 | - | 1.30 | - | 1.94 | - | 1.66 | - |
| | Gemini-2.5-Pro | **0.32** | - | 2.18 | - | **1.23** | - | 1.99 | - | 1.62 | - |
| | GPT-4.1-mini | 0.20 | 0.18 | 1.96 | 1.90 | 1.25 | 1.31 | 1.75 | 1.67 | 1.48 | 1.55 |
| | Gemini-2.5-Flash | 0.27 | **0.32** | 2.09 | 1.99 | 1.38 | 1.30 | 1.86 | 1.76 | 1.63 | 1.61 |
| | DeepSeek-V3 | 0.15 | 0.17 | 2.08 | 1.84 | **1.23** | 1.40 | 1.89 | **1.65** | 1.50 | 1.44 |
| | LLaMA-3.1-70b-Instruct | 0.19 | 0.21 | **1.85** | **1.83** | 1.48 | 1.48 | **1.69** | 1.69 | 1.65 | 1.62 |
| | DeepSeek-R1-Distill-Qwen-32B | 0.19 | 0.18 | 2.07 | 1.95 | **1.23** | **1.14** | 1.86 | 1.83 | 1.58 | 1.50 |
| | LLaMA-3.1-8b-Instruct | 0.28 | 0.26 | 2.03 | 1.88 | 1.24 | **1.14** | 1.85 | 1.76 | 1.62 | 1.57 |
| | Gemini-1.5-Flash-8b | 0.22 | 0.24 | 2.04 | 1.87 | 1.25 | 1.35 | 1.86 | 1.69 | **1.35** | **1.39** |
| Bear | GPT-5 | 0.35 | - | 2.19 | - | 1.39 | - | 1.91 | - | 1.75 | - |
| | Gemini-2.5-Pro | **0.40** | - | 2.06 | - | 1.48 | - | 1.96 | - | **1.40** | - |
| | GPT-4.1-mini | 0.25 | 0.26 | 2.05 | 1.95 | 1.32 | 1.32 | 1.82 | 1.79 | 1.57 | 1.67 |
| | Gemini-2.5-Flash | 0.38 | **0.37** | 2.05 | 2.09 | 1.51 | 1.47 | 1.94 | 1.89 | 1.69 | 1.68 |
| | DeepSeek-V3 | 0.15 | 0.21 | 2.12 | 1.94 | **1.26** | 1.27 | 1.87 | 1.78 | 1.68 | 1.67 |
| | LLaMA-3.1-70b-Instruct | 0.25 | 0.26 | **1.82** | **1.84** | 1.44 | 1.43 | **1.67** | 1.75 | 1.74 | 1.69 |
| | DeepSeek-R1-Distill-Qwen-32B | 0.33 | 0.31 | 2.14 | 2.11 | 1.32 | 1.27 | 1.96 | 1.94 | 1.55 | 1.56 |
| | LLaMA-3.1-8b-Instruct | 0.28 | 0.31 | 2.29 | 1.93 | 1.37 | **1.26** | 2.13 | 1.81 | 1.62 | 1.50 |
| | Gemini-1.5-Flash-8b | 0.36 | 0.35 | 2.05 | 1.91 | 1.27 | 1.44 | 1.81 | **1.72** | **1.40** | **1.38** |
| Bull | GPT-5 | **0.37** | - | 1.27 | - | 1.52 | - | 1.29 | - | 1.59 | - |
| | Gemini-2.5-Pro | 0.36 | - | 1.31 | - | 1.65 | - | 1.36 | - | 1.73 | - |
| | GPT-4.1-mini | 0.25 | 0.17 | 1.33 | 1.41 | 1.44 | 1.47 | 1.40 | 1.46 | 1.53 | 1.58 |
| | Gemini-2.5-Flash | 0.31 | **0.30** | 1.27 | 1.28 | 1.61 | 1.60 | 1.39 | 1.37 | 1.59 | 1.59 |
| | DeepSeek-V3 | 0.17 | 0.14 | **1.25** | 1.35 | 1.40 | 1.40 | **1.27** | 1.42 | 1.60 | 1.64 |
| | LLaMA-3.1-70b-Instruct | 0.24 | 0.24 | 1.42 | 1.41 | 1.47 | 1.48 | 1.50 | 1.42 | 1.76 | 1.69 |
| | DeepSeek-R1-Distill-Qwen-32B | 0.21 | 0.20 | 1.31 | **1.26** | 1.41 | 1.46 | 1.30 | **1.25** | **1.39** | 1.46 |
| | LLaMA-3.1-8b-Instruct | 0.21 | 0.22 | 1.48 | 1.51 | **1.32** | **1.34** | 1.41 | 1.52 | 1.66 | 1.72 |
| | Gemini-1.5-Flash-8b | 0.18 | 0.19 | 1.33 | 1.40 | **1.32** | 1.42 | 1.42 | 1.50 | 1.42 | **1.45** |

Turning to the scoring task (Table 29), signal-direction accuracy (`Acc Signal`) typically falls in the 0.20–0.40 range, close to random chance under our label design. This suggests that models struggle to reliably infer the correct sign of performance from textual or structural descriptions alone. CoT rarely influences this outcome in a systematic way. For the error-based metrics, CoT occasionally reduces magnitude errors slightly—for instance, `Gemini-2.5-Flash` improves its `MAE-Performance` from 2.09 to 1.99 and `MAE-Robustness` from 1.38 to 1.30 on the Overall split. Similar small improvements appear for `DeepSeek-V3`. However, these changes are narrow in comparison to the overall error levels, and other models show no improvement or even worse results. This indicates that CoT alone does not offer robust calibration for evaluation-style scoring.

The near-random performance can be attributed to two primary challenges. The evaluation task involves mapping formulaic structure to out-of-sample quality under different market conditions. However, without access to data or backtesting, language-based heuristics are poor substitutes. Additionally, regime sensitivity plays a significant role: Bear and Bull markets emphasize different signal semantics, and naive momentum or mean-reversion narratives may sometimes be helpful in Bull markets but fail to generalize. The inconsistent CoT effects suggest that producing longer verbal reasoning is not a substitute for statistical evidence. A more effective approach may involve

coupling reasoning with empirical retrieval, such as including summaries of historical IC distributions by regime or using execution-style prompts that encode stylized financial facts. Finally, given that observed differences are small and results tend to cluster around random, it is important to report multiple metrics—Top-$K$ precision, NDCG@k, `Acc Signal`, and MAE families—rather than relying on a single indicator, which could overstate model performance in this task.

### E.3 ADDITIONAL RESULT OF ATOMIC EVALUATION TASK

The following tables show full metrics in both *Signal Classification* and *Pairwise Selection* across the CSI300 and SP500 markets. Across both markets, the binary *Signal Classification* results in Table 30 show that all models struggle to reliably identify noisy factors when noise is treated as the positive class. Accuracy often sits only slightly above random, but the more informative metrics: F1, precision, and especially recall on the noise class, remain modest, indicating that the models frequently fail to flag truly noisy factors and instead keep them as "useful" signals. CoT prompting does not systematically help: for some models it slightly reshapes the precision–recall trade-off, but it also often reduces recall on noise, suggesting that adding reasoning steps makes the decision rule more rigid rather than more discriminative. Among all systems, GPT-5 is the only one that reaches consistently non-trivial performance on SP500 (with clearly above-random F1 and a more balanced precision–recall profile), while other models behave much closer to noisy or weak heuristics in this classification setting.

Table 30: Binary *Signal Classification* factor evaluation performance under Origin vs. CoT prompting for CSI300 and SP500. Where *noise* signal is positive label.

| Model | CSI300 | | | | | | | | SP500 | | | | | | | |
|---|---|---|---|---|---|---|---|---|---|---|---|---|---|---|---|---|
| | Accuracy | | F1 | | Precision | | Recall | | Accuracy | | F1 | | Precision | | Recall | |
| | Orig | CoT | Orig | CoT | Orig | CoT | Orig | CoT | Orig | CoT | Orig | CoT | Orig | CoT | Orig | CoT |
| DeepSeek-V3 | 0.46 | 0.40 | 0.42 | 0.34 | 0.45 | 0.38 | 0.39 | 0.31 | 0.51 | 0.51 | 0.49 | 0.44 | 0.51 | 0.51 | 0.47 | 0.39 |
| Gemini-2.5 Flash | 0.39 | 0.37 | 0.26 | 0.25 | 0.33 | 0.31 | 0.22 | 0.21 | 0.55 | 0.53 | 0.42 | 0.42 | 0.58 | 0.55 | 0.33 | 0.34 |
| Gemini-2.5 Pro | 0.44 | 0.42 | 0.36 | 0.35 | 0.42 | 0.39 | 0.32 | 0.31 | 0.47 | 0.50 | 0.33 | 0.42 | 0.45 | 0.51 | 0.26 | 0.36 |
| GPT-4.1 Mini | 0.41 | 0.39 | 0.30 | 0.12 | 0.37 | 0.21 | 0.25 | 0.08 | 0.49 | 0.50 | 0.33 | 0.26 | 0.48 | 0.49 | 0.25 | 0.18 |
| GPT-5 | 0.41 | 0.42 | 0.36 | 0.36 | 0.39 | 0.40 | 0.34 | 0.33 | 0.57 | 0.57 | 0.50 | 0.49 | 0.60 | 0.60 | 0.43 | 0.42 |

In contrast, the *Pairwise Selection* results in Table 31 are noticeably stronger, indicating that LLMs find it easier to decide "which of two factors is better" than to assign an absolute noise label. Here, GPT-5 clearly stands out: it achieves the highest and most stable performance across both markets, with reasonably good accuracy and F1 and only mild degradation when moving from CSI300 to SP500. GPT-4.1 Mini and the Gemini models also show usable (though weaker) pairwise behavior, and CoT prompting tends to help more often than hurt, especially by improving recall (the ability to pick the truly better factor in a pair). Overall, these trends suggest that current LLMs are much more reliable as *comparators* of factors than as absolute *classifiers* of noise, and that larger models (e.g., GPT-5) are the first to reach a level of factor evaluation that is meaningfully above random, particularly on the more liquid and diversified SP500 universe.

Table 31: *Pairwise Selection* factor-evaluation performance under Origin vs. CoT prompting for CSI300 and SP500.

| Model | CSI300 | | | | | | | | SP500 | | | | | | | |
|---|---|---|---|---|---|---|---|---|---|---|---|---|---|---|---|---|
| | Accuracy | | F1 | | Precision | | Recall | | Accuracy | | F1 | | Precision | | Recall | |
| | Orig | CoT | Orig | CoT | Orig | CoT | Orig | CoT | Orig | CoT | Orig | CoT | Orig | CoT | Orig | CoT |
| DeepSeek-V3 | 0.48 | 0.40 | 0.58 | 0.50 | 0.53 | 0.47 | 0.62 | 0.53 | 0.53 | 0.48 | 0.58 | 0.49 | 0.51 | 0.47 | 0.66 | 0.52 |
| Gemini-2.5 Flash | 0.50 | 0.46 | 0.55 | 0.50 | 0.56 | 0.51 | 0.54 | 0.49 | 0.48 | 0.48 | 0.49 | 0.50 | 0.47 | 0.46 | 0.52 | 0.54 |
| Gemini-2.5 Pro | 0.50 | 0.52 | 0.58 | 0.57 | 0.55 | 0.57 | 0.60 | 0.57 | 0.46 | 0.48 | 0.50 | 0.51 | 0.46 | 0.47 | 0.55 | 0.55 |
| GPT-4.1 Mini | 0.44 | 0.48 | 0.47 | 0.63 | 0.50 | 0.52 | 0.45 | 0.79 | 0.44 | 0.44 | 0.45 | 0.57 | 0.43 | 0.45 | 0.47 | 0.77 |
| GPT-5 | 0.64 | 0.66 | 0.66 | 0.68 | 0.70 | 0.72 | 0.63 | 0.65 | 0.52 | 0.55 | 0.54 | 0.56 | 0.51 | 0.53 | 0.58 | 0.60 |

Detail performance results after **supervised fine-tuning (SFT)** on **GPT-4.1 Mini** are provided in Table 32. SFT produces two very different trends across the tasks. For noise-classification, SFT generally harms performance: F1 and recall drop sharply in both markets, and even when accuracy rises, the model becomes more conservative and fails to detect true signals. This reflects typical overfitting to market-specific class patterns, especially because noise classification is fragile and heavily distribution-dependent. In contrast, pairwise factor evaluation shows strong and consistent gains after SFT especially in F1 score. The improvements also transfer well across markets: a model fine-tuned on CSI300 still boosts pairwise accuracy on SP500, and vice versa. This suggests that pairwise comparison captures more stable, structural relationships between factor constructions, such as monotonicity, window-scale interactions, and composition strength, which are less tied to any single market's return distribution. In other words, pairwise selection benefits from SFT because it teaches the model generalizable relative-ranking rules, while binary noise classification suffers because it forces the model to memorize market-specific noise patterns rather than learning universal structure.

Table 32: Performance of GPT-4.1 Mini and fine-tuned variants on two evaluation tasks: *Noise Classification* (left block) and *Pairwise Factor Evaluation* (right block). "O" means vanilla prompting while "C" stands for CoT.

| Market | Model | Noise Classification | | | | | | | | Pairwise Selection | | | | | | | |
| | | Accuracy | | F1 | | Precision | | Recall | | Accuracy | | F1 | | Precision | | Recall | |
| | | O | C | O | C | O | C | O | C | O | C | O | C | O | C | O | C |
| | GPT-4.1 Mini | 0.41 | 0.39 | 0.30 | 0.12 | 0.37 | 0.21 | 0.25 | 0.08 | 0.44 | 0.48 | 0.47 | 0.63 | 0.50 | 0.52 | 0.45 | 0.79 |
| **CSI300** | FT on CSI300 | 0.50 | 0.48 | 0.12 | 0.12 | 0.54 | 0.41 | 0.07 | 0.07 | 0.86 | 0.83 | 0.87 | 0.85 | 0.85 | 0.82 | 0.89 | 0.88 |
| | FT on SP500 | 0.47 | 0.51 | 0.04 | 0.14 | 0.20 | 0.57 | 0.02 | 0.08 | 0.50 | 0.50 | 0.55 | 0.55 | 0.56 | 0.56 | 0.54 | 0.54 |
| | GPT-4.1 Mini | 0.49 | 0.50 | 0.33 | 0.26 | 0.48 | 0.49 | 0.25 | 0.18 | 0.44 | 0.44 | 0.45 | 0.57 | 0.43 | 0.45 | 0.47 | 0.77 |
| **SP500** | FT on CSI300 | 0.52 | 0.53 | 0.21 | 0.16 | 0.62 | 0.75 | 0.13 | 0.09 | 0.64 | 0.64 | 0.68 | 0.68 | 0.60 | 0.60 | 0.79 | 0.80 |
| | FT on SP500 | 0.51 | 0.49 | 0.06 | 0.06 | 0.75 | 0.38 | 0.03 | 0.03 | 0.78 | 0.78 | 0.78 | 0.77 | 0.78 | 0.77 | 0.78 | 0.77 |

### E.4  ADDITIONAL RESULT OF SEARCHING TASK

In this section, we provide detailed results for three different search algorithms: Chain-of-Experience (CoE), Tree-of-Thought (ToT), and Evolutionary Algorithm (EA). Tables 33, 34, and 35 present the performance of various models across several evaluation metrics, including search cost, success rate, improved success, improved ratio, and diversity.

From the results of the CoE search (Table 33), we observe that Gemini-1.5-Flash-8b and Gemini-2.5-Flash lead in terms of success rate and improved ratio, demonstrating their efficiency in converging to high-quality factors. While DeepSeek-V3 shows solid performance with a high success rate of 0.97, it lags slightly behind in terms of the improved success ratio (0.73), highlighting the trade-off between search cost and the quality of the factors discovered.

In the ToT search (Table 34), Gemini-2.5-Flash achieves the highest performance with an improved success ratio of 0.59 and a success rate of 0.95, closely followed by Gemini-2.5-Pro and GPT-5. This shows that the ToT approach, which explores multiple candidate factors in parallel, provides a good balance between exploration and exploitation, especially for larger models like GPT-5. However, smaller models like LLaMA3.1-70b-Instruct tend to perform less consistently in the ToT setting, which may be due to its limited capacity for handling the complexity of the search task.

Finally, the EA search (Table 35) shows that Gemini-2.5-Flash excels with a perfect success rate of 1.00, leading the models in both improved success and best update rate, indicating that evolutionary algorithms can effectively optimize factor pools. GPT-5 also performs well in this setup, with an improved ratio of 0.91, although LLaMA3.1-70b-Instruct lags behind in terms of search cost and success rate, which may indicate that the model struggles with the continuous refinement process in an EA framework.

Table 33: Results for **CoE** search. Quality includes search cost and success rate; performance includes improved success, improved ratio, and diversity. Values rounded to 2 decimals.

| Model | Quality | | Performance | | |
|---|---|---|---|---|---|
| | Search Cost | Success Rate | Improved Success | Improved Ratio | Diversity |
| DeepSeek-V3 | 1.04 | 0.97 | 0.73 | 0.08 | 0.36 |
| Gemini-1.5-Flash-8b | 1.43 | 1.00 | 0.73 | 0.24 | 0.44 |
| Gemini-2.5-Flash | 1.08 | 0.96 | 0.87 | 0.36 | 0.48 |
| Gemini-2.5-Pro | 1.11 | 0.95 | 0.80 | 0.44 | 0.55 |
| GPT-4.1-Mini | 1.21 | 0.81 | 0.67 | 0.24 | 0.46 |
| GPT-5 | 1.36 | 0.73 | 0.80 | 0.44 | 0.50 |
| LLaMA3.1-70b-Instruct | 1.02 | 0.98 | 0.73 | 0.12 | 0.36 |

Table 34: Results for **ToT** search. Quality includes search cost and success rate; performance includes improved success, improved ratio, and diversity. Values rounded to 2 decimals.

| Model | Quality | | Performance | | |
|---|---|---|---|---|---|
| | Search Cost | Success Rate | Improved Success | Improved Ratio | Diversity |
| DeepSeek-V3 | 2.66 | 0.94 | 0.80 | 0.39 | 0.88 |
| Gemini-1.5-Flash-8b | 3.00 | 0.91 | 0.80 | 0.61 | 0.86 |
| Gemini-2.5-Flash | 2.55 | 0.95 | 0.87 | 0.59 | 0.71 |
| Gemini-2.5-Pro | 2.58 | 0.94 | 0.87 | 0.63 | 0.70 |
| GPT-4.1-Mini | 1.25 | 0.95 | 0.87 | 0.59 | 0.69 |
| GPT-5 | 1.04 | 0.99 | 0.87 | 0.66 | 0.70 |
| LLaMA3.1-70b-Instruct | 1.52 | 0.89 | 0.87 | 0.50 | 0.80 |

Table 35: Results for **EA** search. Quality includes search cost and success rate; performance includes best update rate, improved ratio, and diversity. Values rounded to 2 decimals.

| Model | Quality | | Performance | | |
|---|---|---|---|---|---|
| | Search Cost | Success Rate | Best Update Rate | Improved Ratio | Diversity |
| DeepSeek-V3 | 2.80 | 0.95 | 0.08 | 0.35 | 0.63 |
| Gemini-1.5-Flash-8b | 2.25 | 0.98 | 0.28 | 0.70 | 0.70 |
| Gemini-2.5-Flash | 2.15 | 1.00 | 0.29 | 1.00 | 0.53 |
| Gemini-2.5-Pro | 2.70 | 0.97 | 0.30 | 0.73 | 0.63 |
| GPT-4.1-Mini | 1.50 | 0.89 | 0.35 | 0.90 | 0.47 |
| GPT-5 | 1.25 | 0.96 | 0.48 | 0.91 | 0.45 |
| LLaMA3.1-70b-Instruct | 2.45 | 0.86 | 0.50 | 0.94 | 0.49 |

We further analyze the effect of temperature in Figure 14 and the impact of capacity in a single round of EA searching, as shown in Figure 15.

In terms of temperature, as illustrated in Figure 14, the results reveal that the choice of temperature significantly influences the performance across different search algorithms. Lower temperature values (0.75) generally lead to more stable and consistent performance, as reflected in higher success rates and improved ratios for most models. Conversely, higher temperatures (1.5) tend to increase diversity but often at the cost of reduced search efficiency and success rate, suggesting a trade-off between exploration and exploitation.

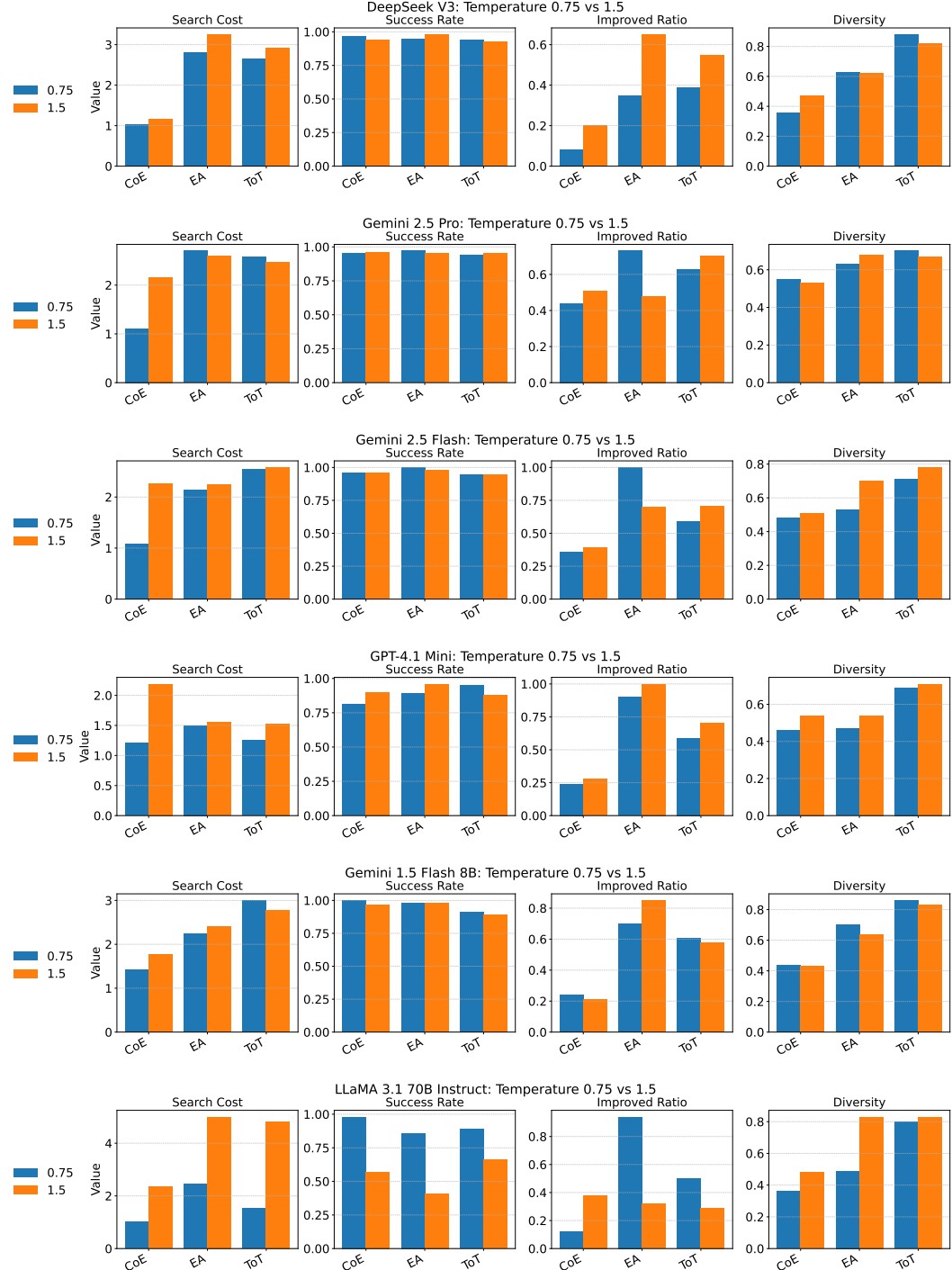

Figure 14: Temperature 0.75 vs 1.5: Search Cost, Success Rate, Improved Ratio, and Diversity across models.

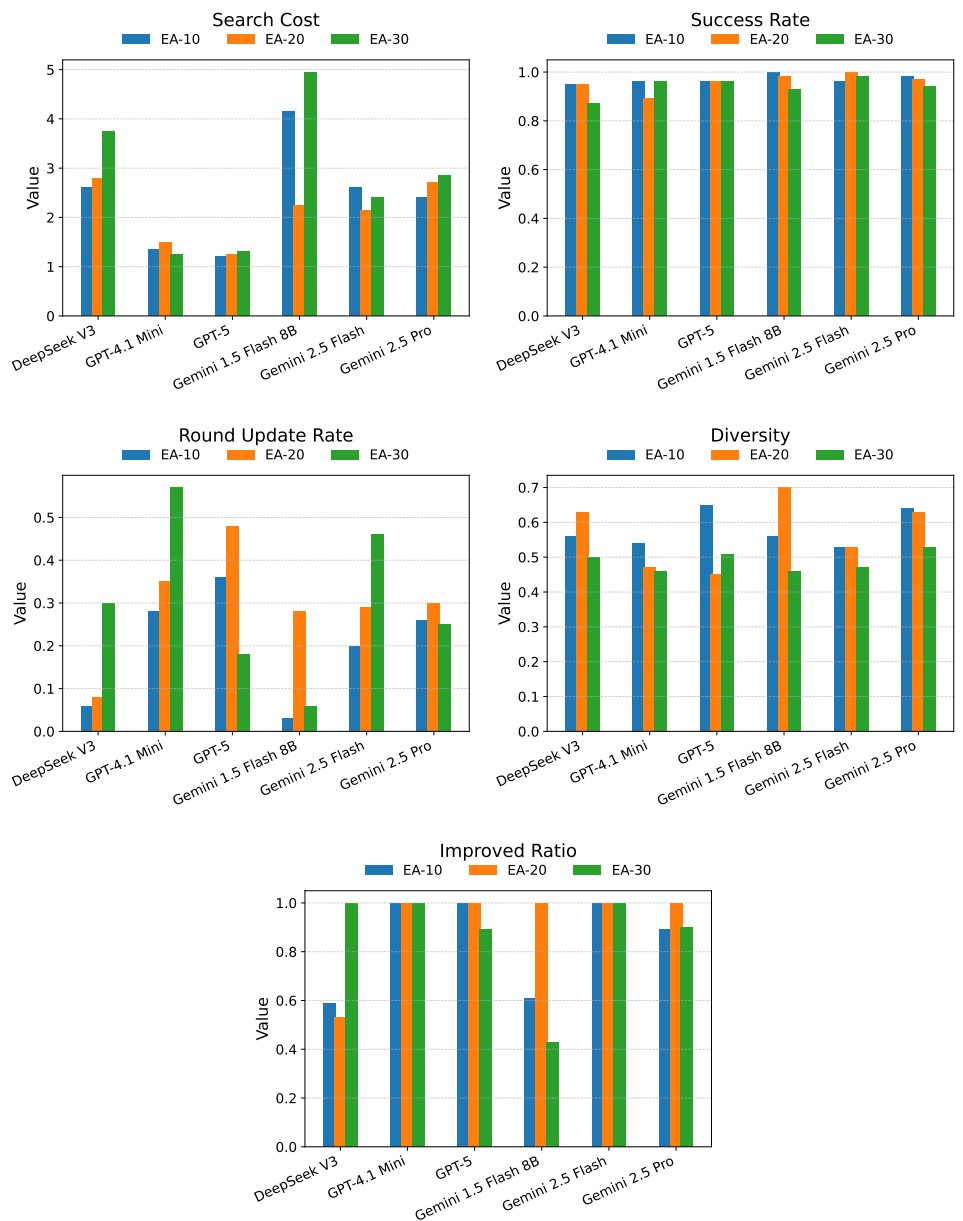

Figure 15: Evaluation of EA search capacity under different round sizes (10, 20, 30 factors generated per round). Metrics reported include search cost, success rate, round update rate, diversity, and improved ratio across six representative LLMs (DeepSeek V3, GPT-4.1 Mini, GPT-5, Gemini-1.5 Flash-8B, Gemini-2.5 Flash, Gemini-2.5 Pro).

### E.5 ANALYSIS OF FAILURE CASES

**Failure Analysis in Generation and Searching.** To better understand the limitations of LLMs, we conduct a systematic analysis of failure cases in factor generation and searching. We categorize the errors into four representative types, with their cases in Table 36:

*(1) Output format errors.* These arise when the model produces results that deviate from the prescribed structured format, rendering them unparsable.

*(2) Syntax errors.* These occur when the generated expressions are not syntactically valid. A common pattern, particularly in the DeepSeek series, is the omission of closing parentheses or the presence of malformed operator calls.

*(3) Invalid expressions.* In this case, the expression contains unsupported operators outside the provided guidelines, or applies valid operators incorrectly—for instance, by using arguments with invalid ranges or mismatched arity.

*(4) Low-quality expressions.* These refer to cases where the generated formulas are excessively complex (e.g., nesting depth greater than five), computationally inefficient (requiring excessive runtime), or unstable, producing a proportion of NaN values that exceeds the admissible threshold.

Table 36: Examples of Common Errors in Alpha Factor Expressions

| Error Type | Example |
|---|---|
| **Output Format Errors** | Expression: $Add(\$close, \$volume$ 
 Error: Missing closing parenthesis. |
| **Syntax Errors** | Expression: $Mean(\$close, 5) + Exp(\$close)$ 
 Error: Missing closing parenthesis for Exp. |
| **Invalid Expressions** | Expression: $Divide(\$close, \$volumne)$ 
 Error: Invalid operator **Divide**, it should be **Div**. |
| **Low-Quality Expressions** | Expression: $Add(Exp(Add(Exp(\$close))))$ 
 Error: Excessive nesting of Exp leads to high computational inefficiency. |

In addition, we observe that some LLMs (such as GPT-4.1-Mini and LLaMA3.1-70B-Instruct) experience the "word salad phenomenon" (shown in Figure 16) during the generation task, particularly when Chain-of-Thought (CoT) is enabled. This issue typically arises when generating content using CoT and in high temperature setting, leading to incoherent or disjointed outputs.

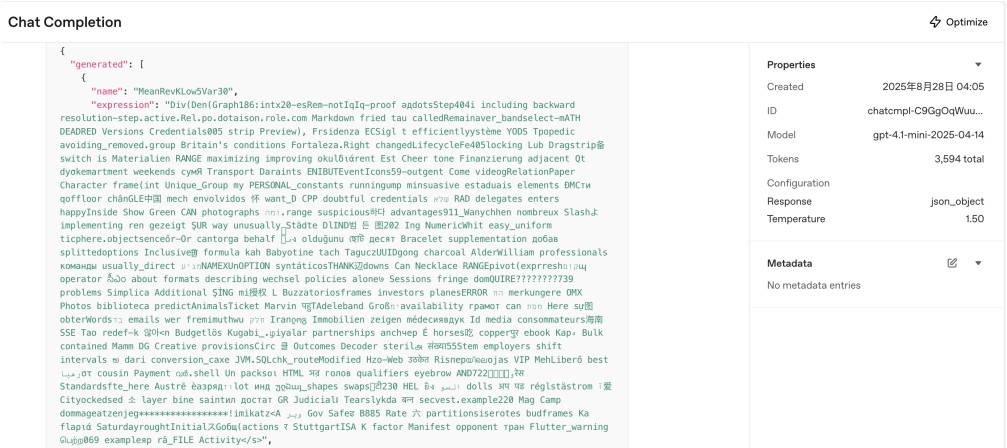

Figure 16: An example of "word salad phenomenon" by GPT-4.1-Mini in factor searching task

