# OpenReview forum: "AlphaBench: Benchmarking Large Language Models in Formulaic Alpha Factor Mining"
_ICLR.cc/2026/Conference — ICLR 2026 Poster_

### Official Review · Reviewer_4Cyb · 2025-10-31

**Soundness:** 3
**Presentation:** 4
**Contribution:** 3
**Rating:** 6
**Confidence:** 5

**Summary:**

The paper introduces a benchmark to evaluate how large language models perform in quantitative finance tasks involving the generation, evaluation, and searching of interpretable factors. Using the CSI300 market data and Qlib framework, the benchmark tests models across metrics such as reliability, stability, accuracy, and cost. Results show that while LLMs can reliably produce syntactically valid and intuitive factor expressions, they struggle to assess factor quality without data, and reasoning strategies offer limited benefit.

**Strengths:**

The paper is well-written and clearly organized. I commend the authors for the scope of their analyses. As a potential user of AlphaBench, I believe that the benchmark dataset will be of interest to many researchers.

It is good to see that frontier models already achieved a supreme performance in generations tasks. The models also achieved a moderate performance in search tasks.

**Weaknesses:**

My concerns are primarily related to the evaluation task. Given its statistical nature, probably LLMs are not well-suited for factor evaluation. LLMs might help human quant researchers to construct creative new factors but it might be better to outsource quantitative assessment to traditional backtesting models.

Section 4.3 reports very weak zero-shot evaluation performance, which is near random. The discussion (Section 5) attributes this mainly to missing supervision and execution context. Although this explanation is convincing, the paper can further test the following:
(1) Provide a controlled ablation showing how much of this failure is due to lack of numeric context versus model scale or prompt design.
(2) Demonstrate even a small fine-tuning (e.g., adding example factor–IC pairs) to show whether performance improves. This would make the benchmark actionable for future model development.

The authors implicitly make two foundational assumptions:
 (1) LLMs can “understand” factor expressions semantically, and
 (2) they can reason about predictive strength without data
These assumptions are not empirically grounded. Alpha factors embed statistical relations, not linguistic ones; their performance often depends on noise sensitivity, normalization, market characteristics, time-specific regimes, and data-specific effects that linguistic/symbolic interpretation cannot capture. Hence, the current design may be more a test of syntactic familiarity (e.g., knowing that Mean + Std are stable factors) rather than genuine financial reasoning.
This can only be tested if the authors can show that human experts (quant researchers) can perform the same ranking/scoring from formulas alone (without access to numerical data). This would establish an upper bound for LLM expectations.

Currently, if I did not misunderstand, ground truth labels are derived from backtests on the same CSI300 data used to generate the factors. If the LLM learned statistical patterns from similar data distributions (e.g., public quant research corpora), then its apparent understanding could partially reflect memorization, not reasoning. A stronger design would use out-of-sample or cross-market validation (e.g., use CSI300 for training labels and SP500 for testing).

Another potential concern is that the paper’s contributions may appeal to a financial AI audience rather than the broader general AI community. It is difficult to identify insights that generalize beyond the domain of quantitative finance. The authors could strengthen the paper’s broader relevance by adding a sentence or two in the conclusion to highlight how AlphaBench’s methodology can inform LLM benchmarking or reasoning research in other structured domains.

**Questions:**

Please see above (weaknesses).

---

> ### Author Response · Authors · 2025-11-19
> **LLM Evaluation Challenges, SFT Insights, and Cross-Domain Relevance**
>
> **1. On whether LLMs are suitable for factor evaluation**
>
> We fully agree with the reviewer: **LLMs are not well-suited for absolute factor evaluation**, because factor performance is a statistical outcome, not a linguistic property. Our design of the atomic tasks was precisely to test this assumption, and the results confirm it:
>
> * **Zero-shot factor evaluation is near random** (Sec. 4.3), consistent with the reviewer’s observation.
> * **Fine-tuning does not meaningfully improve noise classification**, showing that the difficulty is intrinsic to the task and not caused by poor prompting or insufficient supervision.
> * However, **fine-tuning is helpful for pairwise selection**, where relative structural comparison is more learnable.
>
> These findings align with the reviewer’s intuition that while LLMs can assist in generating or modifying factor structures, **factor evaluation at currecrt stage may still rely on backtesting**. At the same time, our work contributes concrete evidence toward understanding why LLMs struggle in this setting and provides a systematic, extensible evaluation protocol, that future research can build upon.
>
> **2. On controlled ablations, numeric context, and small-scale fine-tuning**
>
> We appreciate the reviewer’s suggestion for more controlled ablations. In the revised version, we incorporate several additional analyses:
>
> * **Small-scale supervised fine-tuning** using a limited set of factor–IC pairs.
>   The results show:
>
>   * **clear gains on pairwise selection**, validating that the benchmark can support future model development;
>   * **minimal improvement on absolute noise classification**, indicating the fundamental difficulty of inferring statistical strength from formulas alone.
>
> * **Cross-market SFT experiments** (CSI300 → S&P500, and S&P500 → CSI300) to mitigate concerns about memorization or distribution leakage.
>   These experiments demonstrate asymmetric transfer behavior, suggesting that models are learning structural patterns rather than memorizing factor–IC associations.
>
> Regarding the searching task, in response to the reviewer’s concern about market-specific bias, we have extended our experiments to include **searching tests on S&P500**. In the further stage, we will also add **cross-market search evaluation**, for example initializing searchers on CSI300 and evaluating gains on S&P500 (and vice versa). This better reflects robustness and helps isolate true search capability from dataset-specific noise.
>
> **3. On broader relevance beyond finance**
>
> We thank the reviewer for highlighting the need to clarify broader impact. In the revised conclusion, we explicitly emphasize that AlphaBench’s methodology generalizes to any **executable-DSL reasoning** setting, including:
>
> * spreadsheet and analytics formulas,
> * symbolic regression and equation discovery,
> * rule-based analytical pipelines.
>
> The core paradigm **LLMs generate structured DSL expressions, and objective executors evaluate them**, extends naturally beyond quantitative finance. We have updated the manuscript to make this relevance more explicit.

---

### Official Review · Reviewer_QP1w · 2025-11-01

**Soundness:** 3
**Presentation:** 3
**Contribution:** 2
**Rating:** 6
**Confidence:** 4

**Summary:**

This paper introduces AlphaBench, the first benchmark for evaluating LLMs in Alpha Factor Mining, where it's an important task in quantitative finance focused on discovering interpretable mathematical expressions that predict asset returns.

The benchmark includes: 687 generation prompts, 1170 evaluation instructions, 27 search tasks, 10 LLMs, and different prompting strategies. AlphaBench covers comprehensive evaluation metrics for Generation, Evaluation, and Searching.

**Strengths:**

1. The benchmark focuses on the factor mining task, but is comprehensive. The paper provides lots of insights: a) LLMs show high reliability in generating syntactically valid factors, but accuracy drops significantly for complex instructions. b) Factor evaluation remains a major bottleneck. 3) LLM-guided search improves factor quality at a reasonable computational cost; d) CoT sometimes doesn't work on large models. That's important for the FinTech domain.

2. The authors build on a real-market dataset using Qlib-compatible backtesting. They also provide detailed documentation for the curated dataset.

**Weaknesses:**

1. I have some confusion about the factor evaluation task. The factor evaluation (ranking/scoring) task sometimes outputs near-random results across all models. Could the author provide more explanation on that?

2. The dataset covers only daily equity factors on the CSI-300 (China) with ~1,700 factors.

3. The search step is helpful, and search quality is measured via IC improvement from backtesting. But this could vary due to noise in short-term financial returns.

**Questions:**

See in Weaknesses

---

> ### Author Response · Authors · 2025-11-19
> **On Evaluation Difficulty, Dataset Scale, and Search Robustness**
>
> **1. On why factor evaluation results appear near-random**
>
> We appreciate the reviewer’s question. The near-random performance in factor evaluation (ranking/scoring) is expected and consistent across all models. This stems from the nature of factor evaluation itself:
>
> - **Factor performance is inherently statistical**, depending on long-horizon return dynamics, regime shifts, cross-sectional dispersion, and execution details that are *not encoded in the formula*.
> - **Short-window returns are highly noisy**, andto best of our knowledge, there is no public traditional models can reliably predict a factor’s IC trajectory from its symbolic structure alone.
>
> From the atomic tasks we include in the revised version. Across **both CSI300 (CN)** and **S&P500 (US)**:
>
> - Absolute noise classification stays near-random
> - Prompting, AST-style structure, and added market context do not help
> - Even supervised fine-tuning fails to significantly improve classification
>
> These results reinforce that the limitation of LLM lies in the task.
>
> **2. On dataset scope**
>
> The reviewer notes that the dataset contains ~1,700 daily factors on CSI300. In the revised version, we clarify the following:
>
> - Our full system at current stage has generated **over 10,000 factors** with IC-based statistics, far beyond the original ~1,700 used in early experiments.
> - AlphaBench can be designed as a **scalable data generator**: the searching suite continuously produces new *(factor, metric)* pairs during generation and backtesting.
> - We additionally incorporate **S&P500 (US)** for cross-market analysis in the revised atomic tasks to avoid market-specific bias and demonstrate generality.
>
> **3. On noise in short-term returns and search-quality measurement**
>
> We agree that short-term financial returns are noisy. This is precisely why backtesting is indispensable and why factor evaluation cannot rely solely on formula structure or LLM intuition. In **AlphaBench**, we evaluate searchers on a *minimal* and *controlled* search task:
>  **the goal is to improve performance on the current testing window using only static rolling-window backtest data.**
>  This isolates the search ability itself without adding external stabilization layers.
>
> In practical quantitative workflows (Tang et al., 2025; Shi et al., 2025 ), however, factor-search pipelines typically apply **post-processing steps** such as longer-horizon validation, regime checks, or stability filters, to further mitigate short-term noise. Our framework intentionally separates these issues:
>
> - The LLM acts as a **structure-based generator**.
> - The backtest provides **statistical grounding** within the current window.
> - Real-world systems may incorporate additional filtering steps, but our evaluation focuses on the searcher’s intrinsic ability to improve rolling-window IC.
>
>
>
> Tang, Ziyi, et al. "Alphaagent: Llm-driven alpha mining with regularized exploration to counteract alpha decay." *Proceedings of the 31st ACM SIGKDD Conference on Knowledge Discovery and Data Mining V. 2*. 2025.
>
> Shi, Hao, et al. "Alphaforge: A framework to mine and dynamically combine formulaic alpha factors." *Proceedings of the AAAI Conference on Artificial Intelligence*. Vol. 39. No. 12. 2025.

---

### Official Review · Reviewer_Mm9t · 2025-11-01

**Soundness:** 3
**Presentation:** 3
**Contribution:** 3
**Rating:** 6
**Confidence:** 3

**Summary:**

1. The paper presents - AlphaBench which introduces first benchmark for evaluating LLMs in (FAFM) which is pretty noval and insightful
2. Three main pillars at which model measures are- generation, evaluation, and searching tasks for financial factors and their discovery.
3. A far I could evaluate in manuscript - GPT-5 performs best overall;
4. Chain-of-Thought (CoT) yields minimal gains which was also expected, sometimes reduces stability.
5. For all models, evaluation or judgement remains a big challenge
6. Gemini models are competitive; open-source models lag behind.

**Strengths:**

1. A very detailed and comprehensive coverage of FAFM lifecycle.
2. Very good benchmarking with quantitative metrics (IC etc..)
3. The paper also presents a very strong validation on real financial datasets (CSI300, 2020–2025).
4. Manuscript also rightly highlights trade-offs between model size, cost.

**Weaknesses:**

1. LLMs struggle in factor evaluation — low accuracy in ranking/scoring that was observed on some experiments
2. CoT prompting often hurts large-model performance.
3. Lack of supervised data limits evaluation reliability. Some strong large scale data would be helpful
4. Benchmark restricted to daily equity factors; excludes intraday or multi-asset tests.

**Questions:**

1. How can supervised or weakly labeled data be created for training factor evaluators?
2. Can structured representations improve evaluation interpretability and what experiments can be done more on it?
5. What is the role of specific domain knowledge in guiding LLM outputs?

---

> ### Author Response · Authors · 2025-11-19
> **On Data Generation, Structured Inputs, and Domain Constraints in AlphaBench**
>
> **(1) On creating supervised or weakly labeled data for training factor evaluators**
>
>  • We have already constructed a dataset containing more than 10,000 factors with their IC/RIC-based performance statistics. Moreover, the searching suite in AlphaBench continuously generates additional (factor, performance) pairs during factor generation and backtesting.
>  • This means AlphaBench naturally acts as a scalable *data generator* for future supervised training. Researchers can easily extract labeled examples from any stage of the factor evolution pipeline, enabling both fully supervised and weakly supervised learning setups.
>  • We will clarify this contribution in the revised manuscript and emphasize that AlphaBench provides *both* high-quality data and a procedure for expanding labeled datasets on demand.
>
> **(2) On whether structured representations improve interpretability and what further experiments can be done**
>
>  • We evaluated minimal atomic tasks: (i) noise vs. non-noise factor classification, and (ii) noise vs. non-noise A–B selection. These tasks reveal how well LLMs understand factor semantics with limited supervision.
>  • We incorporated structured representations by converting factor formulas into AST-style operator trees. An example is shown below for the factor `Div(Skew(Delta(Log($close), 1), 22), Add(Med($volume, 22), 1e-12))`.
>
> The AST for it:
>
> ```
> NODE(Div,
>   CHILD(NODE(Skew,
>     CHILD(NODE(Delta,
>       CHILD(NODE(Log,
>         CHILD(NODE($close))
>       )),
>       CHILD(NODE(1))
>     )),
>     CHILD(NODE(22))
>   )),
>   CHILD(NODE(Add,
>     CHILD(NODE(Med,
>       CHILD(NODE($volume)),
>       CHILD(NODE(22))
>     )),
>     CHILD(NODE(1e-12))
>   ))
> )
> ```
>
>  • However, in our experiments, these tree-structured inputs did **not** lead to noticeable improvements in classification or selection accuracy. Our interpretation is that text-based ASTs do not sufficiently encode compositional and quantitative relationships.
>  • Future directions include testing *graph-structured encoders*, Graph-LLMs, or symbolic program embedding models that can better leverage operator topology and numeric relationships. Additionally, richer supervision (e.g., factor decomposition rationales or multi-step factor rewriting tasks) may reveal whether structure helps in more complex settings.
>
> **(3) On the role of domain knowledge and its effect on LLM outputs**
>
>  • We introduced market-environment context (e.g., bull/bear regime, volume/volatility summaries) as additional inputs in the noise-classification task. Empirically, this *did not* improve classification performance. For example,
>
> ```
> US_MARKET_STYLE = """US large-cap (S&P 500) style card:
> • Microstructure: decimalized quotes, no daily price limits, short-selling widespread, deep liquidity, pre/post-market sessions, HFT common.
> • Typical pitfalls: 1-day gap/open artifacts; raw ranges without scaling; indicator clones; overfit microstructure wiggles.
> • More credible patterns: multi-horizon momentum vs short-term reversal; regime filters (vol/turnover); ranking & volatility scaling; structured price–volume interactions with smoothing."""
>
> CN_MARKET_STYLE = """China A-shares (CSI 300) style card:
> • Microstructure: daily price limits (≈±10% typical), midday break, T+1 selling rule, short-selling constraints, higher retail participation, frequent limit-up/down.
> • Typical pitfalls: raw gaps/limit touches without normalization; noon-break artifacts; scale effects (price/volume level) without comparability.
> • More credible patterns: regime-aware momentum/reversal that handle limits; turnover/volatility filters; rank/vol-scaling; fast/slow window interactions and robust smoothing."""
> ```
>
>  • This suggests that binary noise detection is fundamentally difficult when only symbolic formula structure is available, and domain signals alone are insufficient for guiding LLMs without more detailed supervision under evaluation task.
>
>  • In other factor-generation/searching experiments, LLMs operate within a Qlib-style formulaic DSL. The predefined operator set, structural rules, and admissible function signatures play a critical role in constraining the output space. These domain-specific restrictions ensure that generated expressions follow a unified syntax, prevent invalid or non-executable formulas, and significantly reduce generation drift. Without such structured domain knowledge, LLM outputs become unstable and deviate from valid factor formats, making downstream evaluation and evolution impossible.

---

### Author Response · Authors · 2025-11-19
**Common Response to Reviewers: On Evaluation Tasks and Broader Implications**

We sincerely thank all reviewers for the constructive and insightful feedback. Several comments across the reviews raised similar questions regarding: (1) the feasibility of LLM-based factor evaluation, (2) the design and purpose of our atomic tasks, and (3) the broader utility of AlphaBench as a data and evaluation framework. We summarize the shared concerns and our unified clarifications below.

### **1. Why design atomic evaluation tasks?**

Across multiple reviews, there was interest in whether LLMs can act as *fast evaluators*, either as partial replacements for backtesting or as inexpensive early-stage filters during factor search.

To study this question cleanly, we introduced two *minimal, well-controlled* atomic tasks:

1. **Noise vs. non-noise factor classification**
2. **A–B pairwise selection (choose the better factor)**

These tasks remove confounding variables and test the core ability of LLMs to infer factor quality *solely from structure*. They also allow us to systematically study whether additional supervision, structured representations (AST trees), or contextual information can help.

### **2. What did these tasks reveal?**

The results across all reviewers’ questions consistently show:

- **Absolute noise classification is extremely difficult** for current LLMs
   → Performance is near-random even after careful prompting and structured input.
- **Pairwise selection is learnable**
   → Fine-tuning improves performance, and cross-market transfer is possible.
- **AST-style structured representations do not significantly help**, suggesting that text-tree structures may not capture factor compositionality effectively.
- **Adding market context does not improve classification**, confirming that the difficulty lies in the inherent statistical nature of factor performance, not in prompt design.

These findings support a unified conclusion:
 **LLMs can assist with structure-based generation and generation, but cannot yet replace statistical backtesting.**

### **3. Why expand evaluation tasks in the revised version?**

Several reviewers asked whether our evaluation suite is sufficient to characterize the limitations of LLMs.
 To address this, we have expanded the evaluation in the revised manuscript:

- Both CSI300 and S&P500 datasets included in all atomic tasks
- Fine-tuning and zero-shot comparisons across markets
- Cross-market transferability analysis
- Tests with structured operator-tree representations
- Additional reasoning about why noise classification remains fundamentally statistical

---

### Author Response · Authors · 2025-12-01
**Summary of reviewer's question and our response**

Dear ICLR 2026 AC, SAC, PC and all reviewers,

Thanks for all of yours effort in this period. First, we would like to restate the main contributions of our work. To our knowledge, this is the **first systematic benchmark** for evaluating LLMs in *formulaic alpha factor mining*, an important task in quantitative investment. We provide a complete evaluation suite and unified standards, highlighting both the strengths and weaknesses of current LLMs. In addition, **AlphaBench** includes an end-to-end data pipeline for the factor-searching task: our search and backtesting modules automatically generate large volumes of paired *factor–performance* data, which can be directly used to develop specialized models for this domain, including both generators and evaluators. We believe our work offers meaningful insights for future research not only in quantitative finance but also in related areas with similar characteristics, such as symbolic regression.

We also acknowledge that reviewers’ concerns focused mainly on the evaluation methodology. In response, we conducted additional experiments to provide clearer evidence and deeper insights. Specifically, we decomposed the original evaluation into two **atomic subtasks**, allowing more precise measurement of LLM performance. The new results further reveal the detailed failure modes of current LLMs. We additionally explored fine-tuning approaches, such as supervised fine-tuning, and report new findings and observations in the revised version.

Finally, we will continue to update AlphaBench in future versions, including expanded experiments on **cross-market generalization** and broader evaluation protocols.

---

### Meta-Review · Area_Chair_yJGs · 2025-12-17

**Summary:**

This paper develops a new benchmark to evaluate how large language models (LLMs) perform in quantitative finance tasks, especially in generating, evaluating, and searching of interpretable factors. This might be the first systematic benchmark for evaluating LLMs in formulaic alpha factor mining (FAFM). Extensive evaluations and discussions are presented in the paper.

Reviewers agreed that this paper study a timely and important problem in the FinTech domain. The paper is clearly organized and well written. The proposed benchmark is very comprehensive, and the paper provides some meaningful insights. Meanwhile, reviewers raised several concerns regarding evaluation methodology, dataset coverage, reliability of the search step, controlled ablation studies, etc.

**Reviewer Concerns:**

The authors have provided detailed responses to address concerns on evaluation methodology and ablation studies. In particular, the authors conducted two well-controlled atomic tasks: (1) noise vs. non-noise factor classification, and (2) A–B pairwise selection (choose the better factor). They have also expanded evaluation tasks in the revised paper, such as including both the CSI300 and S&P500 datasets in the evaluations. I think the concerns from reviewers have been well addressed by the authors' responses.

**Reviewer Scores:**

I anticipate that at least 1-2 reviewers would have changed their score from 6 to 8, given the responses from reviewers.

---

### Decision · Program_Chairs · 2026-01-26

Accept (Poster)